# Assessing the global contribution of marine, terrestrial bioaerosols, and desert dust to ice-nucleating particle concentrations

Marios Chatziparaschos[1,2,*], Stelios Myriokefalitakis[3], Nikos Kalivitis[1], Nikos Daskalakis[4], Athanasios Nenes[2,5], María Gonçalves Ageitos[6,7], Montserrat Costa-Surós[6], Carlos Pérez García-Pando[6,8], Mihalis Vrekoussis[4,9,10] and Maria Kanakidou[1,2,4]

[1]Environmental Chemical Processes Laboratory (ECPL), Department of Chemistry, University of Crete, Heraklion
[2]Center for the Study of Air Quality and Climate Change (C-STACC), Institute of Chemical Engineering Sciences (ICE-HT), Foundation for Research and Technology, Hellas) (FORTH), Patras, Greece
[3]Institute for Environmental Research and Sustainable Development, National Observatory of Athens (NOA), GR-15236 Palea Penteli, Greece
[4]Laboratory for Modelling and Observation of the Earth System (LAMOS), Institute of Environmental Physics (IUP), University of Bremen, Bremen, Germany
[5]Laboratory of Atmospheric Processes and their Impacts (LAPI), School of Architecture, Civil and Environmental Engineering (ENAC), Ecole Polytechnique Federale de Lausanne, Lausanne, Switzerland
[6]Barcelona Supercomputing Center (BSC), Barcelona, Spain
[7]Department of Project and Construction Engineering, Universitat Politècnica de Catalunya (UPC), Barcelona, Spain
[8]Catalan Institution for Research and Advanced Studies (ICREA), Barcelona, Spain
[9]Center of Marine Environmental Sciences (MARUM), University of Bremen, Germany
[10]Climate and Atmosphere Research Center (CARE-C), The Cyprus Institute, Nicosia, Cyprus
*Now at Barcelona Supercomputing Center (BSC), Barcelona, Spain

Correspondence: Maria Kanakidou (mariak@uoc.gr)

**Abstract**. Aerosol-cloud interactions, and particularly ice processes in mixed-phase clouds (MPC), remain a key source of uncertainty in climate change assessments. This study introduces state-of-the-art laboratory-based parameterizations into a global chemistry-transport model to investigate the contributions of mineral dust (specifically K-feldspar and quartz), marine primary organic aerosol (MPOA), and terrestrial primary biological aerosol particles (PBAP) to ice nucleating particles (INP) in MPC. The model suggests that INP originating from PBAP ($INP_{PBAP}$) is the primary source of INP at low altitudes between -10ºC and -20ºC, particularly in the tropics, with a pronounced peak in the Northern Hemisphere (NH) during the boreal summer. $INP_{PBAP}$ contributes over 40% of the total simulated INP column burden at mid-latitudes. Dust-derived INP ($INP_D$) is prominent at high altitudes across all seasons, dominating at temperatures below -20°C, and constitutes over 89% of the INP average column burden at high latitudes in the NH and about 74% at high latitudes in the Southern Hemisphere (SH). MPOA-derived INP ($INP_{MPOA}$) prevails in the SH at low altitudes, particularly at subpolar and polar latitudes for temperatures above -20ºC, where they represent between 17% and 36% of the INP column population, depending on the season. When evaluated against available global observational INP data, the model achieves its highest predictability across all temperature ranges when both $INP_D$ and $INP_{MPOA}$ are included as independent INP sources. The addition of $INP_{PBAP}$ does

not enhance the model's ability to reproduce the available observations; however, INP$_{PBAP}$ remains a key contributor to warm-temperature ice nucleation events. Therefore, consideration of dust, marine and terrestrial bioaerosols as distinct INP species is required to simulate ice nucleation in climate models.

## 1 Introduction

Cloud ice exerts a multifaceted influence on climate (Zhang et al., 2020), affecting the radiative balance
(Zhou et al., 2016; Yi, 2022), climate feedbacks (e.g., temperature, albedo, water vapor feedbacks) (Choi et al., 2014), precipitation (Sorooshian et al., 2009), and climate sensitivity (Murray et al., 2021). In mixed-phase clouds (MPC), ice particles and cloud droplets coexist in an unstable thermodynamic equilibrium owing to subzero temperatures. A significant feature of MPC is the degree of homogeneity in mixing of ice particles and liquid droplets that affects the rate of precipitation formation (Korolev and Milbrandt, 2022).
These clouds are prevalent throughout the troposphere, observed across all latitudes, spanning from polar regions to the tropics (D'Alessandro et al., 2019). As a result, they significantly impact precipitation rates and the radiative energy balance on both regional and global scales (Hofer et al., 2024).

In MPC, the most important ice formation process is immersion freezing, where an ice-nucleating particle (INP) becomes immersed in a supercooled droplet and initiates freezing, typically occurring between -5°C
and -35°C (Pruppacher et al., 1998; Kanji et al., 2017 and references therein). Contact freezing happens when an INP collides with a supercooled droplet, triggering freezing on contact, often in turbulent conditions. This process is defined as separate from immersion freezing because of empirical evidence that some INP are more effective in this mode than when immersed in liquid (Shaw et al., 2005). Condensation freezing occurs when water vapor condenses and freezes simultaneously upon contact with an INP under specific
supersaturation conditions. This process is less frequent compared to immersion freezing in atmospheric clouds (Vali et al., 2015). Deposition nucleation, where water vapor directly deposits as ice onto an INP without passing through the liquid phase, is more relevant at much colder temperatures (below -20°C) and is less significant in the typical temperature range of mixed-phase clouds (Kanji et al., 2017 and references therein) but it might still be important for cirrus clouds (Cziczo et al., 2013). Once formed, ice particles in
MPC can grow at the expense of evaporating cloud droplets when the ambient water vapor pressure lies between the saturation water vapor pressure with respect to ice ($e_{s,i}$) and water ($e_{s,w}$). This is known as the Wegener-Bergeron-Findeisen (WBF) process, which is caused by the difference in supersaturation between the liquid and ice phases (Pruppacher et al., 1998), with the latter being consistently lower. Other factors that alter ambient water vapor pressure are the vertical velocity as a source of water vapor, and the integrated ice
crystal surface area, which depletes supersaturation by consuming the available water vapor (Korolev, 2007). For the WBF process to occur, initial ice formation is required, which can happen through heterogeneous freezing at relatively warmer temperatures (Murray et al., 2012), a process that is crucial for ice formation in MPC. The presence of aerosols is necessary for heterogenous freezing since they provide a surface for the ice to form on. These aerosols called ice nucleating particles (INP), a select subgroup of aerosol particles -

approximately 1 in $10^5$ particles in continental air - can activate at temperatures warmer than -30°C (DeMott et al., 2010) - that initiate ice formation heterogeneously (Vali et al., 2015). Once the first ice crystals are formed, triggered by INP, they can rapidly grow via the WBF process and multiply via secondary ice processes (Georgakaki et al., 2022; Korolev and Leisner, 2020), therefore, affecting cloud properties and albedo.

The simulation of ice crystal concentrations in climate models is therefore important for determining the properties of MPC, and uncertainties in the INP levels can lead to discrepancies in the modelled top-of-atmosphere radiative flux, and estimates of climate sensitivity to greenhouse gases (Vergara-Temprado et al., 2018; Murray et al., 2021). With climate change, and considering temperature changes alone, liquid water may progressively replace ice in MPC of a given altitude, making them more reflective. However, the

magnitude of this so-called cloud-phase feedback is highly uncertain (Frey and Kay, 2018), partly due to uncertainties in the present and future spatiotemporal distribution of INP (Zhao et al., 2021; Hoose et al., 2010; Raman et al., 2023) and the structure of layered cloud systems that would promote glaciation through the seeder-feeder mechanism, where ice crystals from an upper cloud (seeder) fall into a lower-lying liquid cloud (feeder) and grow there by riming or vapor deposition via the WBF process (Ramelli et al., 2021). The

inability of models to represent the occurrence frequency and glaciation state of MPC is also hypothesized as a major reason for disagreement between climate model results (Bodas-Salcedo et al., 2016) and reanalysis products (Naud et al., 2014) with regard to the annual mean downward solar (short-wave, SW) radiation.

     INP can originate from several sources, each with its own set of characteristics and properties. INP prediction in models typically relies on empirical parameterizations that are subject to considerable uncertainty and

challenges. Advancing the predictability of INP abundance with reasonable spatiotemporal resolution will require an increased focus on research that bridges the measurement and modeling communities. Additionally, coupling cloud processes to simulated aerosols also makes cloud physics simulations increasingly susceptible to uncertainties in the simulation of INP, due to the lack of sufficient observational constraints.

Many substances have been shown to nucleate ice, depending on the cloud type and the predominant freezing mechanisms. However, observational evidence suggests that only a small number of particle types (<10) must be represented to adequately predict INP in climate models (Burrows et al., 2022). In immersion freezing, which is the dominant mode of primary ice formation in MPC (Hande and Hoose, 2017), desert dust, marine primary organic aerosols (MPOA) and terrestrial primary biological aerosol particles (PBAP) have emerged

as the most relevant INP types (Spracklen and Heald, 2014; McCluskey et al., 2018; Chatziparaschos et al., 2023; Cornwell et al., 2023). These findings align with the physical understanding that INP are often efficiently activated as cloud condensation nuclei due to internal mixing with soluble components and their large size. These particles are likely enclosed within cloud droplets, making them available for immersion freezing but unavailable for contact freezing. The latter is constrained by the collision rate between interstitial

110 particles and supercooled liquid droplets, a process that remains poorly understood at a fundamental level in real clouds (Kanji et al., 2017). Consequently, the role of contact freezing and its contribution to primary ice formation in MPC is characterized by high uncertainty.

## 1.1 Dust INP ($INP_D$)

Globally, airborne mineral dust from deserts and other drylands is considered the dominant INP source at
115 temperatures below -20°C (Murray et al., 2012; Kanji et al., 2017). For instance, Pratt et al. (2009) identified that ~50% of the ice crystal residual particles sampled in clouds at high altitudes over Wyoming were mineral dust. Studies have subsequently demonstrated that dust mineralogy plays a pivotal role in influencing the ice-nucleating efficiencies of mineral dust (Atkinson et al., 2013; Boose et al., 2016, 2019; Cziczo et al., 2013; Hoose et al., 2008), with several studies finding potassium enriched (K-) feldspar to be the most INP active
120 dust mineral (Atkinson et al., 2013; Augustin-Bauditz et al., 2014; Zolles et al., 2015; Peckhaus et al., 2016; Harrison et al., 2019; Holden et al., 2019). Additionally, quartz is a major component of airborne mineral dust (Glaccum and Prospero, 1980; Perlwitz et al., 2015a, b) and exhibits a strong INP activity (Atkinson et al., 2013; Zolles et al., 2015; Losey et al., 2018; Kumar et al., 2019b; Holden et al., 2019). Quartz particles have substantially lower INP efficiencies than K-feldspar particles (Harrison et al., 2019), but their much
125 higher abundance in the atmosphere can lead to an important INP contribution (Chatziparaschos et al., 2023). There are also other minerals present in dust, such as smectite (Kumar et al., 2023) and kaolinite (Wex et al., 2014), but their activity is rather low compared to K-feldspar and quartz. Some laboratory studies have found that organic-rich soil dust from agricultural lands is more INP active than inorganic desert dust (Suski et al., 2018), although this fraction is typically not represented in climate models (Burrows et al. 2022).

130 ## 1.2 PBAP INP ($INP_{PBAP}$)

While it is well established that dust particles are the most abundant source of INP at temperatures below -20°C, it is equally well established that bioaerosols, i.e. atmospheric particles of biological origin, for instance terrestrial bioaerosol or PBAP, such as fungal spores, bacteria, pollen, and plant debris, are highly efficient ice nucleators at warm temperatures (Huang et al., 2021; Cornwell et al., 2022) and may have a
135 substantial effect on ice crystal formation (Prenni et al., 2009). Only a small fraction of biological material can trigger ice nucleation (Huang et al., 2021), with INP concentrations estimated at approximately 1 to 2 per liter (Pöschl et al., 2010). These values are, however, about 2 orders of magnitude lower than the highest observed INP number concentrations from dust (Murray et al., 2012). Although these INP levels are much lower than usually observed for warm MPC, ice crystal formation from PBAP can be subsequently enhanced
140 by ice multiplication processes ("secondary ice") (Korolev and Leisner, 2020), which are most efficient at temperatures warmer than -15°C (Georgakaki et al., 2022; Sullivan et al., 2018). Recent advances in remote sensing and model/sensor fusion have enabled detection of secondary ice processes in clouds, revealing that they hold particular significance in temperatures ranging from -5°C to -20°C (Wieder et al., 2022; Billault-

Roux et al., 2023). At the same time, the occurrence of secondary ice production (SIP) may actually mitigate uncertainties in INP predictions, given that ice crystal formation can be a self-limiting process (e.g., Sullivan et al., 2018).

Global simulations including biological particles have estimated that PBAP contribute to droplet freezing rates of clouds in the mid-latitude atmosphere, accounting for $1 \times 10^{-5}$ % of the global average ice nucleation rates, with an upper-most estimate of 0.6% (Hoose et al., 2010). Spracklen and Heald (2014) suggest that PBAP can dominate immersion freezing rates at altitudes corresponding to pressures between 400 and 600 hPa, in agreement with observational studies (Pratt et al., 2009). Field measurements have observed enhancements in both PBAP and INP related to rainfall events (Huffman et al., 2013; Tobo et al., 2013; Mason et al., 2015; Kluska et al., 2020) and suggested that PBAP emissions are affected by relative humidity (Prenni et al., 2013). Tobo et al. (2013) found that in a mid-latitude ponderosa pine forest, the variation pattern of INP concentration was like that of fluorescent biological aerosol particles during summertime, suggesting that PBAP are a critical contributor to INP concentrations. Consequently, PBAP acting as INP at warm freezing temperatures may be an important type of INP in regions where they are abundant and may have a noteworthy impact on climate. However, modelling of PBAP remains challenging due to the complexity of ecosystem processes and a lack of observations (Burrows et al., 2022).

**1.3 MPOA INP (INP_MPOA)**

Organic material with INP potential is also associated with sea spray aerosol (SS) emissions from the ocean. Historically, SS was considered unimportant as a source of atmospheric INP. However, recent observational and modelling studies have revealed that emitted SS mixed with MPOA is potentially an important or even the only type of INP of marine origin (Burrows et al., 2013; Wilson et al., 2015; De Mott et al., 2016). McCluskey et al. (2018a, 2018b, 2019) confirmed that SS was strongly associated with MPOA as the primary organic INP source. Wilson et al. (2015) showed experimentally that marine organic material associated with phytoplankton and cell exudates in the sea surface microlayer is an important contributor to atmospheric INP. In an exploratory model study, Burrows et al. (2013) showed that marine biogenic INP are likely to significantly contribute to the concentration of INP in the near-surface air over the Southern Ocean. INP derived from MPOA has potentially a remarkable effect on solar radiation, cloud glaciation and precipitation at high and mid-latitudes in remote marine regions especially during winter (Wilson et al., 2015). Huang et al. (2018) found that marine INP had only a small effect on cloud ice number concentration and effective radius, and did not significantly affect the global radiative balance. However, they emphasized that the relative importance of MPOA as an INP was found to be significantly dependent on the type of ice nucleation parameterization scheme that is chosen. Marine INP is not the main source of INP across all ocean areas. Gong et al. (2019) concluded that SS contributed less to the total INP in the atmosphere in Cape Verde than the other types of INP such as dust. In contrast, Vergara-Temprado et al. (2018) and McCluskey et al. (2019) proposed that MPOA is an important contributor to INP in remote marine environments. Marine INP concentrations are found to maximize over areas of high biological activity and wind speed in the Southern

Oceans, the North Atlantic, and the North Pacific, while mineral dust from deserts (the main terrestrial source of INP) accounts for only a small fraction of the INP observed in these regions (Wilson et al., 2015; Huang et al., 2018).

In this study, we enhance our comprehension of the global spatiotemporal distribution of INP by incorporating both marine and terrestrial biogenic INP types, namely MPOA and PBAP, the latter including

bacteria and fungal spores unless otherwise noted, building upon our earlier investigation of mineral dust-derived INP (Chatziparaschos et al., 2023). The methodology involves utilizing a 3-dimensional chemistry transport model for conducting simulations. Heterogeneous nucleation in the immersion mode is parameterized based on the singular hypothesis (Vali et al., 2015), using aerosol-specific parameters derived from laboratory and field observations. We identify major sources of INP and the specific aerosol types

capable of acting as INP, dependent on location and season. Figure 1 presents a comprehensive schematic illustration of INP derived from mineral dust, MPOA and PBAP that is used as the basis of our modelling approach. It includes emission processes, ice nucleation mechanisms, and the aerosol indirect effect, indicating the role of aerosols in cloud interactions and their impact on climate.

The simulations of INP-relevant species are evaluated with focus on the PBAP that is the new component in the model and its simulations have not been evaluated in earlier publications. Thus, the PBAP emissions, their seasonal variation, burdens and the calculated contributions are extensively evaluated against observations and other modelling studies. Furthermore, the simulated INP concentrations are rigorously evaluated against available INP observations sourced from the BACCHUS and Wex et al. (2019) databases.

For this purpose, INP concentrations are presented in this study in two ways as $[INP]_{ambient}$ and as $[INP]_{T:}$ $[INP]_{ambient}$ is calculated at ambient model temperature relevant to non-deep convective mixed-phase clouds using the ambient (model) temperature at each simulated level. $[INP]_T$ is calculated at a fixed temperature relevant to vertically extended clouds as deep-convective systems (Figure 1). $[INP]_{ambient}$ and $[INP]_T$ metrics can vary significantly. Finally, we assess the relative contribution of bioaerosols to ice nucleation via

immersion freezing in comparison to mineral dust and pinpoint regions and altitudes where this contribution holds significance.

## 2. Methods

### 2.1 Model description

The well-documented 3-dimensional global chemistry-transport model TM4-ECPL (Kanakidou et al., 2020;

Myriokefalitakis et al., 2015, 2016) is used here, driven by ERA-Interim reanalysis (Dee et al., 2011) meteorological fields from the European Centre for Medium-Range Weather Forecasts (ECMWF). The model is run with a horizontal resolution of 3° longitude by 2° latitude with 25 hybrid pressure vertical levels from the surface up to 0.1hPa (~ 65 km) and a model time step of 30 min. Therefore, we perform multi-year

simulations that provide climatologically meaningful results. The TM4-ECPL model considers lognormal aerosol distributions in fine and coarse modes and allows hygroscopic growth of particles, as well as removal by large-scale and convective precipitation and gravitational settling. Advection of the tracers in the model is parameterized using the slopes scheme (Russell and Lerner, 1981), and convective transport is parameterized based on Tiedtke (1989) and the Olivié et al. (2004) scheme. Vertical diffusion is parameterized as described in Louis (1979). For wet deposition, both large-scale and convective precipitation are considered. In-cloud and below-cloud scavenging are parameterized in TM4-ECPL, applied as described by Jeuken et al. (2001) and references therein. For all fine aerosol components, dry deposition is parameterized similarly to that of non-sea-salt sulfate, which follows Tsigaridis et al. (2006). Also, gravitational settling (Seinfeld and Pandis, 1998) is applied to all aerosol components.

## 2.2 Atmospheric cycle of INP-relevant aerosols

Desert dust emissions are calculated online as described in Tegen et al. (2002) and implemented as in Van Noije et al. (2014) accounting for particle size distribution and based on vegetation type and cover, dust source areas, snow cover, soil moisture, and surface wind speed. Following Chatziparaschos et al. (2023), we consider both K-feldspar and quartz as the INP active mineral species of dust, whose fractions in the soil-surface are taken from the global soil mineralogy atlas of Claquin et al. (1999), with updates provided in Nickovic et al. (2012). The calculation of the emitted mass fractions of K-feldspar and quartz in the accumulation and coarse modes [with dry mass median radii (lognormal standard deviation) of $0.34\mu m$ (1.59) and $1.75\mu m$ (2.00), respectively] is based on brittle fragmentation theory (Kok, 2011). Thus, the mass emission of each mineral is calculated by applying the respective mineral-emitted mass fractions to the dust emission fluxes. All minerals are emitted externally mixed, and dust particle density is equal to 2650 kg m$^{-3}$ (Ginoux et al., 2004; Tegen et al., 2002).

This study considers PBAP as terrestrial bacteria and fungal spores. In light of the well-documented discrepancies in INP activity spanning several orders of magnitude between various bioaerosol types (Murray et al., 2012; Kanji et al., 2017), it appears imprudent to assume that any individual species is representative of ambient PBAP. Therefore, we use parameterizations based on concurrent measurements of INP together with broad classification of bioaerosol (bacteria, fungal spore) types (Tobo et al., 2013). Terrestrial bacteria (BCT) emissions are parameterized as proposed in Burrows et al. (2009), where near-surface observations were used in combination with model simulations to determine the optimal BCT flux rates for particles with a diameter of 1 μm [monodisperse spherical particles], across six different ecosystems: coastal: 900; crops: 704; grassland: 648; land ice: 7.7; shrubs: 502; and wetlands: 196 m$^{-2}$ s$^{-1}$. For the present study, the Olson Global Ecosystem Database (Olson, 1992), originally available for 74 different land types on a spatial scale of 0.5∘×0.5∘, was grouped into 10 ecosystem groups as proposed by Burrows et al. (2009). Consequently, the total bacteria flux is calculated as a sum of bacteria fluxes per ecosystem, weighted by the corresponding area of each ecosystem within the model's grid box. In TM4-ECPL, upon emission, the insoluble fraction of

PBAP becomes progressively soluble due to atmospheric ageing. This process, which has been seen to occur, for instance, by degradation of RNA (Paytan et al., 2003) is parameterized based on the simulated oxidant levels, as for all other continental organic aerosols in the model (Tsigaridis and Kanakidou, 2003; Tsigaridis et al., 2006). Thus, the mean turnover of aged PBAP is estimated by applying a hydrophobic to hydrophilic conversion rate that depends on the atmospheric oxidants and is equivalent to a global mean turnover of 1.15

days (Tsigaridis and Kanakidou, 2003; Kanakidou et al., 2012). Regarding deposition, the hydrophilic aerosols are removed from the troposphere faster by dry and wet deposition than the hydrophobic ones.

Fungal spores (FNG) fluxes are treated as linearly dependent on the leaf area index (LAI) and specific humidity based on a parameterization proposed by Hummel et al. (2015). This parameterization is informed by field measurements of fluorescent biological aerosol particles conducted at diverse locations across

Europe; thus, it might not be very accurate for other regions and will be improved when data from other regions of the globe will become available. In the TM4-ECPL, the parameterization that is used to calculate FNG emissions online uses monthly averaged LAI distributions from the Global Land Cover 2000 database (European Commission, Joint Research Centre, 2003) as Hummel et al. (2015), which represents a climatology for the present study since no changes in vegetation from year-to year are taken into account,

and 3-hourly averaged specific humidity and temperature data, as provided by ERA-Interim. Bioaerosol sizes range from fine to coarse, but since their shapes are not accurately known, for the present work, FNG are assumed to be monodisperse spherical particles of 3 μm diameter with 1000 kg m$^{-3}$ density (Hummel et al., 2015). The organic matter to organic carbon ratio (OM:OC) of all PBAP is set equal to 2.6, and the molecular weight equal to 31 g mol$^{-1}$, using mannitol as a model compound (Heald and Spracklen, 2009;

Myriokefalitakis et al., 2016).

The role of pollen as INP has been demonstrated in previous studies (Hoose et al., 2010; Pummer et al., 2012); however, its experimental determination is challenging due to its large size and spatiotemporal variability. With a mean diameter ranging from 10 to 125 μm (Jacobson and Streets, 2009) models assume that pollen is emitted as an entirely soluble aerosol (Hoose et al., 2010), resulting in substantial wet and dry

deposition (Myriokefalitakis et al., 2017). Therefore, considering its size, pollen is anticipated to have much lower number concentrations compared to other PBAP at MPC altitudes, except possibly during periods of strong pollen emissions in the spring. However, pollen potentially impacts low clouds as effective cloud condensation nuclei (Subba et al., 2023). Consequently, uncertainties in pollen emission, size distribution, hygroscopicity, and ice activity that vary among pollen species (Pummer et al., 2015), lead us not to consider

pollen as INP in this study. Overall, a fundamental understanding of bioaerosol types and the mechanisms of their emission from the biosphere into the atmosphere is lacking, highlighting the need for process studies to build confidence in INP model extrapolations in a changing biosphere.

MPOA emissions are calculated online by the model considering the partitioning between insoluble marine organics and sea salt, as described in detail in Myriokefalitakis et al. (2010). The parameterization is based

on O'Dowd et al. (2008), modified by Vignati et al. (2010). MPOA is calculated as a fraction of the submicron sea-salt aerosol based on chlorophyll-a (Chl-a) present in the ocean surface layer. The organic mass fraction is calculated as a linear relation to Chl-a, valid for Chl-a concentrations below 1.43 µg m$^{-3}$ (for larger values, the percentage is kept constant at 0.76). MPOA is emitted in the fine mode together with SS with a magnitude that depends on Chl-a in the seawater and assuming that it is entirely insoluble as determined by O'Dowd et

al. (2008) and applying the same modeling approach as in Vignati et al. (2010). Additionally, based on Facchini et al. (2008), we adopted a coarse mode MPOA source (Gong, 2003; Myriokefalitakis et al., 2010). The sea-salt source is calculated online by TM4-ECPL driven by wind speed at every time-step, following Gong (2003) and fitted for accumulation and coarse modes as described in Vignati et al. (2010). In TM4-ECPL, Chl-a concentrations are satellite-derived monthly average MODIS observations at a resolution of

1°x1°. The density of water-insoluble organic mass was set to 1000 kg m$^{-3}$ (Vignati et al., 2010) and that of dry sea salt to 2165 kg m$^{-3}$ (O'Dowd et al., 2008). The ageing of insoluble MPOA is taken into account, applying a constant first order loss rate that corresponds to a global mean turnover time of about 1.15 days (Cooke et al., 1999; Tsigaridis and Kanakidou, 2007).

**2.3 Ice nucleation parameterizations**

**2.3.1. Mineral dust: K-feldspar and quartz**

The effect of dust minerals on the INP concentration is parameterized using the singular approximation based on the laboratory-derived active site density parameterizations for K-feldspar and quartz minerals provided in Harrison et al. (2019). The parameterization for K-feldspar is valid in the temperature range between -

3.5°C and -37.5°C, while for quartz, between -10.5°C and -37.5°C. To calculate aerosol surface area, we assume that each mineral dust particle is spherical and externally mixed (Atkinson et al., 2013; Vergara-Temprado et al., 2017). The density of ice nucleation active surface sites, derived from ice nucleation frozen fraction for a polydisperse aerosol sample, is assumed to follow a Poisson distribution:

$$INP_D = \sum_i \sum_j n_{i,j} \left\{ 1 - e^{[-S_{i,j} n_{s(i)}(T)]} \right\}$$    Eq. 1


where $INP_D$ is the number concentration of ice formed on the mineral dust aerosol, $n_{i,j}$ is the total dust mineral particle number concentration (per m$^{-3}$), where index i corresponds to mineral type (quartz and K-feldspar) and index j to size mode (accumulation and coarse mode). $S_{i,j}$ is the individual dust particle mean surface area (cm$^{-2}$) in size mode j. Air temperature, T, is given in degrees Kelvin and $n_{s(i)}$ corresponds to the active

site density of each mineral (Harrison et al., 2019). Further details on the methodology and the evaluation of the simulated INP derived from mineral dust are provided in Chatziparaschos et al. (2023).

### 2.3.2 PBAP

The number concentration of INP derived from PBAP is based on the parameterization of Tobo et al. (2013), which considers the dependence of INP on temperature and the number concentration of aerosol particles with diameters larger than 0.5µm. Particles with diameters above this threshold are assumed to have a sufficient surface area to provide active sites for ice nucleation (DeMott et al., 2010). Tobo et al. (2013) derived a parameterization of $INP_{PBAP}$ from observations of INP and ambient fluorescent biological aerosol particles (FBAP) in a mid-latitude ponderosa pine forest ecosystem. They, thus, proposed to calculate $INP_{PBAP}$ number concentration for temperatures ranging from about -9°C to -34°C and the number concentration of FBAP, hereafter replaced by that of PBAP, as follows:

$$INP_{PBAP} = \left(N_{PBAP,>0.5\mu m}\right)^{a'(273.16-T)+\beta'} e^{\gamma'(273.16-T)+\delta'} \qquad \text{Eq. 2}$$

where $\alpha'$ =-0.108, $\beta'$ = 3.8, $\gamma'$ = 0, and $\delta'$ = 4.605 and $N_{PBAP}$ is the number concentration (m$^{-3}$) of PBAP (FNG and BCT) larger than 0.5µm, and T is the air temperature in Kelvin. There are two limitations in using this parameterization. Firstly, this parameterization has been derived from a mid-latitude ponderosa pine forest ecosystem and therefore its application on a global scale remains a rough approximation. Secondly, FBAP may also contain non-PBAP (e.g., mineral dust particles, secondary organic aerosol particles) that contribute to fluorescence (Bones et al., 2010; Lee et al., 2013; Morrison et al., 2020). The number fractions of FBAP to the total aerosol particles in the super-micron size range could be relatively similar to those of PBAP under dry conditions (Tobo et al., 2013). Indeed, observations by Negron et al. (2020) suggest that airborne bacteria may be unambiguously detected with autofluorescence. However, during and after rain events, PBAP were underestimated by more than a factor of 2 (Huffman et al., 2013), suggesting that FBAP reflect only a portion of PBAP. Overall, there is ample evidence supporting a correlation between INP and total fluorescent particles (Huffman et al., 2013). However, the exact identity of these fluorescing particles has not yet been clarified adequately, and it remains challenging to determine better proxies for biological origin INP that can be readily measured in the environment.

### 2.3.3 MPOA

The contribution of marine organic material to INP is parameterized according to Wilson et al. (2015), based on the total organic carbon (TOC) in the sea spray originating from the sea surface microlayer and the ice-nucleating temperature. It considers the active sites density per unit mass of TOC contained in insoluble MPOA and is derived from a spectrum of temperatures ranging from -6°C to -27°C. A factor of 1.9  is used for the conversion between MPOA and TOC in the model, as suggested in Burrows et al. (2013). The number

concentration of INP<sub>MPOA</sub> (m$^{-3}$) per TOC (g) originating from MPOA is calculated following the equation below.

$$INP_{MPOA} = C_{TOC}e^{[11.2186-(0.4459*T)]}$$ 
Eq. 3

where $C_{TOC}$ is the total organic carbon mass concentration (g/m$^3$) of marine particles and T is the air temperature in °C.

## 2.4 Simulations and evaluation

Simulations were performed from 2009 to 2016 using the year specific meteorological fields and emissions as described in the previous sections. Prior to the evaluation of INP, we provide an evaluation of the INP-
relevant aerosols. The simulated mineral dust mass concentrations and deposition fluxes were evaluated against global observations in Chatziparaschos et al. (2023), and a summary of the evaluation is presented below. We also assess the MPOA simulations using observations from Mace Head and Amsterdam Island, incorporating findings from multiple studies (Sciare et al., 2009; Rinaldi et al., 2010). Evaluations of PBAP, as well as individual BCT and FNG concentrations, are based on a compilation of long-term measurements
of atmospheric fluorescent bioaerosols and number concentrations from various locations, as listed in Elbert et al. (2007), Bauer et al. (2008), Burrows et al. (2009) and Genitsaris et al. (2017).

The simulated INP are compared against the available INP observations from the databases of BACCHUS (Impact of Biogenic versus Anthropogenic emissions on Clouds and Climate: towards a Holistic Under
Standing) (http://www.bacchus-env.eu/in/index.php, last accessed on 26 March 2020) and Wex et al. (2019) (https://doi.pangaea.de/10.1594/PANGAEA.899701, last access on 14 February 2022). The observational data span from 2009 to 2016 and originate from different campaigns (locations are shown in supplementary Fig. S1). As outlined in the introduction [INP]<sub>T</sub> and [INP]<sub>ambient</sub> are determined by the simulated particle concentration and for [INP]<sub>T</sub> a specific temperature (T), while for [INP]<sub>ambient</sub> the model's ambient temperature
is used. [INP]<sub>T</sub> is the appropriate metric when comparing modelled INP concentrations to observations, as measurements are typically conducted by exposing particles to specific, controlled temperatures within the instruments. Thus, for the model evaluation, [INP]<sub>T</sub> concentrations are used that are calculated at the temperature at which the measurements were performed. All model results are compared to observations for the specific month and year as well as the location of observation, except for those reported by Yin et al.
(2012) and Bigg campaigns (Bigg, 1990, 1973), which cover temporally scattered measurements (between 1963 and 2003) and are therefore compared to the modelled multi-annual monthly mean INP concentration from 2009 to 2016. Three statistical metrics are used to assess the ability of the model to correctly predict INP concentrations: i) the percentage of INP simulated values within one order of magnitude from the observed values (Pt$_1$), ii) the percentage within one and half orders of magnitude (Pt$_{1.5}$) and iii) the modified

normalized mean bias (mnMB). The correlation between observed and predicted INP (as log/log regression) is presented by the correlation coefficient (R) of the regression of the logarithms of the concentrations.

## 3.    Results and Discussion

### 3.1    INP-relevant aerosols
#### 3.1.1    Global aerosol simulations

Figure 2 displays the multi-annual mean surface concentrations of dust (K-feldspar and quartz), marine primary organic aerosol (MPOA), and primary biological aerosol particles (PBAP), with PBAP components shown separately as bacteria (BCT) and fungi (FNG), along with their zonal mean distributions as calculated by the model, averaged over the study period. A portion of these aerosols act as INP, as discussed in Section
2.3. Dust aerosols peak over desert regions and exhibit notable concentrations in the outflow regions of the Sahara and Gobi deserts, predominantly in the Northern Hemisphere (NH), as well as downwind of Patagonia over the South Atlantic. MPOA, driven by surface wind speeds and seawater chlorophyll levels, reaches its highest concentrations in the mid-high latitudes of both the NH and Southern Hemisphere (SH). BCT and
FNG concentrations peak over continental regions, particularly across tropical zones.

#### 3.1.2    Evaluation of simulated INP-relevant aerosols

**3.1.2.1 Dust**

The modelled mineral dust mass concentration and deposition fluxes were thoroughly evaluated against global observations in a previous study (Chatziparaschos et al., 2023). A monthly-collocated evaluation of
the modelled dust surface concentration was performed against in-situ measurements representative of Saharan dust sources and transport regions from four stations (M'Bour, Bambey, Cinzana, and Banizoumbou) located at the edge of the Sahel (Lebel et al., 2010), three stations (Miami, Barbados, and Cayenne) located on the American continent, downwind of the Atlantic dust transport ( Prospero et al., 2020), and two stations, Finokalia (Crete, Greece) (Kalivitis et al., 2007) and Agia Marina (Cyprus) (Pikridas et al.,
2018) situated along the dust transport over Mediterranean. This comparison suggested a slight positive model bias of 8.3 $\mu g\ m^{-3}$ on average that represents 5 % of the total average dust concentration observed at these sites. The ability of TM4-ECPL to represent the global dust cycle was further evaluated against the modelled surface dust concentration and deposition with climatological, globally distributed observations [Climatological annual means of dust surface from the Rosenstiel School of Marine and Atmospheric Science
(RSMAS) of the University of Miami (Arimoto et al., 1995; Prospero, 1999) and the African Aerosol Multidisciplinary Analysis (AMMA) (Marticorena et al., 2010)]. A tendency to overestimate dust surface concentrations was also noted in comparisons with these annual mean climatological observations, with an overall normalized mean bias of 44.1%. For deposition model errors were higher than for surface

concentration fields, with a normalized mean bias of 59.2 %. Despite these discrepancies, which must be considered in evaluating INP simulations, the model reliably captures the geographic distribution of atmospheric dust within an acceptable order of magnitude.

### 3.1.2.2 MPOA

The evaluation of organic carbon contained in MPOA was previously conducted by Myriokefalitakis et al. (2010), estimating an annual global source of MPOA about 4 TgC·yr$^{-1}$. TM4-EPCL captures the general geographic variability, although it tends to underestimate organic carbon concentrations, suggesting a possible underestimation of emissions or missing sources. The uncertainties are particularly tied to parameterizations of sea spray emissions, which are highly variable spatially and temporally and are estimated to affect the MPOA concentrations by a factor of 4, as well as to the fractional contribution of organic matter in sea spray particles that has on average an uncertainty of 33% (Vignati et al., 2009). We further evaluate the MPOA concentrations and MPOA fraction in SS by comparison with observations from Mace Head (North Atlantic; Rinaldi et al. (2010)) and Amsterdam Island (Austral Ocean; Sciare et al. (2009)) as depicted in Figure 3.

Figure 3a compares simulated MPOA concentrations (averaged over the simulation period) with marine sector observations at Mace Head in 2006 (Rinaldi et al., 2010; total water-insoluble and water-soluble organic matter). The model generally captures the seasonal pattern, with elevated values in spring and summer and lower values in winter, but tends to slightly underestimate the annual mean concentration by about 3%. The underestimation is most pronounced in summer, while spring and fall values appear to be overestimated. Observed organic carbon (OC) levels at Amsterdam Island in the Southern Indian Ocean, measured from 2003 to 2007 by Sciare et al. (2009), are primarily attributed to MPOA (Myriokefalitakis et al., 2010) and are compared with model results in Figure 3b. Overall, the model underestimates the observations by 50% with the largest discrepancy in the Southern Hemisphere summer. When comparing observations from both the Northern and Southern Hemispheres to model results (Figure S2) an overall underprediction of 29% (nMB) is evident. These discrepancies may reflect deficiencies in the applied MPOA source and sink parameterizations. Sea salt emissions and chlorophyll-a concentrations, are key drivers of the spatial distribution of MPOA, and play a significant role in this discrepancy. Furthermore, the coarse resolution of the model may not capture sharp latitudinal gradients in localbiological activity. A possible underestimation of the marine source associated with the specific distribution of phytoplankton species, in addition to the use of monthly averages for Chl-a from MODIS, which can smooth out short-term variations, can significantly affect MPOA emissions. Therefore, a factor of two uncertainty in the simulated MPOA is plausible and the observed underprediction (3%-50%) should be considered when evaluating the INP simulations.

### 3.1.2.3 PBAP

The simulated BCT, FNG, and total PBAP number concentrations are compared with observations at various locations in Figures 4a-c. Figure 4a presents the comparison of simulated BCT with observations from

Burrows et al. (2009) and Genitsaris et al. (2017). Actively wet discharged ascospores (AAS) and basidiospores (ABS) data are compared with simulated FNG (Figure 4b). AAS and ABS are assumed to contribute one-third each to total fungal spore concentration (Burrows et al., 2009). The observations are listed in Elbert et al. (2007) [AAS-ABS] and Bauer et al. (2008) [fungal spore concentrations].

Figure 4a shows that the model underestimates BCT concentrations across most regions and seasons by 75% (nMB) on average. This bias may be partly due to the monodisperse approach of bacterial emissions or their assumed diameter (1μm). The comparison of ABS and AAS with simulated FNG (Figure 4b) following the approach of Hoose et al. (2010) shows both significant overestimation and underestimation by the model, but the FNG concentrations listed in Bauer et al. (2008) agree well with the simulated concentrations, which assume a monodisperse aerosol of 3 μm diameter. Overall, simulations of fungal spores tend to overestimate measurements with a nMB of 7.2%. Taking into account that the model results are monthly averages for the 2°×3° model grids and thus their comparability with point measurements is limited, as well as the uncertainties in the observations and their spatial and temporal representativeness, the agreement between measurements and observations is satisfactory overall.

Although the simulated BCT concentrations are underestimated, the total PBAP (BCT, FNG and pollen) number concentrations are in the correct order of magnitude when compared to fluorescent PBAP number observations averaged over the reported period (Figure 4c); the model tends to overestimate total PBAP by 17%. Furthermore, the model captures reasonably well the seasonal variations that follow the observed patterns (Supplementary Figure S3), although it tends to overestimate the total PBAP number concentration, in summer and autumn, when biological activity is high. This may indicate that either fungal spores or pollen may be less important contributors to the total PBAP number at the monitoring sites than simulated. The discrepancy between model simulations and observations highlights the complexity of biological aerosol emissions and the influence of local environmental factors on bioaerosol concentrations.

In summary, the evaluation of INP-relevant aerosols simulations shows that the model underpredicts MPOA (by about 30%; 3% - 50%) and overpredicts dust surface concentrations and deposition (by about 60%). PBAP concentrations are also slightly overpredicted (by about 20%), with BCT being significantly underpredicted and FNG moderately overpredicted. The model captures the seasonal variation of observed PBAP, while the simulated MPOA in the SH shows no significant variation across seasons. These model features must be taken into consideration when discussing the results of the INP simulations, since uncertainties in the INP simulations include uncertainties in both the INP parametrization and INP-relevant aerosol simulated concentrations and affect the inferred importance of various INP types in the model.

## 3.2    Global INP simulations

### 3.2.1    INP global distributions

Figure 5 (a-c) depicts the simulated spatial distributions of INP concentrations derived from each studied aerosol type present at 600 hPa and with the ability to freeze at -20°C in immersion mode (hereafter called $INP_D$ at -20°C and noted as $INP_D$[600hPa, -20°C], $INP_{MPOA}$ at -20°C and noted as $INP_{MPOA}$[600hPa, -20°C] and $INP_{PBAP}$ at -20°C and noted as $INP_{PBAP}$[600hPa, -20°C], that is INP from dust, MPOA and PBAP (BCT and FNG), respectively). This metric is explained in Sect. 2.4 and Figure 1 $[INP]_T$. The chosen pressure level is representative of the low free troposphere and average temperatures broadly consistent with those of the INP measurements. These conditions of temperature and pressure are representative of low latitude MPC's glaciation, where most of the INP is simulated to occur (Fig. 5). Figure 5d shows the total $[INP]_T$ distribution calculated accounting for all the INP types.

$INP_D$[600hPa, -20°C] (Fig. 5a) are higher in the mid-latitudes of the NH than in the SH due to the location of dust sources and long-range atmospheric transport patterns that favor the presence of atmospheric dust in the NH. Throughout much of the low and mid-latitudes of the NH, $INP_D$[600hPa, -20°C] outnumbers by far $INP_{MPOA}$[600hPa, -20°C] and $INP_{PBAP}$[600hPa, -20°C] (Fig. 5a-c). $INP_{MPOA}$ at -20°C dominates in the oceanic regions, particularly in the SH, and $INP_{PBAP}$[600hPa, -20°C] prevails over tropical and equatorial continental sites and up to +/- 40° latitude outside of strong dust source regions.

Remarkably, there are oceanic regions in the South Atlantic Ocean and Pacific Ocean (such as the south hemisphere tropical west coasts of South Africa and South America) where PBAP has the potential to form ice crystals at the outflow of the continental air (Fig. 5d, circles 1 and 5), enhancing INP concentrations in marine atmosphere. These findings indicate that PBAP derived from terrestrial vegetation ecosystems can play a significant role in driving atmospheric INP concentrations both within and downwind of source areas. In addition to the presence and thus potential influence of PBAP in marine environments, $INP_D$[600hPa, -20°C] shows considerable levels in many oceanic regions. This is particularly evident over the central Atlantic, downstream of the Sahara, where $INP_D$[600hPa, -20°C] surpass those of both $INP_{MPOA}$[600hPa, -20°C] and $INP_{PBAP}$[600hPa, -20°C] (Fig. 5d, circle 2). $INP_D$[600hPa, -20°C] shows also significant levels over the North Pacific, where dust is carried by continental outflows within the boundary layer or/and the free troposphere. Mineral dust and MPOA control the INP population over the South Atlantic adjacent to northeastern coast of South America due to both emissions from the Patagonia desert and marine biota (Fig. 5d, circle 3). Over the southern Indian Ocean $INP_{MPOA}$[600hPa, -20°C] dominates (Fig. 5d, circle 4). The simulated increase of INP by MPOA towards the SH is consistent with findings by Huang et al. (2018). Note that the spatial distribution of INP at a fixed pressure level and temperature shown in Fig. 5 and discussed above needs to be complemented by the vertical distribution of INP, which represents the combined effect of sources, long-range transport, vertical mixing and boundary layer height changes, that differ between low and high latitudes.  In this context, figure 6 depicts multi-annual zonal mean vertical profiles of ambient INP

number concentrations that are derived from mineral dust, PBAP and MPOA calculated at modelled ambient temperature ($INP_D$, $INP_{PBAP}$, and $INP_{MPOA}$, respectively; [$INP_{ambient}$] shown in Figure 1) (Fig. 6a-c) along with the total ambient INP concentration (Fig. 6d).

$INP_D$ (Fig. 6a) presents considerable concentrations ($>5 \times 10^{-1}$ $L^{-1}$) at temperatures below -20°C in agreement with findings by Hoose and Möhler (2012). Previous studies suggested that dust is the dominant source of immersion mode INP at temperatures colder than −25°C (Murray et al., 2012; Kanji et al., 2017). However, over the dust belt, dust has been found to initiate freezing in midlevel supercooled stratiform clouds at temperatures of −10°C and below (Liu et al., 2008; Zhang et al., 2012). These temperatures are warmer than those reported by Ansmann et al. (2008), who found no evidence of ice formation in supercooled stratiform clouds with cloud-top temperatures above −18°C, but colder temperatures than those found by Sassen et al. (2003), who attributed the glaciation of altocumulus clouds at −5°C to African dust. Although mineral dust can act as INP at warm temperatures, this depends on, among other factors, type and fraction of different minerals [K-feldspar and quartz fraction (Chatziparaschos et al., 2023; Harrison et al., 2019)], particle size (Chen et al., 2021), particle concentration per droplet, and biological nanoscale fragments attached to dust particles (Augustin-Bauditz et al., 2016). In our model, the potential mixing of dust with biological material is not considered and INP types are assumed to be externally mixed aerosols.

The simulated $INP_{PBAP}$ shows high number concentrations with concentrations $>10$ $L^{-1}$ at low altitudes between 40°S and 90°N at temperatures warmer than -16°C (Figure 6c). The temperature range where simulated $INP_{PBAP}$ exhibit considerable concentrations is close to the temperature range where secondary ice formation is expected to be active, i.e., between -5°C and -20°C (Korolev and Leisner, 2020), with peak impact around -15 °C (Georgakaki et al., 2022). Therefore, although not contributing significant primary ice particles, $INP_{PBAP}$ may help sustain secondary ice formation, by contributing primary ice at relatively warm freezing temperatures for SIP. That said, SIP is most effective for situations where there are seeder-feeder cloud configurations (i.e., layered clouds or vertically developed clouds that have internal seeding; Georgakaki and Nenes, 2023) so it remains to be investigated whether the "niche" impact of INP from PBAP on SIP is climatically important.

The simulated $INP_{MPOA}$ number concentrations are low ($10^{-1}$ to 1 $L^{-1}$ at 800hPa, $<$-20°C; Figure 6b) and consistent with the simulated concentration range shown in Wilson et al. (2015) and Vergara-Temprado et al. (2018). Despite the relatively large sources of MPOA in the SH, a region mostly covered by oceans, the resulting $INP_{total}$ concentrations remain lower than those simulated over the NH (Figure 6d), which is mostly continental and where dust and PBAP are mainly present. This pattern of low INP concentration in the SH agrees with observations in the Southern Ocean (McCluskey et al., 2018; Welti et al., 2020). The relatively low $INP_{MPOA}$ number concentrations in the SH (Figures 5b, 6b) could be translated into less cloud droplet freezing, enhancing cloud reflectivity (Vergara-Temprado et al., 2018). Low INP concentrations could lead to the initial growth of large ice (i.e., in the seed region) and can influence the evolution of the microphysical

cloud structure in the lower cloud levels (i.e., in the feeder region) enhancing precipitation (Borys et al., 2003; Ramelli et al., 2021). Vergara-Temprado et al. (2018) suggest that $INP_{MPOA}$ are negatively correlated with shortwave (SW) radiation up to an INP concentration of 1 $L^{-1}$, above which the reflected radiation drops sharply as the ice processes become more efficient and deplete most of the liquid water.

### 3.2.2 Contribution of INP types to the total INP

Figure 7 shows the multi-year average zonal mean of the percentage contribution of each INP type to the total INP as simulated by the model. $INP_{PBAP}$ is the primary type of INP between −12°C and −20°C. This agrees with a recent observational and modelling study by Cornwell et al. (2023). Overall, mineral dust (Fig. 7a) dominates INP levels at high altitudes in most of the globe in all seasons (Fig. S4). At intermediate pressure levels (up to 600hPa and T <-20 ºC), $INP_D$ dominates over the North Hemisphere (NH), where the

major sources of dust minerals are located (Sahara, Gobi deserts and Arabian Peninsula), as also reported by Chatziparaschos et al. (2023). Between 50°S and 90°N, the simulated concentrations of INP are driven mostly by mineral dust and PBAP. MPOA plays a minor role there, contributing less than 20% of the total INP between 800 and 600 hPa (60°N and 90°N).

PBAP is more ice-active than mineral dust at relatively high temperatures (Murray et al., 2012; Tobo et al.,

2013; Harrison et al., 2019). Therefore, $INP_{PBAP}$ contribution to the total INP is more than 80% between -12ºC and -16ºC. $INP_{PBAP}$ mainly affects the tropical and subtropical atmosphere during the whole year at temperatures higher than about -16ºC and extends to the pole in the NH. Note that MPC were found to occur more frequently in the convective towers in the tropics than at the mid-latitudes and the Arctic (Costa et al., 2017). In the boreal summer, the high $INP_{PBAP}$ contributions to total INP are shifted to northern latitudes due

to vegetation growth (Fig. S4). These results suggest that PBAP may be the dominant type of INP at relatively warm temperatures, consistent with previous studies (Tobo et al., 2013; Murray et al., 2012; DeMott and Prenni, 2010). At temperatures colder than -20°C, nonbiological aerosol particles such as mineral dust are effective INP (Murray et al., 2012; Si et al., 2019) and dominate the INP population (Fig. 7a). In the NH, INP concentrations mainly originating from PBAP and mineral dust are higher than those in the SH where

$INP_{MPOA}$ dominates between 60°-90°S. There is a pronounced seasonality in dust INP (Figure S5), with large concentrations observed between 40°N and 90°N during the boreal winter and spring (DJF, MAM) when transported dust can influence INP concentration over the Arctic. In contrast, $INP_{MPOA}$ shows minimal seasonal variation and consistently dominates the SH between 40°S and 80°S (Fig. S4c, S5c). Between 60°S and 60°N, $INP_{PBAP}$ is the most prevalent INP type at higher pressure and temperature levels, especially in the

NH during the boreal summer (JJA) and autumn (SON), when its contribution to total INP increases (Fig. S4b).

Thus, the seasonal variation of INP composition highlights the influence of temperature, atmospheric circulation, and biological activity on INP. At low altitudes and warm temperatures, INP populations are

influenced by the prevalence of INP of biological origin ($INP_{MPOA}$ and $INP_{PBAP}$), with MPOA affecting INP concentrations mainly in the SH and PBAP in the NH. $INP_D$ is the primary contributor to the INP population at high altitudes, where temperatures are low, displaying a consistent pattern across all seasons.

Figure 8 provides a comprehensive view of the global distribution of INP in the tropospheric column from the surface up to 300 hPa (Fig. 8a), calculated at the ambient modelled temperature, as well as the percentage contribution of each INP type per season and for three latitudinal zones (Fig. 8c). High INP column burdens are simulated in the NH and especially over the Sahara, parts of the Middle East, and the northern part of South America. These regions are known to be significant dust emission and biological aerosol sources (Amazon). Lower INP column burdens are simulated over the polar and oceanic regions, indicating that the contributions from INP sources in these areas are limited. The simulated INP column burdens are greater in the NH than in the SH, which is in agreement with previous studies (Vergara-Temprado et al., 2017). Figure 8b shows the associated interannual variability of the simulated INP multiyear mean columns indicated by the coefficient of variation that is calculated as the standard deviation of the annual mean columns to the multiyear annual mean and is expressed in percent. We find that locally the interannual variability can exceed 150%, particularly in the tropics and the polar regions and has to be kept in mind when interpreting modeling results. Figure 8c shows that mineral dust is the dominant source of INP in the high northern latitudes (60°-90°N) across all seasons, with contributions exceeding 89% in every season. $INP_{PBAP}$ and $INP_{MPOA}$ play minimal roles, with $INP_{PBAP}$ contribution to the column burden of the total INP peaking at about 8.5% in the fall (SON). In mid-latitudes and the tropics (60°S-60°N), INP contributions are more seasonally variable, highlighting the interplay between dust and biological particles in this region. Mineral dust dominates during MAM (spring NH) and SON (autumn NH), while $INP_{PBAP}$ plays a prominent role in DJF (winter NH), contributing about 65%. In the high southern latitudes (60-90°S), mineral dust remains the main source of INP, especially in DJF (summer SH), where it accounts for about 82.5%. However, in contrast to the other latitude zones, $INP_{MPOA}$ shows significant contributions to the total INP column burden in MAM (about 37%) and JJA (about 31.5%) in this latitude zone, while $INP_{PBAP}$ contribution is negligible.

To further illustrate the different components of ambient INP depending on atmospheric temperature, Figure 9 presents the percent contributions of INP types calculated at modelled ambient temperature and classified in three temperature ranges: [-40°C, -30°C], [-30°C, -20°C], and [-20°C, -10°C]. The figure highlights the spatial variability as a function of temperature of the contribution of the studied species to the total simulated INP at the modelled ambient temperature. In order to plot this figure, all model results are selected for which the modelled temperature falls within the respective temperature range. The percent contributions of each INP type to the total INP in each of the model grids and instances are calculated and then averaged per model grid, for all altitudes. According to our simulations, mineral dust contributes significantly at all temperatures, especially below -20°C, where its influence is widespread, and maximizes over desert areas such as the Sahara. At the coldest temperature range (-40°C to -30°C), dust dominates globally, extending its influence

over oceanic regions even in the SH. At this temperature range MPOA contribution is apparent mainly in the Southern Ocean high latitudes but also in the tropical Pacific, while PBAP contribution to INP is negligible.

In the middle temperature range (-30°C to -20°C) dust remains the dominant INP type, while PBAP contribution to INP becomes apparent over land. MPOA remains relevant in the tropical Pacific and the Southern Ocean. At higher temperatures (-20°C to -10°C), PBAP becomes the dominant INP type in tropical and subtropical regions, while MPOA dominates over the Southern Ocean (60°S–90°S), where it accounts for nearly 90% of the total INP, highlighting the strong influence of marine aerosols in polar ocean regions. In this temperature range, dust INP contributes mainly to total INP in high latitudes of the NH (e.g., the Arctic).

The recent observational study by Creamean et al. (2022) in the central Arctic revealed a strong seasonality of INP during the year, with lower concentrations in winter and spring controlled by transport from lower latitudes, and enhanced concentrations of INP in summer, likely of marine biological origin. This only partially agrees with our results, which show that INP concentrations in the Arctic region are mainly influenced by transported airborne dust in winter and spring, by $INP_{PBAP}$ in summer and fall and that $INP_{PBAP}$ contributes more than $INP_{MPOA}$ in summer (Fig. S4). This discrepancy may be due to the underprediction of MPOA and overprediction of PBAP and dust concentrations discussed in Sect. 3.1.2.

### 3.2.3   Comparison of INP predictions with observations

Cloud-resolved high-resolution modelling studies indicate that clouds can present sensitivity to INP perturbations that exceed one order of magnitude (Fan et al., 2017). For the simulation of INP in models to be deemed satisfactory, prediction errors must be constrained to less than an order of magnitude for the majority of observations (Cornwell et al., 2023). Figure 10 shows the comparison of INP observations with simulated INP, for each INP type separately and for all possible combinations. All observations were compared with spatiotemporally co-located simulated concentrations, except observations out of the simulation window, which were compared with climatological monthly mean geographically co-located model results. The $Pt_1$, $Pt_{1.5}$, the correlation coefficient (R) and the modified normalized mean bias (mnMB) using the logarithm of the concentrations defined in Sect.2.4 are used to assess the ability of the model to correctly predict INP concentrations for each of the depicted cases and are reported in this figure. Note also that the climatological data by Bigg (1973, 1990), although very useful due to their geographic coverage, they are by about 2-3 orders of magnitude higher than recent observations made closer to the period simulated in our study (i.e. McCluskey et al. (2018), Tatzelt et al. (2022), Moore et al. (2024)). This difference has to be kept in mind when comparing model results with past observations.

Considering dust minerals as the sole INP types leads to underestimation against observations (Fig. 10a; $Pt_1$ about 25%, R=0.84 and mnMB about 112%). Even if dust is the most abundant aerosol in the atmosphere,

the simulated dust-derived INP cannot predict the observed INP, especially at high temperatures (>-15°C) and relatively low INP levels, since mineral dust particles likely become ice-active only at low temperatures (Chatziparaschos et al., 2023; Cornwell et al., 2023). Notably, the results show that there is some overestimation for colder temperatures, which may be partly related to the overestimation of dust in the model. Findings in agreement with the literature (e.g. Vergara-Temprado et al., 2018) and our earlier study by Chatziparaschos et al. (2023).

$INP_{MPOA}$ alone significantly improves the prediction of the observed INP at high temperatures, increasing the predictability of the model from about 25% for dust alone to about 65% for MPOA alone with almost the same mnMB (about 111%) ($Pt_1$, Fig. 10b). However, this improvement is accompanied by a decrease in the correlation coefficient (R). At low (<-30°C) and mid (-20°C) temperatures, $INP_{MPOA}$ underestimates the observations (Fig. 10b). The discrepancy between observed INP and $INP_{MPOA}$ can be partially attributed to missing INP sources, especially dust, and partially to the uncertainty in the $INP_{MPOA}$ simulations resulting from both the MPOA simulations and the $INP_{MPOA}$ parameterization.

Figure10c shows the comparison between simulated $INP_{PBAP}$ and observed INP. $INP_{PBAP}$ overestimate the observed INP across temperature ranges warmer than -30°C, followed by a subsequent underestimation below this temperature. The model's predictability increases ($Pt_1 = 61\%$) compared to $INP_D$ ($Pt_1 = 25\%$) but decreases compared to $INP_{MPOA}$ ($Pt_1 \sim 65\%$), while it improves the mnMB (about 188%). This overestimation could be attributed to uncertainties in parameterizations for PBAP emissions and for the ice nuclei activity of PBAP. Since only a 20% overestimation of PBAP concentrations by the model can be deduced when compared to observations, an overestimation of the INP scheme seems to be the most plausible reason for the large $INP_{PBAP}$ overestimation.

The deviation of simulated individual INP types from total INP observations, shown in Figure S6 as a function of binned temperatures, indicates that at temperatures of about -25 to -30°C, both $INP_D$ and $INP_{PBAP}$ separately overestimate the observed total INP. For warmer temperatures, Figure S6 shows that a major contribution to the model overestimation of the observed total INP is due to $INP_{PBAP}$. However, as seen in Figure S7, $INP_{PBAP}$ alone compares well with total INP observations in the NH and the extratropical SH (Figure S7 top and mid-rows), indicating an overactive $INP_{PBAP}$ parameterization. Overall, among the three different types of INP when considered individually, the highest correlation (r =0.84, Fig. 10a) between simulated and observed INP concentrations is found for $INP_D$, which however fails to reproduce the low levels of INP at relatively warm temperatures.

When considering all INP types, the model's agreement with the INP observations is moderate, with a correlation coefficient of 0.85, about 59% ($Pt_1$) predictability and mnMB of about 214% (Fig. 10f), which is the smallest among all tested cases (Fig. 10). Considering only INP from MPOA in addition to dust improves the predictive ability of the model (about 71%, $Pt_1$) and has the highest correlation coefficient of 0.86 and a

moderate NMB (Fig. 10d). The inclusion of MPOA improves the model's comparison between -6°C and -25°C where MPOA exhibits high ice activity. Including $INP_{PBAP}$ in the simulations in addition to $INP_D$ and $INP_{MPOA}$ leads to an overprediction of INP measurements and reduces the accuracy of the model's predictions (Fig. 10f).

In summary, the comparison of model results with observations reveals that dust particles are a minor INP

source at warm temperatures but increase in importance at colder temperatures. INP at temperatures in the vicinity of zero (warmer than -20°C) are typically related to INP from biogenic sources, as simulated by PBAP and MPOA. Without improved representations of the sources and ice-nucleating activities of biological INP, models will struggle to simulate total INP concentrations at warmer temperatures and the resulting MPC. These findings are consistent with previous observational studies showing that biological INP

make up a greater fraction of continental INP at warm temperatures (Mason et al., 2015; Gong et al., 2022). The model achieves its highest predictability and correlation against observed INP across all temperature ranges when both $INP_D$ and $INP_{MPOA}$ are included, but the overprediction of dust aerosol suggests the need to additionally account for $INP_{PBAP}$ in climate modeling.

**3.3 Sources of uncertainties and implications for our results.**

Our results suggest that for climate prediction, with the current knowledge of ice nuclei properties of dust, PBAP and MPOA and as a first approximation, consideration of mineral dust and MPOA as the main contributors to INP globally could be sufficient, although PBAP regionally and at warm temperatures is an important contributor to INP. However, there are large uncertainties associated with these results related

both with the number concentrations of the various INP types and with the ice nucleating activity of the INP relevant aerosols. Parameterizations and model evaluation are based on observations geographically and temporally limited; therefore, their global applicability remains an open issue. In Section 3.1.2, we evaluated the uncertainties in the simulated dust, MPOA, and PBAP concentrations by comparisons with available observations and found significant overprediction of dust (~ 60%) and PBAP (~ 20%) and underprediction

of MPOA (3% to 50%).

The uncertainties in the simulation of INP number concentrations combine the biases in the INP-relevant aerosol simulations with those of the parameterizations of the INP activity of different types of aerosols. Laboratory studies proved that the ice nuclei activity of bacterial and fungal spores can vary substantially across different species (Pummer et al., 2012; Murray et al., 2012). The species composition of airborne

fungal and bacterial communities exhibits considerable geographic variability (Dietzel et al., 2019). Thus, it is reasonable to expect that the ice nucleating efficiency of airborne PBAP may strongly differ geographically. In addition, the INP parameterizations used here were developed based on a small number

of samples from the mid-latitudinal ponderosa pine forest ecosystem (Tobo et al., 2013) and thus may not be representative of the global simulation of INP$_{PBAP}$.

Furthermore, McCluskey et al. (2018) have reported that the Wilson et al. (2015) parameterization for MPOA ice activity may overpredict INP$_{MPOA}$. Our model tends to slightly underpredict INP in the temperature range around -25°C, consistent with the earlier discussed underprediction of MPOA levels by the model. Thus, the overly active parameterization of Wilson et al. (2015), shown by McCluskey et al. (2018), may be counteracting the model's underestimation of MPOA.

Finally, INP dust parameterizations can introduce significant biases in the calculations: INP$_D$ simulations by CAM6 model were found to vary by a factor of 2 at temperatures lower than −25°C when two different parameterizations were used (McCluskey et al., 2023) with the one overpredicting INP from dust aerosol. Given the significant differences in freezing efficiency between marine and dust aerosols, simulated dust concentrations play a critical role in influencing study results, particularly in marine regions. An
overestimation of INP$_D$ could mask the potential effect of other INP types on cloud properties.

Dust particles typically have ice nucleating activities up to two to three orders of magnitude greater than marine particles at the same temperature (Cornwell et al., 2023; McCluskey et al., 2018). However, there are large uncertainties in this hypothesis since dust minerals may be carriers of biogenic ice-active matter (Hill et al., 2016; O'Sullivan et al., 2016), which would enhance dust nucleation activity at higher temperatures.
The existence of PBAP on mineral dust or dust coating by pollutants during the atmospheric ageing of dust (Iwata and Matsuki, 2018) and their effect on the ice nucleating activity of dust require dedicated studies and improved parameterizations. More generally, the co-existence, as internal mixing, of the various INP types has not been studied and could lead to an enhancement or reduction of the ice activity of the particles. In addition, atmospheric processes, such as the ageing of aerosols, can lead to the degradation of ice-nucleation
activity. For instance, ageing with sulfuric acid can either greatly reduce the immersion mode ice-nucleation activity or have little effect (Kumar et al., 2019a, b; Jahl et al., 2021), while ammonium salts can cause suspended particles of K-feldspar and quartz to nucleate ice at substantially higher temperatures than they do in pure water (Whale et al., 2018). Although there are conflicting findings and limited understanding regarding how to parameterize the ice-nucleation activity of aged particles, potential ageing effects directly
on INP are not considered in our study. The effects of atmospheric ageing on ice-nucleating activity are probably a minor source of error in this study, in comparison to the introduced bias from the employed INP parameterizations and model performance in simulating INP-relevant particles. The results presented in this study, and particularly the relative importance of various INP types, which is discussed assuming externally mixed INP, have to be revisited when additional information of the mixing of INP types and how this affects
the overall properties of the INP become available.

**4 Conclusions**

In this study, we performed global model simulations of mineral dust, primary marine organic aerosols, and terrestrial bioaerosols (bacterial and fungal spores) and investigated their contribution to atmospheric ice nucleation using laboratory-derived parameterizations based on the singular hypothesis approach. At relatively warm temperatures (warmer than −20 °C), the majority of INP are typically of biological origin, while at lower temperatures and higher altitudes, INP from mineral dust dominates globally. INP from dust contributes more to total INP in the mid-latitudes in the NH than in the SH due to the location of the dust sources and the long-range atmospheric transport patterns. INP from terrestrial bioaerosol peaks in the low and mid-troposphere and could be important in reproducing/representing atmospheric INP populations at warm temperatures. Simulated concentrations of INP from terrestrial bioaerosol vary with season and meteorological factors that affect both the source of PBAP and its ability to act as INP. INP from terrestrial bioaerosol has the potential to form ice crystals in the subtropics at the outflow of continental air. Simulated INP from marine bioaerosol is found primarily over oceans and coastal areas and dominates between 40°-90°S (Southern Ocean), with high concentrations in regions of high sea spray and phytoplankton activity that influence the source of MPOA.

The model achieves its highest predictability against observed INP across all temperature ranges when both $INP_D$ and $INP_{MPOA}$ are included. Further inclusion of PBAP in our model leads to an overestimation of the measurements, which might indicate an overestimation of the INP scheme for PBAP. Our study suggests that $INP_D$ and $INP_{MPOA}$ could sufficiently predict observed INP concentrations globally. However, bioaerosols are an important source of warm-temperature INP, affecting mainly low altitudes.

The current model biases can likely be attributed to the uncertainties in the parameterizations for the ice-nucleating activity of each particle type, simplified source functions for particle emissions, and description of aerosol transport. Uncertainties exist, particularly regarding the impact of atmospheric mixing and processing of different INP species on INP properties. For instance, dust minerals could be carriers of biogenic ice-active macromolecules and bioaerosol or be coated by sulphate and/or nitrate or other pollutants during the atmospheric ageing of dust. These processes are at present neglected or heavily parameterized in our model and could lead to the enhancement or reduction of INP ice-activation properties. Future studies should also investigate the impact of atmospheric mixing and ageing on INP concentrations and constrain them by measurements over receptor areas downwind of source regions. Due to the large variability of INP with space and time, effort must be put into acquiring more data on INP ambient levels as well as the ice nucleating properties of individual aerosol types and how these change with atmospheric ageing. These are needed to build regionally and globally representative datasets and reduce the uncertainty in the parameterizations of their sources in numerical models. Tackling these research priorities is essential for developing a more comprehensive understanding of the atmospheric variability of INP and their impacts in different on cloud regimes and climate.

**Acknowledgements.** This work has been supported by the European Union Horizon 2020 project FORCeS under grant agreement No 821205. The early stages of this work have been supported by the project "PANhellenic infrastructure for Atmospheric Composition and climatE change" (PANACEA; MIS 5021516), which is implemented under the Action "Reinforcement of the Research and Innovation Infrastructure", funded by the Operational Programme "Competitiveness, Entrepreneurship and Innovation" (NSRF 2014-2020) and co-financed by Greece and the European Union (European Regional Development Fund). ND, MK and MV acknowledge support from the Deutsche Forschungsgemeinschaft (DFG) under Germany´s Excellence Strategy (University Allowance, EXC 2077, University of Bremen). CPG-P, MC-S, MGA, and MC acknowledge support from the European Research Council under the European Union's Horizon 2020 research and innovation programme (grant n. 773051; FRAGMENT), the Horizon Europe programme (Grant Agreement No 101137680 via project CERTAINTY) and from the AXA Research Fund. MC-S has received funding from the European Union's Horizon 2020 research and innovation programme, under the Marie Skłodowska-Curie grant agreements, reference 754433 from the call H2020-MSCA-COFUND-2016. AN acknowledges support from project PyroTRACH (ERC-2016-COG) funded from H2020-EU.1.1. - Excellent Science - European Research Council (ERC), project ID 726165. SM and MK acknowledge support from the REINFORCE research project implemented in the framework of H.F.R.I call "Basic research Financing (Horizontal support of all Sciences)" under the National Recovery and Resilience Plan "Greece 2.0" funded by the European Union — Next Generation EU (H.F.R.I. Project Number: 15155).

**Author contributions.** MK and MCh conceived the study. MCh modified the model to account for INP and performed the simulations, visualized and analyzed data, and wrote the initial version of the paper. SM provided the initial code for bioaerosols of the model. MK supervised the work with feedback from AN and CPG-P and MK and CPG-P re-edited the paper. All authors provided scientific feedback contributed to the review and guidance, commented and provided revisions in subsequent versions editing of the paper before submission.

**Competing interests:** One author of the manuscript is an Atmos. Chem. Physics editor.

**Data availability.** The modeling data used for this analysis are available at 10.5281/zenodo.14616453

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

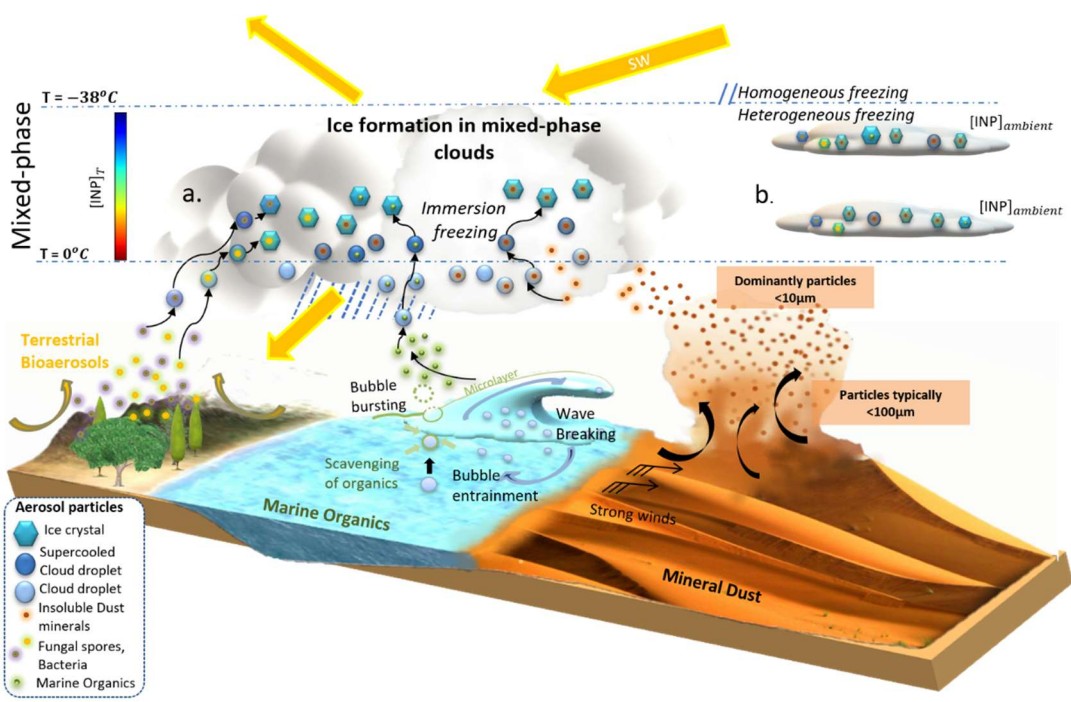

**Figure 1:** Illustration of the formation of INP from mineral dust, MPOA and PBAP, as well as the two ways in which we display INP concentrations in the present study: [INP]$_{ambient}$ is calculated at ambient model temperature relevant to non-deep convective mixed-phase clouds using the ambient temperature in the model temperature level (b.), and [INP]$_T$ is calculated at a fixed temperature relevant for vertically extended clouds as deep-convective systems (a.). The figure is
based on Chatziparaschos et al. (2023).

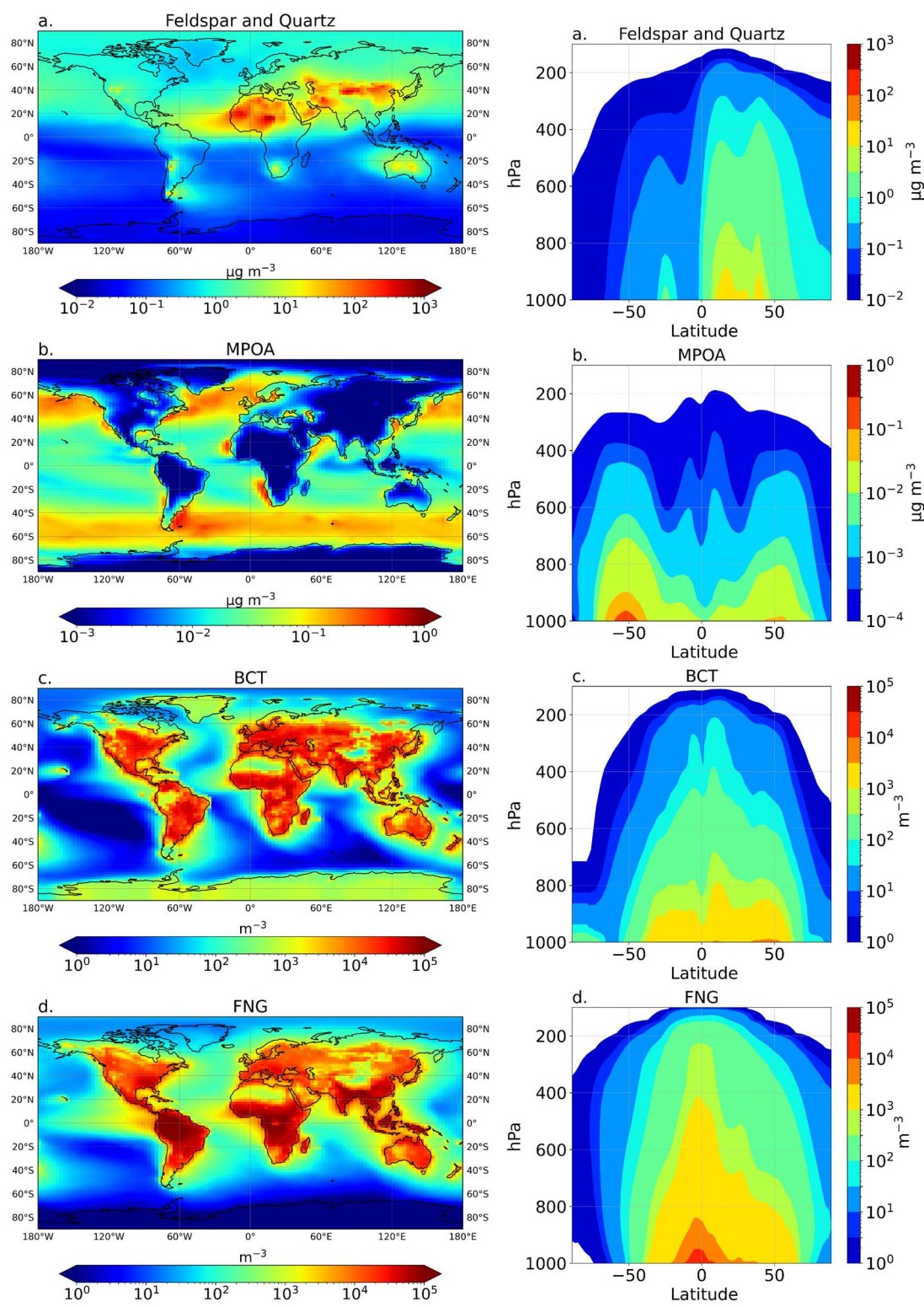

**Figure 2:** Simulated multi-year surface mean (left column) and zonal mean (right column) of (a) mineral dust (sum of K-feldspar and quartz) mass concentrations (µg m⁻³), (b) primary marine organic aerosol mass concentrations (µg m⁻³), (c) bacteria number concentrations (m⁻³), (d) fungal spore number concentrations (m⁻³).

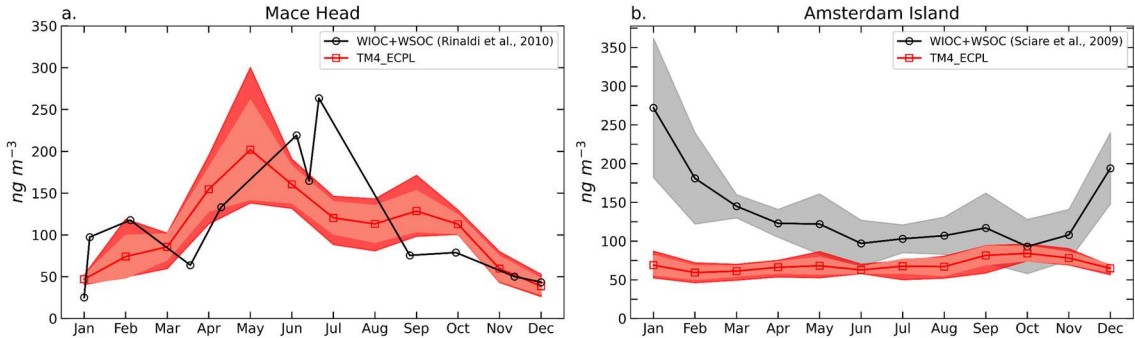

**Figure 3:** Monthly averaged concentrations of MPOA in ng-C m⁻³ at (a) Mace Head, (b) Amsterdam Island. Observations from Rinaldi et al. (2010) and Sciare et al. (2009), respectively.

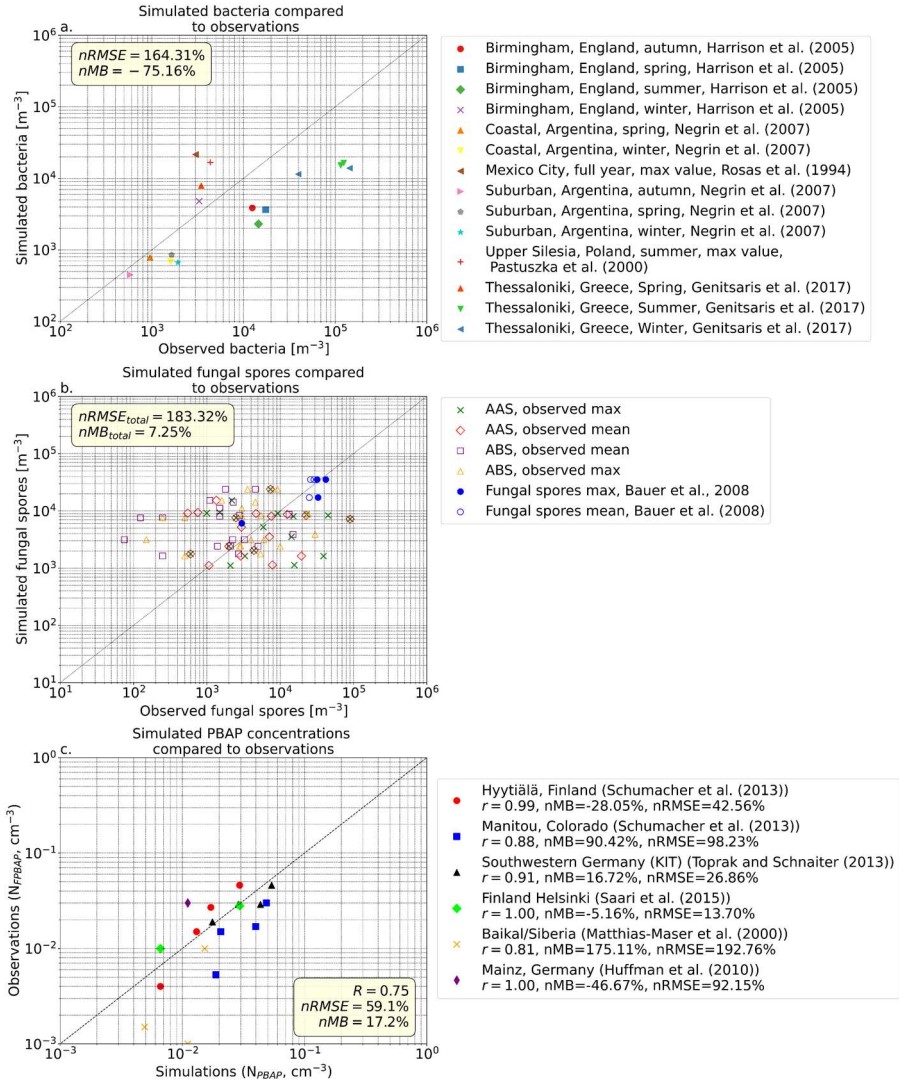

**Figure 4:** Simulated bioaerosol concentrations compared to observations. The observations are averages over different time periods, and the simulated data are taken for the corresponding months at the grid boxes containing the measurement site. (a). Bacteria concentrations; observations include concentrations listed in the supplement of Burrows et al. (2009) and Genitsaris et al. (2017). (b) Fungal spore concentrations, actively wet discharged ascospores (AAS) and basidiospores

(ABS) compared to the simulated FNG. In the model, AAS and ABS are assumed to contribute one-third each to total

fungal spores. The observations are from the references listed in Elbert et al (2007) and Bauer et al. (2008). (c) Observations of fluorescent PBAP (FBAP) compared with model simulations of PBAP (for this comparison calculated as the sum of BCT, FNG and pollen). Observations are from the compilation of long-term measurements of atmospheric fluorescent bioaerosol number concentrations listed in Petersson Sjögren et al. (2023).

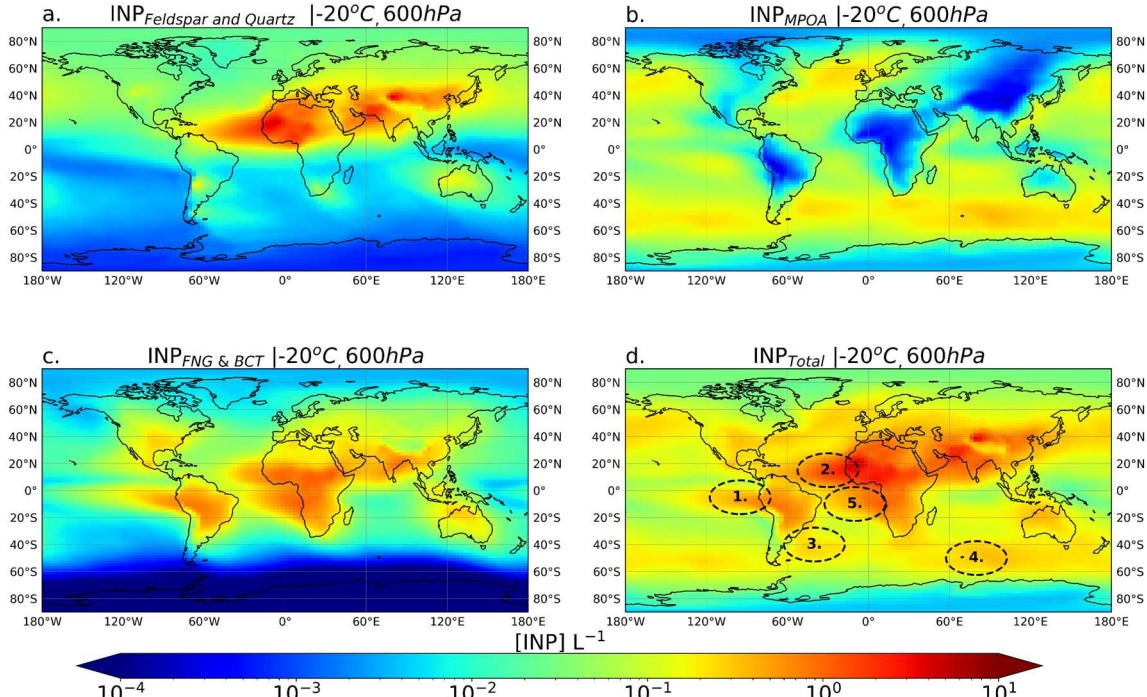

**Figure 5**: Multi-year mean distributions calculated by TM4-ECPL for INP number concentrations derived from dust (a), MPOA (b), PBAP (c), and the sum of the aforementioned INP (d) at a pressure level of 600 hPa for an activation temperature of −20ºC. Circles labelled 1 to 5 indicate areas of interest, with numbering details provided in the main text.

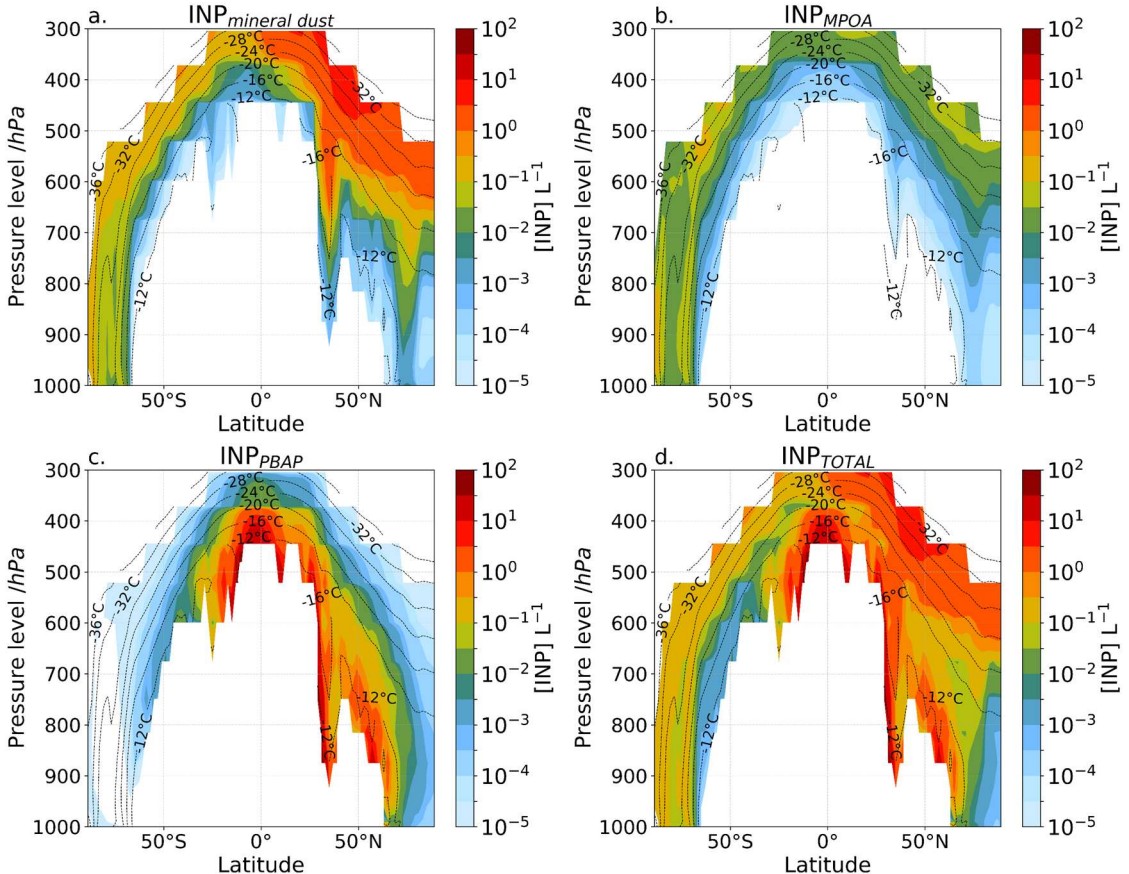

**Figure 6:** Multi-annual averaged zonal mean profiles of INP number concentration calculated at modelled ambient temperature by TM4-ECPL and plotted only where INP number concentration is larger than $10^{-5}$ L$^{-1}$ (i.e. 0.01 m$^{-3}$) and accounting for (a) mineral dust, (b) marine bioaerosols, (c) fungal spores and bacteria (PBAP) and (d) all INP types in the model. The black contour dashed lines show the annual mean temperature of the model. The colours show the INP number concentration.

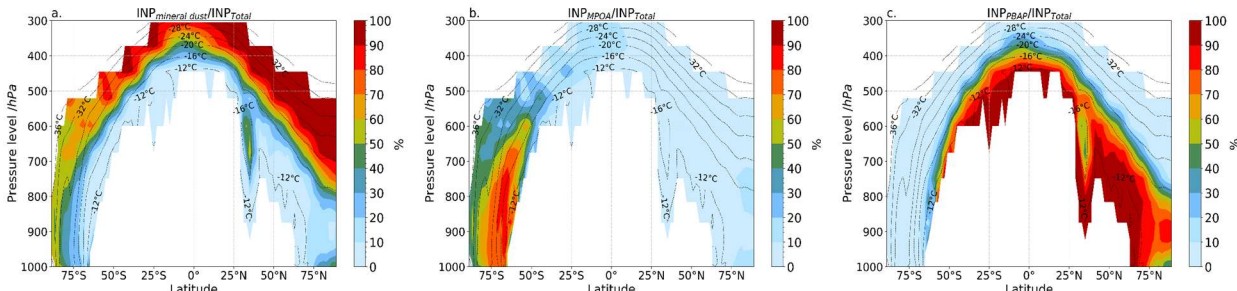

**Figure 7:** Multi-annual averaged zonal mean profiles of the percentage contribution of each species to the total INP number concentration, calculated at modelled ambient temperature by TM4-ECPL and plotted only where INP number concentration is larger than $10^{-5}$ L$^{-1}$ (i.e. 0.01 m$^{-3}$), showing contributions from (a) mineral dust, (b) marine bioaerosols and (c) fungal spores and bacteria (PBAP). Black dashed contour lines represent the model's annual mean temperature.

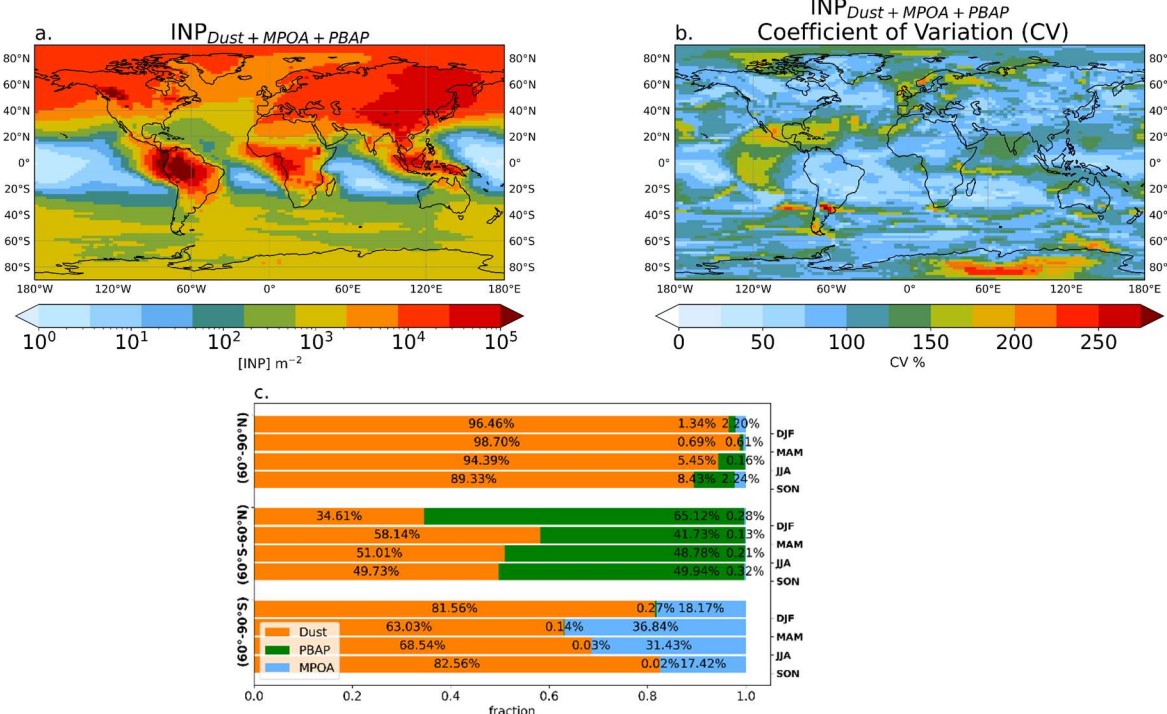

Figure 8: (a) Annual mean INP column number concentrations up to 300 hPa, for INP at modelled ambient temperature,(b) the interannual variability expressed by the coefficient of variation that is calculated as the standard deviation of the annual mean columns to the multiyear annual mean and is expressed in percent and (c) seasonal percentage contributions of INP from mineral dust (orange), MPOA (light blue), and PBAP (green), with data separated by middle latitudes (60°S–60°N) and high latitudes (60°–90°N, 60°–90°S).

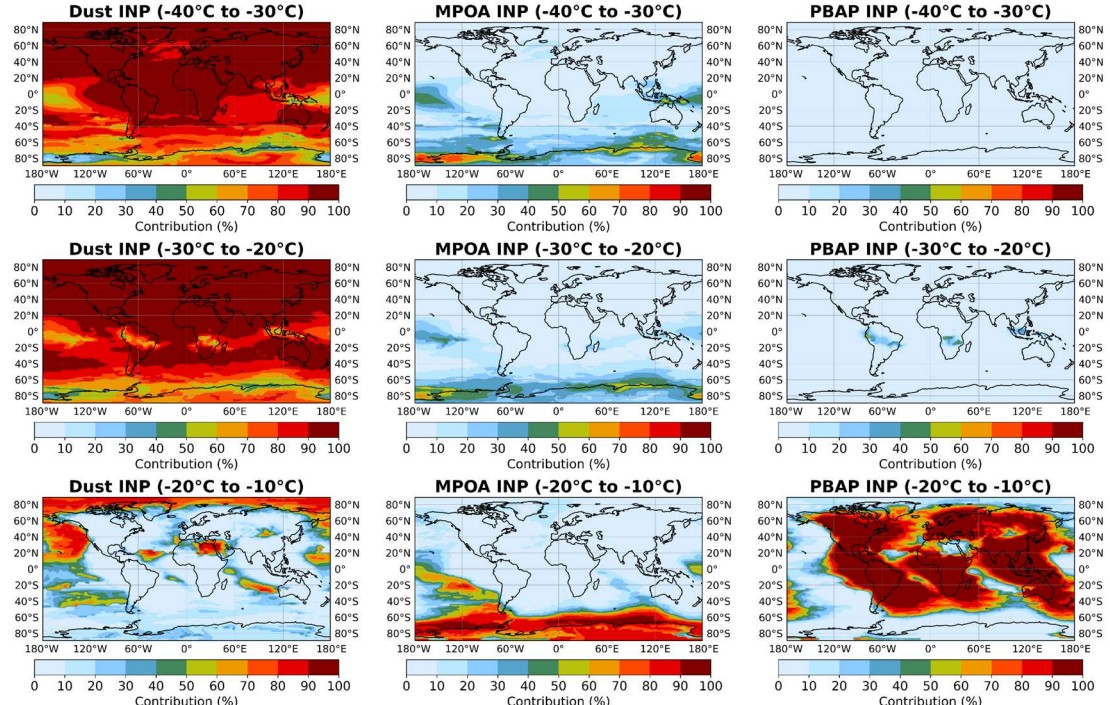

**Figure 9:** Multi-annual averaged percentage contributions of INP sources to the total INP column (surface up to 300 hPa): INP from mineral dust (left column), INP from MPOA (middle column), and INP from PBAP (right column), categorized by modelled temperature ranges: (top row) -40°C to -30°C, (middle row) -30°C to -20°C, and (bottom row) -20°C to -10°C. INP are calculated at modelled ambient temperature.

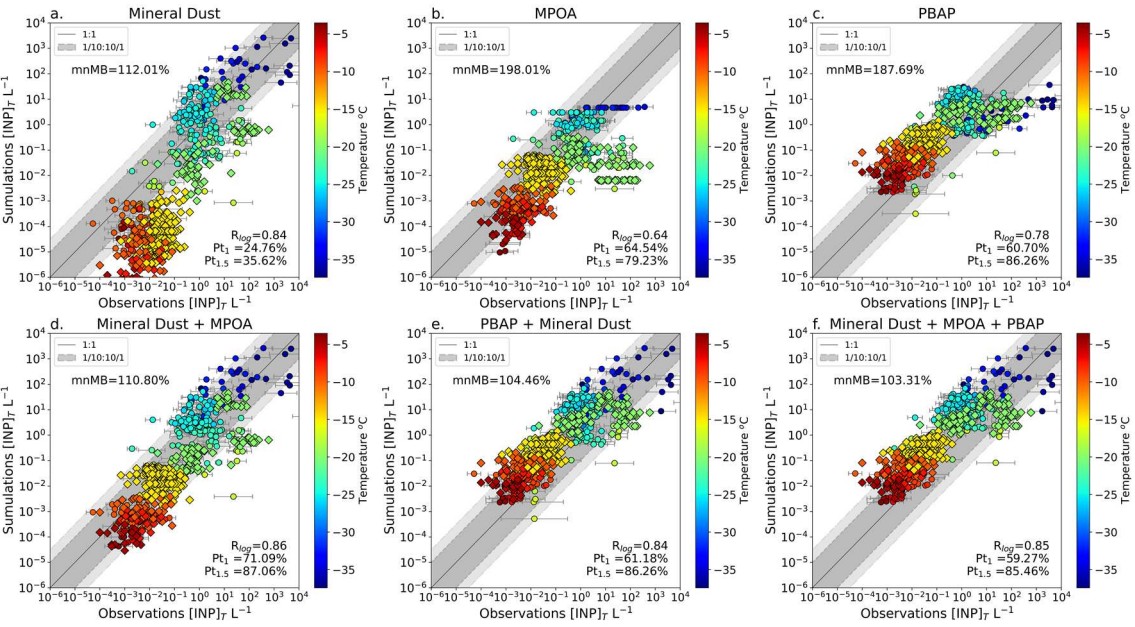

**Figure 10:** Comparison of INP concentrations calculated at the temperature of the measurements against observations accounting for simulated mineral dust (a), MPOA (b), PBAP (c), mineral dust and MPOA (d), PBAP and mineral dust (e), and mineral dust, MPOA and PBAP (f). The dark grey dashed lines represent

one order of magnitude difference between modelled and observed concentrations, and the light-grey dashed lines depict 1.5 orders of magnitude. The simulated values correspond to monthly mean concentrations, and the error bars correspond to the error of the observed monthly mean INP values. The color bars show the corresponding instrument temperature of the measurement in Celsius (a-f). $Pt_1$ and $Pt_{1.5}$ are the percentages of data points reproduced by the model within an order of magnitude and 1.5 orders of magnitude, respectively. R is the correlation coefficient, which is calculated with the logarithm of the values. Diamonds correspond to measurements (Bigg, 1973, 1990; Yin et al., 2012) that are compared with the climatological monthly mean simulations. Circles indicate comparisons between temporally and spatially co-located observations and model results. The location of the observations is shown in Fig. S1 and Table S1.