# Peer review of "Assessing the global contribution of marine, terrestrial bioaerosols, and desert dust to ice-nucleating particle concentrations"

_EGUsphere, 2024_

## Author Comment (AC1)

**REPLIES TO REVIEWER #2 COMMENTS**

We thank the reviewer for the positive and constructive comments and the time they have spent in carefully reading our manuscript. All comments have been considered in the revised version of the manuscript. Below, in black are the reviewer comments and in blue our replies.

**Summary:**

This study by Chatziparaschos et al. examines the spatial and vertical distribution of ice nucleating particles (INPs) from three different sources: dust (D), marine organics (MPOA), and terrestrial bioaerosols (PBAP) through simulations using the TM4-ECPL global chemistry-transport model. They find dust are the dominant INPs at high altitudes/cold temperatures north of ~ 40°S, and across all seasons. PBAP are the most important INP species at warm temperatures (>-16 °C), especially in the Northern Hemisphere summer. Marine organic INPs are found to have minor contributions in the Northern Hemisphere, but to dominate in the Southern Hemisphere across all seasons. Overall, these results are in agreement with prior observational and modeling studies. The best agreement between simulated and observed INP concentrations across the entire temperature range (-5 to -35 °C) are found when both dust and MPOA INPs are included, but not PBAP. All simulations including PBAP INPs have a high bias relative to observations, which may point to an overactive parameterization for this species, which should be explored in future studies.

I found the article easy to read, and the figures are well-labeled and clear. However, the discussion needs to be re-structured into a more logical flow so the conclusions build on each other. This will help reduce some of the repetition of results between paragraphs and different sections. Overall, this work provides a valuable addition to the INP community, but some additional analyses in the comparison between simulated and observed INPs would be valuable and improve the manuscript greatly, so I am recommending minor revisions. Details are provided below:

We have restructured the manuscript, as also suggested by the other reviewer, and improved the discussion of model evaluation and uncertainties of our simulations and used this information for a thorough discussion of the results of the study. Our point-by-point replies to the comments follow in blue:

**Major Comments:**

1. The structure of the discussion is confusing, and lead to a lot of repetition of results, as well as mentioning results before the corresponding figure was presented. Additionally, the evaluation of the model against observations should come earlier in the discussion, since it impacts how the reader interprets the rest of the simulated results. A suggested order is listed below:
    1. Figure 1 (schematic representation of ice nucleation). This can also be removed, as it does not add much to the discussion and is only briefly mentioned. The two types of INP concentration metric indicated in the schematic are not well explained (see major comment #2).
    2. Presentation of the global model results (Figure 5)
    3. Comparison of the model against observations (Figure 6)
    4. Additional figures looking at specific aspects of the model (Fig. 2-4)

We have restructured the manuscript considering the suggestions of both reviewers.

We kept Figure 1 and added relevant discussion in the manuscript as provided in our reply to the next point of the reviewer.

Section 3 of the results now starts with the presentation of INP-relevant distributions (new Figures 2-4 see our reply to comment #1 of reviewer #1) and their evaluation focusing on PBAP, the evaluation of which was not shown earlier (new section 3.1- *INP-relevant aerosols*) (earlier point of the reviewer; new figures 3 and 4). Then, in section 3.2 the simulated global distributions are presented and discussed in 3 sub-sections:

section 3.2.1- *INP global distributions*

section 3.2.2- *Contribution of INP types to the total INP* where the percent contributions of each INP type to the total INP as deduced by the model are shown

section 3.2.3 -*Comparison of INP predictions with observations*

After this section we added a new section on:

"*3.3 Sources of uncertainties and implications for our results.*"

that collects the discussions on the uncertainties that in the initially submitted version were in different parts of the document. In such a way we avoid repetitions and we present the results of our study together with their confidence levels.

The abstract and the conclusions have been accordingly rephrased.

2. Figure 1 presents two different ways in which INP concentrations can be calculated/presented: the concentration which reaches a specific model level/altitude and are active at the model temperature, and the concentration at a specific model level which are active at a different temperature of interest. Since these numbers can be quite different, it would be helpful to the reader to have a description of what these values are and how they differ in the main text instead of just a very brief sentence in the caption for Figure 1. And then the INP metric plotted in each figure needs to be clarified in the main text and/or figure caption. I suspect the difference between these two metrics is driving the very different patterns seen between Fig. 3 and Fig. 4, for example, but this is not clear from either the text or figures.

Indeed, Figure 1 as well as $INP_{ambient}$ and $INP_T$ needed explanation. Therefore, in the revised manuscript we added the following text at the end of the introduction.

"Figure 1 presents a comprehensive schematic illustration of INP formation derived from mineral dust, MPOA and PBAP. It includes emission processes, nucleation mechanisms, and the aerosol indirect effect, indicating the role of aerosols in cloud interactions and their impact on climate. ……

For this purpose, INP concentrations are presented in this study in two ways as $[INP]_{ambient}$ and as $[INP]_T$: $[INP]_{ambient}$ is calculated at ambient model temperature relevant to non-deep convective mixed-phase clouds using the ambient (model) temperature at each simulated level. $[INP]_T$ is calculated at a fixed temperature relevant to vertically extended clouds as deep-convective systems (Figure 1). $[INP]_{ambient}$ and $[INP]_T$ metrics can vary significantly."

later in Section 2.4 we also state:

"As outlined in the introduction $[INP]_T$ and $[INP]_{ambient}$ are determined by the simulated particle concentration and for $[INP]_T$ a specific temperature (T), while for $[INP]_{ambient}$ the model's ambient temperature is used. $[INP]_T$ is the appropriate metric when comparing modelled INP concentrations to observations, as measurements are typically conducted by exposing particles to specific, controlled

temperatures within the instruments. Thus, for the model evaluation, $[INP]_T$ concentrations are used that are calculated at the temperature at which the measurements were performed."

3. The model simulation of the aerosol types underlying the INP simulation, in this case, dust, sea spray, and PBAPs, are either only briefly mentioned as having previously occurred (dust, marine organics) or not covered at all (PBAPs). While reproducing these prior validation studies is unnecessary and impractical, I suggest adding a few sentences where the major conclusions, any biases, etc. of the previous evaluation studies (if any) are discussed. This would allow the reader to understand whether any of these biases may impact the INP results presented here.

All dust, marine organics simulations have been previously presented for specific years, without the INP component with the exception of dust that has been presented in Chatziparaschos et al., 2023. We thus focus the model evaluation on PBAP that has not been published earlier. A new section 3.1 has been added. In the sub-section 3.1.1 the global simulated distributions of the INP-relevant aerosols are presented (new Figure 2 provided in our replies to reviewer #1) and in sub-section 3.1.2 their evaluation by comparison with observations is presented (new figures 3, 4, S2 and S3 provided in our replies to reviewer #1). For dust, we summarize the outcome of the detailed evaluation done in our previous study (Chatziparaschos et al., 2023). For MPOA, in addition to reference to earlier studies, we show comparison of model results with clean air OC observations in the NH (Mace Head) and the SH (Amsterdam island). Finally, this session presents a detailed evaluation of PBAP simulations that has not been done in previous studies. Section 3.1.2 concludes :

'In summary, the evaluation of INP-relevant aerosols simulations shows that the model underpredicts MPOA (by about 30%; 3% - 50%) and overpredicts dust surface concentrations and deposition (by about 60%). PBAP concentrations are also slightly overpredicted (by about 20%), with BCT being significantly underpredicted and FNG moderately overpredicted. The model captures the seasonal variation of observed PBAP, while the simulated MPOA in the SH shows no significant variation across seasons. These model features must be taken into consideration when discussing the results of the INP simulations, since uncertainties in the INP simulations include uncertainties in both the INP parametrization and INP-relevant aerosol simulated concentrations and are affecting the importance of various INP types. '

4. The model validation against observations (Figure 6) could be improved in a few ways. First, based on Figure S1, the datasets used for evaluation are strongly biased towards the Northern Hemisphere, and the Arctic in particular. There is only one oceanic dataset included, and it occurs in a region where Saharan dust and possibly PBAP emissions are expected to dominate over marine INPs, so there are none or very few observations where MPOA INPs would be expected to be the dominant contribution. Oceanic INP measurements have been compiled in Welti et al. (2020), and many of them are from within the 2009-2016 period simulated here. Adding some of these to the evaluation would improve confidence in the simulated MPOA INP concentrations, in particular. More recent measurements of both PBAP and marine INPs exist as well, but would be more difficult to directly compare to the simulations due to the temporal offset. In addition, while the contribution of different model INP species are separated out and assessed, there appears to be no attempt to classify or separate the observations into either measured or expected dominant INP type. Why would you expect a PBAP-only parameterization to correctly simulate dust INP concentrations, for example? The fact that it is still overestimating concentrations when only PBAP are included (Fig. 6c) strongly suggests either the underlying PBAP aerosol emissions are too large and/or the INP parameterization is too active. Not all observations make measurements of INP species/type, but comparing the single-INP type simulations (Fig. 6a-c) to only observations where the INP type is known or can be reasonably estimated would improve confidence in the accuracy of the simulations. This

then has direct bearing on the percent contributions of each type calculated and shown in Fig. 2-4 and S2.

We thank the review for these comments and we tried to improve the simulated INP evaluation, further including information on dominant INP types for each observational data in addition to the evaluation of the INP-relevant aerosol in the new section 3.1.2.

All relevant to this study data from Welti et al. have been used.

Two new figures have been drawn to address the comment of the reviewer on 'separating the observations into either measured or expected dominant INP type'. The new Figure S7 shows the three different types of simulated INP and their comparison to the measured INP levels at each observational site for different latitudes zone in the atmosphere. This clearly shows the importance of different INP types for different latitude zones. For instance, in the NH extratropics (30N-60N) dust and terrestrial PBAP dominate INP levels, while in the SH extratropics (30S-60S) terrestrial (PBAP) and marine (MPOA) bioaerosol are dominating INP levels. In the tropics (30N-30S) dust and terrestrial aerosol appear the dominant types of INP.

[Figure]

**Figure S1:** Comparison of INP concentrations calculated at the temperature of the measurements against observations accounting for mineral dust (orange), MPOA (blue), PBAP (green) separated in high, middle and low latitudes.

A second new Figure S6 has been added in the supplement that shows the difference between simulated INP for each of the aerosol types from the observed INP, all observational data included, classified by temperature bin from 230K to 275K (every 5 degrees). This figure demonstrates that the dust INP shows the smallest deviations from observations among the three INP types at the lowest temperatures (230K to 245K) and at warmer temperatures (255K to 270K) terrestrial PBAP and MPOA are performing better. For the 250K to 255K dust and terrestrial PBAP behave similarly.

[Figure]

**Figure S2:** Order of magnitude / Logarithm of the deviation of simulated concentrations of INP types from total INP observations, as a function of binned temperatures. $INP_{Dust}$ in yellow, $INP_{PBAP}$ in green, $INP_{MPOA}$ in blue.

Relevant discussion is added in the revised manuscript.

**Minor Comments:**

1. Throughout the text, degree symbols are not always superscripted, and the 1 and 1.5 in $Pt_1$ and $Pt_{1.5}$ are not always subscripted.

   corrected

2. Throughout text: what is meant by INP "precursor". Based on context, it seems like INP "species" or "type" would be more appropriate and consistent with the language used by the community.

   Corrected throughout the manuscript

3. Line 26: replace ice "crystals" with ice "processes". DONE
4. Line 41: replace "dust and marine and" with "dust, marine, and" DONE
5. Line 42-44: the last sentence of the abstract does not fit with rest of the paragraph. I suggest just removing it. DONE
6. Line 49: define mixed-phase clouds as MPC here, otherwise you do not define the acronym before you use it. DONE
7. Line 50-51: The sentence "A significant feature…" is confusing. What do you mean by this/why is it important?

   The sentence is now rephrased: 'A significant feature of MPC is the degree of homogeneity of mixing ice particles and liquid droplets that affects the rate of precipitation formation (Korolev and Milbrandt, 2022)'

8. Line 60: remove "saturation" from "saturation water vapor pressure". You appear to be discussing factors that alter ambient vapor pressure and not the saturation value, which is an intrinsic value characteristic of a set of environmental conditions.

   DONE

9. Line 61: add "area" after "ice crystal surface" DONE
10. Line 69-71: Please include references to support that SIP influences "precipitation ranges, cloud properties, and albedo." What are precipitation ranges?

   References are included. 'Precipitation ranges' has been removed.

11. Line 117: "INP contributor" should be "INP contribution" DONE
12. Line 117: "Nevertheless, there are other minerals" should be changed to something like "There are also other minerals" DONE
13. Line 131-132: "Recent advances in remote sensing and model/sensor fusion have enabled the presence of secondary ice in clouds, revealing that this process holds…" should be changed to "Recent advances in remote sensing and model/sensor fusion have enabled **detection** of secondary ice **processes** in clouds, revealing that **they hold**…"

   corrected

14. Line 137: ",with about" should be ", accounting for" DONE
15. Line 149: typo in "sea spay", should be "sea spray" DONE
16. Line 152: Burrows et al. (2013) and DeMott et al. (2016), in addition to Wilson et al. (2015) are more appropriate references instead of Mitts et al. (2021), as they are some of the first studies to suggest this idea.

corrected

17. Line 153: MPOA and SS should be reversed in this sentence.

DONE

18. Line 167-170: This entire sentence is almost word for word copied from Wilson et al. (2015). Either indicate it is a direct quotation, or rephrase in your own words.

We thank the reviewer for pointing this out. The sentence is now rephrased and reads as follows:' Marine INP concentrations are found to maximize over areas of high biological activity and wind speed in the Southern Oceans, the North Atlantic, and the North Pacific, while mineral dust from deserts (the main terrestrial source of INP) accounts for only a small fraction of the INP observed in this region (Wilson et al., 2015; Huang et al., 2018)'

19. Line 179-180: This sentence is a bit misleading. The simulations are based on others' work identifying major INP types, which you (reasonably) take advantage of. Please rephrase to more accurately describe your contribution.

Presentation and discussion of Figure 1 has been changed and extended as stated in our reply to the major comment #2 of the reviewer.

20. Lines 239-241: How representative are "diverse locations across Europe" for PBAP emissions globally? Especially since Fig. 5 indicates the PBAP INP simulations peak in Africa and South America? Can you comment on how representative the Global Land Cover database from 2000 is for simulations that occur 1-2 decades later?

Indeed, these are very good points raised by the reviewer. Parameterizations based on observations over Europe might not be accurate for the flux calculations over other regions of the globe. This is now commented in the manuscript." This parameterization is informed by field measurements of fluorescent biological aerosol particles conducted at diverse locations across Europe; thus, it might not be very

accurate for other regions and will be improved when data from other regions of the globe will become available."

Copernicus provides now Land Cover data at very high spatial resolution for the years 2015-2019 updated every year. Note that our simulations cover the years 2009-2016 and have to be considered as a climatological view of the atmosphere over that period. Thus, using a 2000 Global Land Cover database introduces some uncertainties but is not expected to qualitatively change the conclusions of the study.  In the text we have added : "which represents a climatology for the present study since no changes in vegetation from year-to year are taken into account"

21. Line 269: Are MPOA and SS internally or externally mixed in the model? This will impact what diameters are simulated.

MPOA is considered to be a coating of SS thus internally mixed as described in detail in O'Dowd et al. (2008) and Vignati et al. (2010). To avoid any confusion we rephrased this sentence as follows: MPOA is emitted in the fine mode together with SS in amount that depends on Chl-a in the seawater and assuming that it is entirely insoluble as determined by O'Dowd et al. (2008) and applying the same modeling approach as in (Vignati et al., 2010).

22. Line 288: What is meant by "the spectrum of ice nucleation properties"?

corrected to 'ice nucleation spectrum'

23. Line 317-318: The sentence "Additionally, observations…" does not fit well in this paragraph.

We moved this sentence one sentence up. The text now reads" The number fractions of FBAP to the total aerosol particles in the super-micron size range could be relatively similar to those of PBAP under dry conditions (Tobo et al., 2013). Indeed, observations by Negron et al. (2020) suggest that airborne bacteria may be unambiguously detected with autofluorescence. However, during and after rain events, PBAP were underestimated by more than a factor of 2 (Huffman et al., 2013), suggesting that FBAP reflect only a portion of PBAP."

24. Line 330: In equation 3, is $C_{MPOA}$ meant to be $C_{TOC}$?

corrected

25. Line 340: Simulated INP concentrations were compared to measurements at the same temperature and date. Were the lat/lon and altitude of the measurements also matched?

Yes. Text now reads: 'All model results are compared to observations for the specific month and year as well as location of observation, except for those reported by Yin et al. (2012) and Bigg campaigns (Bigg, 1990, 1973), which cover temporally scattered measurements (between 1963 and 2003) and are therefore compared to the modelled multi-annual monthly mean INP concentration from 2009 to 2016.'

26. Paragraph beginning with line 347: This paragraph jumps around and is hard to follow. The altitudes, temperatures, latitudes, and INP type discussed vary from sentence to sentence. It would be helpful to discuss Figure 2 for all INP types, then move on to figure S2 and the seasonality of the simulations, instead of sprinkling that into some of the paragraphs and then repeating much of that information later in the discussion. Focus this first paragraph on dust and move the couple of sentences about PBAP and MPOA to the next 2 paragraphs.

This section has been reorganized following the reviewer's suggestion.

27. Line 358: "amount of minerals" should be changed to something like "type and fraction of different minerals"

Change performed

28. Line 361-362: The statement that "PBAP are the primary source of INP between −12°C and −20°C." should come after the evaluation of model results demonstrating the overestimation of PBAP by the model. See major comment #1.

In response to the reviewer comment #1 we have added a new section 3.1 evaluating PBAP simulations that has shown an overestimate with a normalized mean bias (NMB) for the total PBAP of 17% and overestimate (NMB 7%) for fungal spores and underestimate (-75%) for bacteria. This evaluation section is now before the discussion of the INP results.

29. Line 372-373: The sentence "INP$_{PBAP}$ mainly affect…" simply repeats what was already said and is unnecessary.

The sentence has been removed.

30. Line 374: "(Fig. 2S)" is actually Fig. S2.

corrected

31. Paragraph beginning with line 369 needs some work, which will be helped by restructuring the discussion in line with major comment #1. The sentence beginning with "In boreal summer…" belongs with a discussion of Fig. S2 and not Fig. 2, and the one beginning "Our results suggest that PBAP…" with the discussion of Fig. 5. The last two sentences of this paragraph jump around between INP types and repeat things that were said in the preceding paragraph.

Corrections done as suggested

32. Line 383: Clarify that the "concentration range **shown in Wilson et al. (2015)…**" are, in fact, simulated concentrations and not actual observations.

For clarity, we rephrased as 'simulated concentration range shown in Wilson et al. (2015)'

33. Line 387: Chubb et al. (2013) presented measurements of SLW and not INPs. Either clarify this, or cite something specific to INPs, such as McCluskey, Hill, et al. (2018) or Welti et al. (2020).

We now refer to McCluskey et al. (2018) and Welti et al. (2020)

34. Line 391: What models or types or models (ie global, cloud resolving) are you referring to? There have been significant improvements in many models between CMIP5 and CMIP6, also.

We replaced models by 'Earth System Models'

35. Line 395: See Gettelman et al. (2020) and Bodas-Salcedo et al. (2019) for improvements in radiation budgets following an increase in SLW in two different models, in contrast to Huang et al. (2018).

This part of the discussion has been removed during the restructuring of the manuscript.

36. Line 395-397: See also McCluskey et al. (2023), which shows significant differences in CAM6 between two dust parameterizations, one of which was used in Huang et al. (2018) and overpredicts dust INP in the Southern Ocean, at least. I am not familiar with the validation of the ECHAM6-HAM2 model used in Huang et al. (2018), but simulated dust concentrations will also strongly impact these results because of the large discrepancy in freezing efficiency between marine and dust aerosol.

We have included the reviewer's comment in the manuscript – in the new section 3.3. on uncertainties we now refer to this source of uncertainty in model simulations:

" INP dust parameterizations can introduce significant biases in the calculations: $INP_D$ simulations by CAM6 model were found to vary by a factor of 2 at temperatures lower than −25°C when two different parameterizations were used (McCluskey et al., 2023) with the one overpredicting INP from dust aerosol. Given the significant differences in freezing efficiency between marine and dust aerosols, simulated dust concentrations play a critical role in influencing study results, particularly in marine regions. An overestimation of $INP_D$ could mask the potential effect of other INP types on cloud properties. "

37. Line 402: "figure 2c" should be capitalized. Also, it is not obvious in Fig. 2c that MPOA are enhanced at the surface, especially not in the Arctic. Please specify exactly what is plotted in Fig. 2- are the concentrations showing only INP active at the model temperature level, such that MPOA do not appear enhanced at the surface because they are not active at very warm temperatures? Please also check the colorbar scale for Fig. 2d-f. Based on Fig. S2, I would expect the MPOA % contribution to be higher in Fig. 2f, since it dominates from 60-90 °S and up to ~500 hPa during all seasons.

The old Figure 2, in the revised manuscript is split in two new Figures 6 and 7 that are also redrawn for better quality. Indeed, these figures (Figure 6) shows only active INP number concentrations at ambient (model's) temperature and the scale is in $L^{-1}$. Figures 7 show fractional contributions of the different types of INP to the total simulated INP and the scale is in percent units. These are computed for ambient (model) temperatures.

[Figure]

**Figure 6:** Multi-annual averaged zonal mean profiles of [INP] number concentration calculated at modelled ambient temperature by TM4-ECPL and accounting for (a) mineral dust, (b) marine bioaerosols, (c) fungal spores and bacteria (PBAP) and (d)all INP types in the model. The black contour dashed lines show the annual mean temperature of the model. The colours show (a-c) the INP number concentration.

[Figure]

**Figure 7:** Multi-annual averaged zonal mean profiles of the percentage contribution of each species to the total INP number concentration, calculated at modelled ambient temperature by TM4-ECPL, showing contributions from (a) mineral dust, (b) marine bioaerosols and (c) fungal spores and bacteria (PBAP). Black dashed contour lines represent the model's annual mean temperature.

38. Line 404: Replace "high pressure levels" with "warmer temperatures" for clarity.

done

39. Line 407-410: Note Fig. S3 in addition to S2, otherwise it is not referenced anywhere in the main text. The statement about PBAP concentrations refers to S3, for example, but is not listed. This location may change if the discussion is re-structured.

done

40. Line 411-412: PBAP INPs appear to dominate at low altitudes in the Northern Hemisphere year-round, based on S2, and likewise for MPOA INPs in the Southern Hemisphere. The seasonality is not clear in Fig. S2, except for >60 °N. Also, what is meant by "showing alternating patterns influenced by vegetation and ocean biota"?

This discussion has been modified:

"In the boreal summer, the high INP$_{PBAP}$ contributions to total INP are shifted to northern latitudes due to vegetation growth (Fig. S4). … There is a pronounced seasonality in dust INP (Figure S5), with large concentrations observed between 40°N and 90°N during the boreal winter and spring (DJF, MAM) when transported dust can influence INP concentration over Arctic.  In contrast, INP$_{MPOA}$ shows minimal seasonal variation and consistently dominates the SH between 40°S and 80°S. Between 60°S and 60°N, INP$_{PBAP}$ is the most prevalent INP type at higher pressure and temperature levels, especially in the NH during the boreal summer (JJA) and autumn (SON), when its contribution to total INP increases."

41. Line 415-417: It is a little difficult to directly compare Fig. S2 and S3 to observational data which was all made at the surface. The concentration of marine INPs active at any given temperature would be expected to be higher near the surface, since that's the source. But since I think you have plotted the number contributing at a seasonally averaged isotherm, your percentages will be very sensitive to the exact parameterization used to determine the temperature dependence. Please clarify if what is being plotted is the INP concentration at the model temperature and altitude. If not, have you explored why the MPOA and PBAP do not appear to peak at the surface? You should also clarify that the region you are discussing here is the Arctic (ie >60 °N), and the altitude/temperature range you are referring to. Could the fact that PBAP appear to contribute more than MPOA in summer be related to the apparent overestimation of INP PBAP by the model (Fig. 6)?

For PBAP and MPOA to act as INP since for PBAP the temperatures should be in the range of [-34°C, -9°C] and for MPOA in the range of [-27°C, -6°C] and for dust k-feldspar in the range of [-37.5°C, -3.5°C] and quartz in the range of [-37.5°C, -10.5°C] as was described in section 2.3. Furthermore, the contribution of the INP types (Figure 7, initially Figure 2d,e,f, and Figures S4, S5, initially Figures S2 and S3) to the total INP depends on both the ambient temperature and on the actual concentrations of all three particle types PBAP, MPOA and dust, whereas the INP_$_{MPOA}$ and its seasonality (Figures S4, S5) depend on the temperature and the actual concentrations of only MPOA which are expected to vary seasonally. The larger contribution of PBAP to INP than MPOA at >60°N in summer could be due to an overestimate of PBAP levels as discussed in new section 3.1.2 on model evaluation.

This is now stated in the manuscript "This discrepancy may be due to the underprediction of MPOA and overprediction of PBAP and dust concentrations discussed in section 3.1.2."

The sensitivity of the results to the INP parameterizations used is discussed in the new section 3.3. on uncertainties.

Furthermore, the figure captions now clarify that the black contour lines represent seasonal (annual) mean isotherms in degrees centigrade.

[Figure]

**Figure S3:** This figure depicts the seasonal percentage contribution of a) mineral dust, (b) fungal spores and bacteria and (c) marine organic aerosols calculated by TM4-ECPL where the total [INP]$_{ambient}$ concentration is larger than $0.01\mathrm{m}^{-3}$. The black contour lines represent seasonal mean isotherms in degrees centigrade.

[Figure]

**Figure S4:** This figure depicts the seasonal concentration of a) mineral dust, (b) fungal spores and bacteria and (c) marine organic aerosols calculated by TM4-ECPL where the total [INP]$_{ambient}$ concentration is larger than $0.01 m^{-3}$. The black contour lines represent seasonal mean isotherms in degrees centigrade.

42. Line 431-432: Capitalize F in figure 3, replace "INP total number concentration column" with "total INP column (number) concentration". Can you also clarify what INP metric is plotted in Fig. 3? Is it the concentration of INPs active at each temperature which reach the range of temperatures listed (ie the -40 to -30°C values are at higher altitudes and the -10 to -20°C values are from near the surface) or the "potential" total concentration active at each temperature, regardless of whether they reach the altitude corresponding to the listed temperature range?

This figure (Figure 3 in the initial submission) shows the total INP column (up to 300 hPa) calculated by sampling the model based on the model temperature that is separated in three distinct temperature bins [-40 °C, -30 °C], [-30 °C, -20 °C] and [-20 °C, -10 °C]. So, it shows the actual (not the potential) total INP active at each temperature bin in the model column up to 300 hPa. This figure has been replaced by a new one (Figure 9 in the revised manuscript) that shows each type of INP separately at the three ranges of temperature. This is now clarified in the manuscript:

"To further illustrate the different components of INP depending on atmospheric temperature, Figure 9 presents the percent contributions of INP types calculated at modelled ambient temperature and classified in three temperature ranges: [-40°C, -30°C], [-30°C, -20°C], and [-20°C, -10°C]. The figure highlights the spatial variability as a function of temperature of the contribution of the studied species to the total simulated INP at modelled ambient temperature. In order to plot this figure, all model results for which modelled temperature falls into the respective temperature range are selected. The percent contributions of each INP type to the total INP in each of the model grids and instances are calculated and then averaged per model grid for all altitudes. "

[Figure]

Figure 9: Multi-annual averaged percentage contributions of INP sources to the total INP column (surface up to 300 hPa): INP from mineral dust (left column), INP from MPOA (middle column), and INP from PBAP (right column), categorized by modelled temperature ranges: (top row) -40°C to -30°C, (middle row) -30°C to -20°C, and (bottom row) -20°C to -10°C. INP are calculated at modelled ambient temperature.

43. Line 434: Is the annual mean % contribution listed also from 2015? How much annual variability is simulated by the model?

All figures now refer to multi-year simulation results.

44. Line 444: "INP sources" is more correctly "INP types" or "INP species"

Corrected throughout the text to INP types

45. Line 444-445: Please clarify what is meant by "correlation between the percentage contribution of each INP type and the concentration ranges of INP." What is a concentration range? The % contribution is calculated from the concentration of one species divided by the total, so must mathematically be correlated with the species concentration.

For clarity this figure has been replaced by a new figure number 8 and discussion has been modified accordingly

[Figure]

Figure 8: (a) Annual mean INP column number concentrations up to 300 hPa, for INP at modelled ambient temperature,(b) the interannual variability expressed by the coefficient of variation that is calculated as the standard deviation of the annual mean columns to the multiyear annual mean and is expressed in percent and (c) seasonal percentage contributions of INP from mineral dust (orange), MPOA (light blue), and PBAP (green), with data separated by middle latitudes (60°S–60°N) and high latitudes (60°–90°N, 60°–90°S).

46. Line 457-458: Is 27% the annual average of INP$_{PBAP}$, if so, for which year? The fall INP$_{PBAP}$ contribution is 45%, but it is lower in the summer, ~30%. Provide an average if you wish to talk about both together.

Following the change of figure to new Figure 8, the contribution of the different INP types is now discussed on a latitudinal zone and seasonal basis. This discussion now reads:

'Figure 8 provides a comprehensive view of the global distribution of INP in the tropospheric column from the surface up to 300 hPa (Fig. 8a), calculated at the ambient modelled temperature, as well as the percentage contribution of each INP type per season and for three latitudinal zones (Fig. 8c). High INP column burdens are simulated in the NH and especially over the Sahara, parts of the Middle East, and the northern part of South America. These regions are known to be significant dust emission and biological aerosol sources (Amazon). Lower INP column burdens are simulated over the polar and oceanic regions, indicating that the contributions from INP sources in these areas are limited. The simulated INP column burdens are greater in the NH than in the SH, which is in agreement with previous studies (Vergara-Temprado et al., 2017). Figure 8b shows the associated interannual variability of the simulated INP multiyear mean columns indicated by the coefficient of variation that is calculated as the standard deviation of the annual mean columns to the multiyear annual mean and is expressed in percent. We find that locally the interannual variability can exceed 150%, particularly in the tropics and the polar regions and has to be kept in mind when interpreting modeling results. Figure 8c shows that mineral dust is the dominant source of INP in the high northern latitudes (60°-90°N) across all

seasons, with contributions exceeding 89% in every season. $INP_{PBAP}$ and $INP_{MPOA}$ play minimal roles, with $INP_{PBAP}$ contribution to the column burden of the total INP peaking at about 8.5% in the fall (SON). In mid-latitudes and the tropics (60°S-60°N), INP contributions are more seasonally variable, highlighting the interplay between dust and biological particles in this region. Mineral dust dominates during MAM (spring NH) and SON (autumn NH), while $INP_{PBAP}$ plays a prominent role in DJF (winter NH), contributing about 65%. In the high southern latitudes (60-90°S), mineral dust remains the main source of INP, especially in DJF (summer SH), where it accounts for about 82.5%. However, in contrast to the other latitude zones, $INP_{MPOA}$ shows significant contributions to the total INP column burden in MAM (about 37%) and JJA (about 31.5%) in this latitude zone, while $INP_{PBAP}$ contribution is negligible.'

47. Line 466 and 472: Do you mean the spatial "distributions of INPs"? Either clarify or just replace with concentration.

This now reads:' the simulated spatial distributions of INP concentrations'

48. Line 467: Why was 600 hPa chosen as an example pressure level? Most of the observations you compare to later (Table S1) appear to be from ground sites, so surface concentrations at any temperature of interest would seem more appropriate. Also, Fig, 4a appears to show INP concentrations below detection limit between ~30 °S and 30 °N at 600hPa. How do you have almost global coverage in Fig. 3 and 5? It is not clear in most of the figures which metric of INP concentration you are plotting, and it is difficult to decipher why there are large differences between the maps.

The pressure of 600 hPa corresponds on average to about 4 km i.e. is above the boundary layer in the free troposphere. The INP samples even from near- ground have been measured at instrument's temperature which is much lower than the ambient one. Accounting for an adiabatic dry air temperature decrease of 6.5 °C/km (about 6°C/km for wet air), at 4km height temperature should be 24-26°C lower than at surface. For an average surface temperature of about 14°C, this ends up to about -14 °C to -16°C average temperature at 4km, which is close to the instrument's temperature when measuring INP.

The following explanation is added in the revised manuscript: "This pressure level corresponds to the low free troposphere and average temperatures broadly consistent those used for the INP measurements."

49. Line 468-469: What is meant by "allow all species to activate and act as INP"?

The sentence now reads. "These conditions of temperature and pressure are representative of MPC's glaciation."

50. Paragraph beginning with line 474: You don't really explain what is meant by $INP_{D-20}$ or $INP_{PBAP-20}$, etc, consider replacing with, eg, "$INP_D$ at -20 °C".

Modified as suggested throughout the manuscript.

51. Lines 475-476: The phrasing "transport patterns from sources that favour dust present in the NH." is confusing. The wording used in the conclusion is clearer.

This sentence has been rephrased as follows: "[$INP_D$[600hPa, -20°C] (Fig. 5a) are higher in the mid-latitudes of the NH than in the SH due to the location of dust sources and long-range atmospheric transport patterns that favour the presence of atmospheric dust in the NH."

52. Line 482: How is the upper troposphere a "marine environment"? In addition to the possible influence of PBAP in marine environments, dust INPs contribute significantly in many areas. This is particularly clear over the central Atlantic, downwind of the Sahara, where dust INP concentrations are larger than both MPOA and PBAP. And dust also contributes significantly over the N Pacific down wind of continental outflow.

This sentence has been rephrased for clarity: "Remarkably, there are oceanic regions in the South Atlantic Ocean and Pacific Ocean (such as the south hemisphere tropical west coasts of South Africa and South America) where PBAP has the potential to form ice crystals at the outflow of the continental air (Fig. 5d, circles 1 and 5), enhancing INP concentrations in marine atmosphere." Further in the discussion of that figure we also comment: "$INP_D$[600hPa, -20°C] shows also significant levels over the North Pacific, where dust is carried by continental outflows. Mineral dust and MPOA control the INP population over South Atlantic adjacent to North east coast of S. America due to Patagonia desert and marine biota (Fig. 5d, circle 3), while over southern Indian Ocean $INP_{MPOA}$[600hPa, -20°C] dominates (Fig. 5d, circle 4)."

53. Line 484: It would be appropriate to add somewhere a discussion of the observations, and the mix of INP types observed or expected to be observed in those locations. There are no observations included, for example, where MPOA are expected to be the dominant INP type, since the only cruise included is immediately down wind of Africa, where significant dust and PBAP emissions are expected. See Major Comment #4.

In the revised version of the manuscript, a new supplementary figure S7 has been added which, using the model results, separates the observations into the expected dominant INP type. See our response to major point 4 for further details.

54. Line 488-489: What altitude/pressure were the simulated concentrations in Fig. 6? If the 600 hPa from Fig. 5, why, when the majority of observations are surface measurements? Did the lat/lon and altitude of the simulated values match each observation?

Comparison is done on lat/lon/altitude basis of the observations and NOT at 600 hPa. We also reported this in our reply to point 25 of the reviewer: 'All model results are compared to observations for the specific month and year as well as location of observation'

55. Line 495: Simulating dust INPs alone appears to compare quite well against the observations below ~-25 °C.

Yes, this is now demonstrated better with the new supplementary figure S6 that shows the difference between simulated INP for each of the aerosol types from the observed INP, all observational data included, classified by temperature bin from 230K to 275K (every 5 degrees). This figure demonstrates that the dust INP shows the smallest deviations from observations among the three INP types at the lowest temperatures (230K to 245K) and at warmer temperatures (255K to 270K) terrestrial PBAP and MPOA are performing better. For the 250K to 255K temperatures dust and terrestrial PBAP behave similarly.

56. Line 501: "PBAP onto mineral" should be "PBAP **on** mineral"

DONE

57. Line 507: Isn't is more likely the discrepancy between the MPOA-only simulation and the observations is due to the observations containing (mostly) measurements that are not MPOA? It would be informative to separate these single-INP type comparisons (Fig. 6a-c) and compare to similar observations. See Major Comment #4.

We followed the reviewer's suggestion, see our reply to major comment # 4.

58. Line 509: Wilson et al. (2015) did not use Chl-a to parameterize MPOA. SML INP concentrations were scaled by measured TOC, and this scaling factor was applied to modeled organic matter in sea spray.

We admit that these sentences are confusing, Chl-a parameterization is used in the model for the MPOA source calculation. As we have also restructured the manuscript and added an evaluation of the INP-relevant aerosols as well as a section 3.3. on uncertainties, the above discussion has been modified and now reads:

"The discrepancy between observed INP and $INP_{MPOA}$ can be partially attributed to missing INP sources, especially dust, and partially to the uncertainty in the $INP_{MPOA}$ simulations resulting from both the MPOA simulations and the $INP_{MPOA}$ parameterization."

59. Line 509-512: The overestimation of Wilson et al. (2015) seen in McCluskey, Ovadnevaite, et (2018) was specific to observations of marine aerosol. Why would you expect a MPOA-only parameterization to overpredict total INP, when dust is much more IN-active? Are the observations around -25 °C expected to be MPOA INPs?

This discussion has been modified in the restructured revised version of the manuscript (see reply above).

60. Line 515: I think you mean the McCluskey, Ovadnevaite, et (2018) parameterization here, not Wilson et al. (2015).

done

61. Line 517-520: This sentence requires more explanation or clarification. Why does sea spray surface area require detailed organic composition information? What do you mean by "SS size variability that occurred due to marine biological processes"? SS will change size once in the atmosphere, but the only direct link to marine biology after emission would be through condensation of emitted gases. What "process is not parameterized in McCluskey et al. (2018)" that is needed? Since you do not implement or otherwise discuss the McCluskey, Ovadnevaite, et (2018) parameterization, this sentence can probably be removed.

This part of the discussion has been removed.

62. Line 523-527: See Raman et al. (2023) for a more nuanced discussion of why equating SML and SS INPs may be a poor assumption. Organic enrichment factors between the SML and SS are hugely variable based on the specific organic composition, in addition to the aerosol production mechanism. The evaluation of TM4-ECPL OC in Myriokefalitakis et al. (2010) indicated a general underestimation of POC, and virtually no seasonality in the model. Even if the Wilson et al. (2015) relationship (which included only Arctic measurements) between SML TOC and atmospheric INPs is globally applicable to SS, you have not evaluated any other parameterization(s), and so have not provided any direct evidence your model is "more realistic". And indeed, the studies which have evaluated both the Wilson (2015) and McCluskey, Ovadnevaite, et (2018) parameterizations generally find the latter works better. An overactive Wilson (2015) parameterization may be counteracting the underestimation of OC by the model. The more complicated approach may be theoretically more realistic, but only if all the component parameterizations are very accurate. Can you comment on this at the fairly coarse model resolution used here?

We thank the reviewer for this comment. Indeed, as shown in Myriokefalitakis et al. (2010) and in new Figure 2 and discussed in section 3.1.2, the model underestimates OC in the marine environment. Here we used the Wilson et al (2015) parameterisation to derive the $INP_{MPOA}$ which overestimates the ice activation of MPOA and indeed this seems to partially balance the underestimate of MPOA. This is now added in the manuscript in section 3.3: 'McCluskey et al. (2018) have reported that the Wilson et al. (2015) parameterization for MPOA ice activity may overpredict $INP_{MPOA}$. Our model tends to slightly underpredict INP in the temperature range around -25°C, consistent with the earlier discussed underprediction of MPOA levels by the model. Thus, the overly active parameterization of Wilson et al. (2015), shown by McCluskey et al. (2018), may be counteracting the model's underestimation of MPOA.'

63. Line 533: Could this discrepancy also be attributed to comparing with observations that are not dominated by PBAP INPs?

Yes, it is possible that the observed total INP is influenced by different types of INP than those simulated. In addition, the deviation of simulated individual INP types from observations, shown in the new Figure S6 as a function of binned temperatures, indicates that at temperatures of about -25 to -30°C, both $INP_D$ and $INP_{PBAP}$ separately overestimate the observed INP. For warmer temperatures, Figure S6 clearly shows that a major contribution to the model overestimation of the observed total INP is due to $INP_{PBAP}$.

64. Line 536: Replace "ice efficiency" with "ice nucleating efficiency"

done

65. Line 537-539: A similar limitation is true of the Wilson (2015) parameterization, which uses data from a small number of Arctic SML samples and then generates a global parameterization using simulated (not observed) atmospheric INP concentrations. Some caveats or discussion is warranted in the preceding paragraph or in the methods, as was given here for the Tobo et al. (2013) parameterization for PBAP.

The following text has been added in section 3.3. on uncertainties: "Parameterisations and model evaluation are based on observations geographically and temporally limited and used for global estimates; therefore, their global applicability remains an open issue."

66. Line 544: "highest correlation is found" should be replaced by "highest correlation between simulated and observed INP concentrations is found".

done

67. Line 549: Add "of" between "correlation coefficient" and "0.88". Remove the reference to Fig. 6c, the correct Fig. 6d is referenced earlier in the sentence. Add "in addition to dust" in between "MPOA" and "improves".

done

68. Line 550-552: The sentence beginning with "Model results are more consistent…" simply repeats what was just said and is unnecessary.

The sentence has been removed

69. Line 554: "contributor" should be plural, "contributor**s**"

done

70. Line 573: Can you expand on what you mean by "model performance"?

This sentence in the uncertainties section 3.3 is rephrased: "The effects of atmospheric ageing on ice-nucleating activity are probably a minor source of error in this study, in comparison to the introduced bias from the employed INP parameterizations and model performance in simulated INP-relevant particles"

The evaluation of simulated INP-relevant particles is now a full section 3.1.2.

71. Line 580: add ", but not $INP_{PBAP}$" after "are included".

done

72. Line 591-594: Did you show a relationship between PBAP and "regional weather patterns" somewhere? Or one between MPOA and "regions of high sea spray and phytoplankton activity"?

No, we did not show such correlations but the parameterizations used for the sources depend on meteorological factors for PBAP and on sea-spray emissions and chlorophyll-a for MPOA. The sentences have been rephrased for accuracy.

"Simulated concentrations of INP from terrestrial bioaerosol vary with season and meteorological factors that affect both the source of PBAP and its ability to act as INP. ….. Simulated INP from marine bioaerosol is found primarily over oceans and coastal areas and dominates between 40°-90°S (Southern Ocean), with high concentrations in regions of high sea spray and phytoplankton activity that influence the source of MPOA."

73. Line 599: remove "at" between "mainly" and "low altitudes".

done

74. Line 600-601: This paragraph would flow better if you remove the first sentence beginning with "Therefore, we propose…" and combine the rest of the paragraph with the very short one in lines 595-599.

done

75. Line 602: replace "could" with "can likely", since you do not show any evidence for this, it is your speculation.

done

76. Line 611: Does "sulfate acids" mean "sulfuric acid"?

corrected

**Figure Notes:**

1. Are all the data in the figures (except for Fig. 6) from 2015? What is the annual variability between the concentrations and percentages shown, if so?

All data presented are multi-year results. In the new Figure 8 we now show the coefficient of variation (Figure 8b) calculated as the standard deviation of the annual mean columns to the multiyear annual mean and is expressed in percent as an indicator of this variability, which locally can exceed 150%, maximizing in the tropical and polar regions.

2. Consider removing Fig. 1, since you barely mention it in the discussion. And/or, add some explanation to the main text to describe the different INP concentration metrics shown in the upper right

The figure is now discussed in the introduction

3. Please clarify which INP concentration metric you are plotting in every figure. See Major Comment #2.

done

4. I'm not sure Fig. 4 adds much to the discussion. Could you instead do the seasonal pie charts currently in Fig. 4 for Fig. 3? Between Fig. 2 and 3, the main conclusions from Fig. 4 have already been discussed.

Seasonal pies and Figure 4 have been removed. Figure 3 has been replaced by the new figure 9.

5. Separating the pie charts in Fig. 3 and 4 into only NH and SH are a bit oversimplified if you want to discuss dominant INP sources/types. As you mentioned, the main sources of PBAP are equatorial. The Southern Ocean and Arctic are very small sources of INPs overall, but also have very different clouds (height, temperatures, SLW) than in the tropics and mid-latitudes. Perhaps having 3 categories, one from 60 °S to 60 °N, plus the Arctic (>60 °N) and Southern Ocean/Antarctic (<60 °S) would make the most sense in terms of INP sources and also cloud regimes.

New figure 8 is drawn following these recommendations.

6. Caption for Fig. 6:
    1. Line 1075: add "simulated" between "accounting for" and "mineral dust"
    2. Line 1076-1078: the dashed lines are hard to see, perhaps replace with "shaded region" in the text, since you also shade the $Pt_1$ and $Pt_{1.5}$ regions.
    3. $Pt_{1.5}$ appears to be missing from the legends
    4. Line 1079: the error bars are the uncertainty on the observed values and not the simulated ones? What kind of uncertainty measurement is plotted (standard error, 90% confidence interval, etc)?
    5. Line 1080: The 1 and 1.5 in $Pt_1$ and $Pt_{1.5}$ should be subscripted.
    6. Line 1081: Do the $Pt_1$ and $Pt_{1.5}$ values consider the error bar on the simulated value, or just the monthly mean when considering whether they agree with the observations?

Figure 10 caption (earlier Figure 6) now reads: Comparison of INP concentrations calculated at the temperature of the measurements against observations accounting for simulated mineral dust (a), MPOA (b), PBAP (c), mineral dust and MPOA (d), PBAP and mineral dust (e), and mineral dust, MPOA and PBAP (f). The dark grey dashed lines represent one order of magnitude difference between modelled and observed concentrations, and the light-grey dashed lines depict 1.5 orders of magnitude. The simulated values correspond to monthly mean concentrations, and the error bars correspond to the

standard deviation of the observed monthly mean INP values. The color bars show the corresponding instrument temperature of the measurement in Celsius (a-f). $Pt_1$ and $Pt_{1.5}$ are the percentages of data points reproduced by the model within an order of magnitude and 1.5 orders of magnitude, respectively. R is the correlation coefficient, which is calculated with the logarithm of the values. Diamonds correspond to measurements (Bigg, 1990, 1973; Yin et al., 2012) that are compared with the climatological monthly mean simulations. Circles indicate comparisons between temporally and spatially co-located observations and model results.

    7. Figure S1 caption: Did you mean to reference Fig. 6 instead of Fig. 4?

8. The colorbar scale on Fig. S3 may be a little too large. It's difficult to tell that dust is significantly more prevalent than PBAP.

We thank the reviewer for the careful reading and the comments. Old figures have been redrawn for better quality and colors, new figures have been added to address the reviewer's comments and suitable for the restructured discussion of the manuscript.

---

## Author Comment (AC2)

**REPLIES TO REVIEWER #1 COMMENTS**

We thank the reviewer for the positive and constructive comments and the time they have spent in carefully reading our manuscript. All comments have been considered in the revised version of the manuscript. Below, in black are the reviewer comments and in blue our replies.

Review of "Assessing the global contribution of marine, terrestrial bioaerosols, and desert dust to ice-nucleating particle concentrations" by Chatziparaschos and co-authors in ACPD.

Summary:

In this study the authors use a global chemical-transport model driven by reanalysis meteorology to simulate the global distribution of ice nucleating particles (INPs) over a period of 6/7 years. The authors build upon the previous study of Chatziparaschos et al. (2023) who considered INPs from quartz- and K-feldspar-containing mineral dust particles. In this study the authors include two new INP types (collectively referred to as bioaerosols): primary marine organic aerosol (PMOA) associated with sea spray aerosol, and primary biological aerosol particles (PBAPs). The PBAPs considered in this study include terrestrial bacteria and fungal spores. Though INP sources of PBAPs can also include pollen, the authors choose to not represent this class due to uncertainties with its treatment, and a likelihood of low concentrations when compared to the other classes. The authors have put effort into sufficiently representing the emissions of the bioaerosols and their subsequent evolution in the atmosphere. Methods for representing the ice-nucleating ability of each INP species are based on existing parameterizations from literature. The model simulation is run from 2009 to 2016 to obtain temporally and spatially collocated INP concentrations for comparison with a global database of INP measurements. The authors compare the global (and seasonal) distributions of simulated INP from the three INP sources, highlighting locations and seasons in which each displays particular importance or sensitivity. Dust is particularly important in the northern hemisphere, especially at higher altitudes, and PMOA is important in the lower atmosphere of the southern hemisphere towards the polar region. This is consistent with other studies. The INP sourced from PBAPs shows particular importance at warmer freezing temperatures (when compared to dust) and across much of the lower latitudes where there is considerable (seasonally dependent) productivity from vegetation. Finally, the authors compare the simulated INP distributions with a global dataset of observations. The authors report that the model is most representative when dust and PMOA are considered the only sources of INP, whereas the addition of PBAPs to either dust or dust+PMOA acts to worsen the comparison. Reasons for this are discussed, including the potential for an over-estimation in the sensitivity of PBAPs to act as INPs. The authors conclude from this that although dust and PMOA produces the best representation for INPs in the climate system (as a whole), the inclusion of PBAPs is still recommended given their likely strong spatial and seasonal variability. I thoroughly enjoyed reading this and believe it will be a valuable addition for the community, but feel it needs restructuring into a more logical and meaningful sequence. I also believe a small amount of extra analysis in the simulated-vs-observed comparisons could be beneficial in drawing out the impact of PBAPs at warmer temperatures (which is where they display the most importance). I therefore recommend minor revisions before being accepted for publication. I have provided more details below.

We have restructured the manuscript, as also suggested by the other reviewer, and improved the discussion of model evaluation and uncertainties of our simulations and used this information for a thorough discussion of the results of the study. Our point-by-point replies to the comments follow in blue:

Major comments:

I have three primary areas of concern.

**1. To trust the simulated INP concentrations the reader must believe that the model is able to capture the precursor aerosol species, which in this case includes dust, sea spray / PMOA, and PBAPs. The evaluation of dust and PMOA is referred to as having occurred, but the reader is redirected to previous studies. There is no evaluation of PBAP concentrations, and no discussion as to whether the simulated concentrations are appropriate. There may be no observations to compare to, and in this case should be stated as such. I strongly recommend that the authors include a sentence or two for dust and PMOA that provides a quantitative summary of what the evaluations show. What is the RMSE for the comparison to observations? Does it display any particular bias? Did the evaluation show the model is representative of the distribution and seasonal cycle? I also recommend the authors include an evaluation (or reasons for its omission) of the simulated PBAP concentrations/distribution.**

We did not present evaluation of the INP-relevant aerosol species because these were done in earlier publications (Vignati et al., Atmos Environ. 2009; Myriokefalitakis et al., Adv. Meteorol., 2010, 1–16, doi.org/10.1155/2010/939171, 2010; Biogeosciences,13(24),6519-6543, doi.org/10.5194/bg-13-6519-2016; Chatziparaschos et al., Atmos. Chem. Phys., 23, 1785–1801, doi.org/10.5194/acp-23-1785-2023, 2023). However, following the reviewer's suggestion, a new section 3.1 on the simulated distributions of INP-relevant aerosols (3.1.1) and their evaluation (3.1.2) is added in the revised version of the manuscript. This section contains 3 new figures:

[Figure]

**Figure 2:** Simulated multi-annual surface mean (left column) and zonal mean (right column) of (a) mineral dust (sum of feldspar and quartz) mass concentrations (µg m⁻³), (b) primary marine organic aerosol mass concentrations (µg m⁻³), (c) bacteria number concentrations (m⁻³), (d) fungal spore number concentrations (m⁻³)

[Figure]

**Figure 3:** Monthly averaged concentrations of MPOA in ng-C m$^{-3}$ at (a) Mace Head, (b) Amsterdam Island in ng-C m$^{-3}$. Observations from Rinaldi et al. (2010) and Sciare et al. (2009), respectively.

[Figure]

**Figure 4:** Simulated bioaerosol concentrations compared to observations. The observations are averages over different time periods, and the simulated data are taken for the corresponding months at the grid boxes containing the measurement site. (a). Bacteria concentrations; observations include concentrations listed in the supplement of Burrows et al. (2009) and Genitsaris et al. (2017). (b) Fungal spore concentrations, actively wet discharged ascospores (AAS) and basidiospores (ABS) compared to the simulated FNG. In the model, AAS and ABS are assumed to contribute one-third each to total fungal

spores. The observations are from the references listed in Elbert et al (2007) and Bauer et al. (2008). (c) Observations of fluorescent PBAP (FBAP) compared with model simulations of PBAP (for this comparison calculated as the sum of BCT, FNG and pollen). Observations are from the compilation of long-term measurements of atmospheric fluorescent bioaerosol number concentrations listed in Petersson Sjögren et al. (2023).

As well as two in the supplementary material:

[Figure]

Figure S2: Simulated concentrations of MPOA compared with Mace Head (green) and Amsterdam Island (violet) measurements.

[Figure]

Figure S3: Seasonal variation of simulated number concentrations of PBAP compared to long-term (>4 weeks) FBAP observations compiled by Petersson Sjögren et al. (2023). Continuous lines are observations and dashed lines simulated number concentrations.

This new Section 3.1 concludes that "In summary, the evaluation of INP-relevant aerosols simulations shows that the model underpredicts MPOA (by about 30%; 3% - 50%) and overpredicts dust surface concentrations and deposition (by about 60%). PBAP concentrations are also slightly overpredicted (by about 20%), with bacteria being significantly underpredicted and fungal spores moderately overpredicted. The model captures the seasonal variation of observed PBAP, while simulated MPOA in the SH shows no significant variation across the seasons. These model features have to be taken into consideration when discussing the results of the INP simulations, since uncertainties in the INP simulations include uncertainties in both the INP parametrization and INP-relevant aerosol simulated concentrations and are affecting the importance of various INP types. "

**2. This is my primary concern. The current structure of the results section is: Figures 2 – 4 present the relative importance of each INP species; Figure 5 establishes the global distribution of INPs in the model; and Figure 6 validates the INP model. At the moment this is backwards. The logical order is:**

1. Here is our simulated INP model in its complete form (Figure 5; simulated INP distribution presented)
2. This is how it compares to the observations (Figure 6; INP model evaluated / validated)
3. Now we dig deeper into the relative importance of each species (Figures 2 – 4)

In this order the reader can then appreciate from the outset that the inclusion of PBAPs does not, in fact, tend to improve the representation of the global dataset of measured INP concentrations. This then has implications for the relative importance of PBAPs as is concluded from Figures 2 – 4. If, as suggested by the INP model evaluation, PBAP concentrations are too high or too ice-active, then this may suggest the 'relative-importance' analysis is biased towards INPs from PBAPs. The authors do a fantastic job at explaining the possible reasons why PBAPs do not improve the evaluation and I do not think that this means the relative-importance analysis needs to be removed. I think it is still entirely valid, but it should be highlighted that there is an associated uncertainty. The restructuring also applies to the abstract and conclusions section. I agree with the author's conclusions but feel they should be re-ordered for clarity.

We have restructured the presentation of the results as suggested by the reviewer. Section 3 of the results now starts with the presentation of INP-relevant aerosol distributions (new Figures 2-4 see our reply to comment #1) and their evaluation focusing on PBAP, the evaluation of which was not shown earlier (new section 3.1- *INP-relevant aerosols*) (earlier point of the reviewer; new figures 3 and 4). Then, in section 3.2 the simulated global distributions are presented and discussed in 3 sub-sections

section 3.2.1- *INP global distributions*

section 3.2.2- *Contribution of INP types to the total INP* where the relative contributions of each INP type to the total INP as deduced by the model are shown

section 3.2.3 -*Comparison of INP predictions with observations*

After this section we added a new section on:

"*3.3 Sources of uncertainties and implications for our results.*" that collects the discussions on the uncertainties that in the initially submitted version were in different parts of the document. In such a way we avoid repetitions and we present the results of our study together with their confidence levels.

The abstract and the conclusions have been accordingly rephrased.

**3. In-line with comment #2 above I believe the INP model evaluation (Figure 5) would benefit from additional analysis. It is clear that PBAPs are active at warmer temperatures than dust. Therefore, if the INP model evaluation of dust+PMOA+PBAPs is made in binned temperature regimes does this show**

better performance (when compared to dust+PMOA only) in the warmer temperature regimes? If so, this would strengthen the hypothesis that PBAPs are indeed important at warm temperatures. Also, the INP measurement database doesn't show particularly great global coverage. Given that the strongest signal from PBAPs is likely in low latitudes / boreal regions during the summer months, could the authors subset the comparison to these particular regions and seasons? This would provide a good basis for testing the importance of PBAPs.

We thank the reviewer for this comment. We have performed the additional analysis as suggested. We have added a new figure 9 an associate discussion in section 3.2.2 that further illustrates the different components of INP depending on atmospheric temperature, Figure 9 presents the percent contributions of INP types across three temperature ranges: [-40°C, -30°C], [-30°C, -20°C], and [-20°C, -10°C]. That figure highlights the spatial variability of the contribution of the studied species to the total simulated INP as a function of temperature. To plot this figure, all model results for which modelled temperature falls into the respective temperature range are selected. The percent contributions of each INP type to the total INP in each of the model grids and instances are calculated and then averaged per model grid, independent of the altitude.

We have also produced Figure S6 with the deviation of INP model results from INP observations per temperature bin that has been added in the supplement and supports the conclusion on the overestimate of INP by PBAP in or model for the warmer temperatures. This figure is now discussed in section 2.3.3.

[Figure]

Figure S1: Order of magnitude / Logarithm of the deviation of simulated concentrations of INP types from total INP observations, as a function of binned temperatures. INP_Dust in yellow, INP_PBAP in green, INP_MPOA in blue.

[Figure]

Figure 9: Multi-annual average percentage contributions of INP sources to the total INP column (surface up to 300 hPa): INP from mineral dust (left column), INP from MPOA (middle column), and INP from PBAP (right column), categorized by modelled temperature ranges: (top row) -40°C to -30°C, (middle row) -30°C to -20°C, and (bottom row) -20°C to -10°C.

We have also added a new Supplementary Figure S7 that shows simulated INP comparisons with observations per latitudinal zonal band and per INP type. This figure is discussed in section 2.3.2 and supports the regional importance of PBAP as INP.

"However, as seen in Figure S7, INP$_{PBAP}$ compares well with INP observations in the NH and the extratropical SH (Figure S7 top and mid-rows). Therefore, in no case our results should diminish the role of INP derived from PBAP, especially in regions where PBAP is more abundant or ecosystems are diverse, such as the Amazon, affecting INP concentrations locally (Prenni et al., 2009). Most of the INP at warmer temperatures than -16°C are bioaerosols, suggesting that without improved representations of the sources and ice-nucleating activities of biological INP, models will struggle to simulate total INP concentrations at warmer temperatures and the resulting MPC"

[Figure]

Figure S2: Comparison of INP concentrations calculated at the temperature of the measurements against observations accounting for mineral dust (orange), MPOA (blue), PBAP (green) separated in high, middle and low latitudes.

Minor comments:

Throughout: degree symbols are sometimes not superscript.

L30. As discussed in comment #2 the INP model evaluation should precede the more detailed analysis. Also, given the uncertainties I recommend changing "INP originating from PBAP…" to "The model suggests INP originating from PBAP…"

Modified as suggested

L50. MPC not yet defined in the manuscript.

It was done in the abstract, now also defined in the introduction

L54. "As a result, they significantly impact precipitation rates and radiative energy balance on both regional and global scales" Please include references to support this statement.

Hofer et al., Nature Communications Earth and Environment, 2024 https://doi.org/10.1038/s43247-024-01524-2 has been added

L56. Collection/collisional processes are also an ice-liquid interaction in MPCs. Can you expand to include other processes?

The following text has been added "In MPC, the most important ice formation process is immersion freezing, where an ice-nucleating particle (INP) becomes immersed in a supercooled droplet and initiates freezing, typically occurring between -5°C and -35°C (Pruppacher et al., 1998; Kanji et al., 2017 and references therein). Contact freezing happens when an INP collides with a supercooled droplet, triggering freezing on contact, often in turbulent conditions. This process is defined as separate from immersion freezing because of empirical evidence that some INP are more effective in this mode than when immersed in liquid (Shaw et al., 2005). Condensation freezing occurs when water vapor condenses and freezes simultaneously upon contact with an INP under specific supersaturation conditions. This process is less frequent compared to immersion freezing in atmospheric clouds (Vali et al., 2015). Deposition nucleation, where water vapor directly deposits as ice onto an INP without passing through the liquid phase, is more relevant at much colder temperatures (below -20°C) and is less significant in the typical temperature range of mixed-phase clouds (Kanji et al., 2017 and references therein) but it might still be important for cirrus clouds (Cziczo et al., 2013)"

Cziczo, D. J., Froyd, K. D., Hoose, C., Jensen, E. J., Diao, M., Zondlo, M. A., Smith, J. B., Twohy, C. H., and Murphy, D. M.: Clarifying the Dominant Sources and Mechanisms of Cirrus Cloud Formation, Science (80-. )., 340, 1320–1324, https://doi.org/10.1126/science.1234145, 2013.

Kanji, Z. A., Ladino Moreno, L. A., Wex, H., Boose, Y., Burkert-Kohn, M., Cziczo, D. J., & Krämer, M. (2017). Overview of Ice Nucleating Particles. *Atmospheric Chemistry and Physics, 17*, 12177–12223. https://doi.org/10.5194/acp-17-12177-2017

Pruppacher, H. R., Klett, J. D., and Wang, P. K.: Microphysics of Clouds and Precipitation, Aerosol Sci. Technol., 28, 381–382, https://doi.org/10.1080/02786829808965531, 1998.

Vali, G., DeMott, P. J., Möhler, O., and Whale, T. F.: Technical Note: A proposal for ice nucleation terminology, Atmos. Chem. Phys., 15, 10263–10270, https://doi.org/10.5194/acp-15-10263-2015, 2015.

L71. "..via the WBF process and multiply via SIP… affecting precipitation ranges, cloud properties, and albedo". Please include references to support his statement.

'precipitation' has been removed and the statement is supported by the existing references (Georgakaki et al., 2022; Korolev and Leisner, 2020).

L80. Could you expand to describe the 'seeder-feeder' mechanism?

We extend that sentence to explain briefly the seeder-feeder mechanism : Text added " where ice crystals from an upper cloud (seeder) fall into a lower-lying liquid cloud (feeder) and grow there by riming or vapor deposition via the WBF process"

L85. "In the absence of a well-established theory for heterogeneous ice nucleation…". What about classical nucleation theory?

'At temperatures warner than -35°C' is added. The sentence now reads" In the absence of a well-established theory for heterogeneous ice nucleation at temperatures warmer than -35°C, INP prediction typically relies on empirical parameterizations that are subject to considerable uncertainty and challenges.

L127. "…with INP concentrations estimated at approximately 1 to 2 per litre". Would be useful to have some context - how does this compare to dust?

This sentence now reads: 'Only a small fraction of biological material can trigger ice nucleation (Huang et al., 2021), with INP concentrations estimated at approximately 1 to 2 per liter (Pöschl et al., 2010). These values are, however, about 2 order of magnitude lower than the highest observed INP number concentrations from dust (Murray et al., 2012).

L134. SIP yet to be defined.

Done (secondary ice particles)

L137. "..warm parts of the mid-latitude clouds..". I do not understand what this is referring to. Please can you rewrite for clarity?

This has been rephrased and now reads: ' PBAP contribute to droplet freezing rates of clouds in the warmer parts of mid-latitude atmosphere'

L137. "..with about 1x10^-5 % of the global average ice nucleation rates.." this is also a confusing statement. Please can you rewrite for clarity?

The entire sentence has been rephrased as follows: Global simulations including biological particles have estimated that PBAP contributes to droplet freezing rates of clouds in the warmer parts of mid-latitude atmosphere, accounting for $1 \times 10^{-5}$ % of the global average ice nucleation rates, with an upper-most estimate of 0.6% (Hoose et al., 2010).

L139. "..altitudes between 400 and 600 hPa..". Please use consistent quantities/units.

The sentence has been rephrased to : "Spracklen and Heald (2014) suggest that PBAP can dominate immersion freezing rates at altitudes corresponding to pressures between 400 and 600 hPa,"

L181. "bioaerosols". This term hasn't been defined yet.

This is now defined in the beginning of the 7th paragraph of the introduction which reads: 'While it is well established that dust particles are the most abundant source of INP at temperatures below -20ºC, it is equally well established that bioaerosols, i.e. atmospheric particles of biological origin, such as fungal spores, bacteria, pollen, and plant debris, are highly efficient ice nucleators at warm temperatures (Huang et al., 2021; Cornwell et al., 2022) and may have a substantial effect on ice crystal formation (Prenni et al., 2009).'

L191. The model is very coarse, both horizontally and vertically. Do the authors think this will have implications for the emission and dispersion of aerosols and modelled INP concentrations?

Indeed, the model has a relatively coarse resolution which enables to perform multiple multiyear global simulations on the existing computer facilities. The spatial resolution of the model implies that the simulated INP concentrations are more representative of a climatology of the background atmosphere than of the INP- related aerosol source regions. This is the reason we have performed multiyear simulations and we analyze the average of them. We have added the following sentence: "Therefore, when performing multiyear simulations, the model provides a climatological view of the troposphere."

L205. Please expand this sentence to briefly say what drives the dust emission sensitivity – i.e., wind speeds, surface moisture, etc

This sentence now reads as follows: "Desert dust emissions are calculated online as described in Tegen et al. (2002) and implemented as in Van Noije et al. (2014) accounting for particle size distribution and

based on vegetation type and cover, dust source areas, snow cover, soil moisture, and surface wind speed."

L215. As per comment #1 please expand and provide a short quantitative summary of the evaluation.

Following one of the major comments of the reviewer, the evaluation of the INP-relevant aerosols is now part of a new section 3.1.2. See our reply to major comment #1

L245. "FNG are assumed to be … 3um diameter with 1000 kgm-3 density". Is this appropriate? Do the authors have a sense of the uncertainty of this and its variability across the globe?

Indeed, this number has a large uncertainty but it is commonly used for fungal spores modeling. We consider that the sentence as was written points this uncertainty : "Bioaerosol sizes range from fine to coarse, but since their shapes are not accurately known, for the present work, FNG are assumed to be monodisperse spherical particles of 3 μm diameter with 1000 kg m$^{-3}$ density (Hummel et al., 2015)." So, no change has been done.

Did the authors evaluate the simulated bacteria + fungal spore concentrations against observations? Is this a key observational dataset we are missing?

This is now included in the revised manuscript in the new Section 3.1.2, see our reply to major comment #1

L278. As per comment #1 please expand and provide a short quantitative summary of the evaluation.

Evaluation is now in an entirely new section 3.1. It first summarizes finding from published evaluations of dust and of MPOA and presents the evaluation of MPOA and PBAP simulations. (see reply to comment #1)

L298. Please include some statistics that summarise the evaluation. This evaluation is actually shown in Figure 6a so could be referenced.

Statistics on this evaluation together with discussion are provided in section 3.2.2 where the INP simulations are evaluated for each INP type separately and combined.

"Considering dust minerals as the sole INP types leads to underestimation against observations (Fig. 10a; Pt$_1$ about 24% and R=0.84). Even if dust is the most abundant aerosol in the atmosphere, the simulated dust-derived INP cannot predict the observed INP, especially at high temperatures (>-15ºC) and relatively low INP levels, since mineral dust particles likely become ice-active only at low temperatures (Chatziparaschos et al., 2023; Cornwell et al., 2023). Notably, the results show that there is some overestimation for colder temperatures, which may be partly related to the overestimation of dust in the model. Findings in Figure 10a are largely in agreement with the literature and have been thoroughly discussed in Chatziparaschos et al. (2023)."

[Figure]

**Figure 10:** Comparison of INP concentrations calculated at the temperature of the measurements against observations accounting for simulated mineral dust (a), MPOA (b), PBAP (c), mineral dust and MPOA (d), PBAP and mineral dust (e), and mineral dust, MPOA and PBAP (f). The dark grey dashed lines represent one order of magnitude difference between modelled and observed concentrations, and the light-grey dashed lines depict 1.5 orders of magnitude. The simulated values correspond to monthly mean concentrations, and the error bars correspond to the error of the observed monthly mean INP values. The color bars show the corresponding instrument temperature of the measurement in Celsius (a-f). $Pt_1$ and $Pt_{1.5}$ are the percentages of data points reproduced by the model within an order of magnitude and 1.5 orders of magnitude, respectively. R is the correlation coefficient, which is calculated with the logarithm of the values. Diamonds correspond to measurements (Bigg, 1990, 1973; Yin et al., 2012) that are compared with the climatological monthly mean simulations. Circles indicate comparisons between temporally and spatially co-located observations and model results.

L370. Missing minus sign.

corrected

L372. Specific to this line but more general. How relevant are the tropical and subtropical atmospheres for MPCs? Are they commonly observed here or more prevalent in the higher latitudes?

The following text has been added: "Note that MPC were found to occur more frequently in the convective towers in the tropics than at the mid-latitudes and the Arctic (Costa et al., 2017)."

Costa, A., Meyer, J., Afchine, A., Luebke, A., Günther, G., Dorsey, J. R., Gallagher, M. W., Ehrlich, A., Wendisch, M., Baumgardner, D., Wex, H., and Krämer, M.: Classification of Arctic, midlatitude and tropical clouds in the mixed-phase temperature regime, Atmos. Chem. Phys., 17, 12219–12238, https://doi.org/10.5194/acp-17-12219-2017, 2017.

L404. I suggest adding a value that signifies a threshold for "high pressure levels".

We have rephrased this part of the discussion because it is the temperature that affects the INP activity of the different types of INP. Now in section 3.2.2 we provide such information:

"INP$_{PBAP}$ is the primary type of INP between −12°C and −20°C…. INP$_{PBAP}$ contribution to the total INP is more than 80% between -12°C and -16°C. … At temperatures colder than -20°C, nonbiological

aerosol particles such as mineral dust are effective INP (Murray et al., 2012; Si et al., 2019) and dominate the INP population (Fig. 7a)."j

L497. "Even if dust is the most abundant… cannot predict the observed INP, especially at high temperatures". Is this consistent with other studies?

Yes. In section 3.2.3 we now state: "Findings in Figure 10a are largely in agreement with the literature (e.g. Vergara-Temprado et al., 2018) and have been thoroughly discussed in Chatziparaschos et al. (2023). "

L502/3. "…largely in agreement with literature and have been thoroughly discussed in ..". This shouldn't be redirected to a different study. Please summarise the discussion from Chatziparaschos et al. 2023.

This paragraph now reads:

"Considering dust minerals Considering dust minerals as the sole INP types leads to underestimation against observations (Fig. 10a; Pt$_1$ about 24% and R=0.84). Even if dust is the most abundant aerosol in the atmosphere, the simulated dust-derived INP cannot predict the observed INP, especially at high temperatures (>-15°C) and relatively low INP levels, since mineral dust particles likely become ice-active only at low temperatures (Chatziparaschos et al., 2023; Cornwell et al., 2023). Notably, the results show that there is some overestimation for colder temperatures, which may be partly related to the overestimation of dust in the model. Findings in Figure 10a are largely in agreement with the literature (e.g. Vergara-Temprado et al., 2018) and our earlier study by Chatziparaschos et al. (2023). "

The evaluation of dust simulations by Chatziparaschos et al. (2023) is summarized in the new section 3.1.2.

L505. "… increasing the predictability of the model to 70%..". Recommend including "increasing the predictability of the model FROM XX% FOR DUST to 70%..."

This sentence now reads: 'INP$_{MPOA}$ alone significantly improves the prediction of the observed INP at high temperatures, increasing the predictability of the model from 24% for dust alone to 77% for MPOA alone (Pt1, Figs. 10a and 10b).'

L506. I think this is where the additional analysis suggested in comment #3 would fit – and enhance the rigor of the evaluation.

Done – see our reply to major comment #3

L511/2. "The model tends to slightly overpredict INP only in the temperature range around -25C". Could the parameterisation exhibit an incorrect temperature-dependence?

We have checked – the temperature dependences are correct.

L532. Have you tried looking at only-fungal-spore INPs vs only-bacteria INPs? Would this help identify which sub-species of bioaerosol is driving the discrepancy?

In the new section 3.1.2 we now evaluate separately bacteria and fungal spores. We conclude: "PBAP concentrations are also slightly overpredicted (by about 20%), with bacteria significantly underpredicted and fungal spores moderately overpredicted." See figures in our reply to major comment #1

L600 onwards. In this paragraph the authors list a number of potential sources of uncertainty and reasons for model bias. What is required in order to constrain or address these? More laboratory measurements? More in-situ observations? More locations? Additional techniques to establish particle composition at sampling site? It would be good to identify what is currently lacking and is needed in future studies/campaigns.

Missing data availability statement.

We have added the following text at the end of the conclusion: "Due to the large variability of INP with space and time, effort must be put into acquiring more data on INP ambient levels as well as the ice nucleating properties of individual aerosol types as well as how these change with atmospheric ageing. These are needed to build regionally and globally representative datasets and reduce the uncertainty in the parameterizations of their sources in numerical models. Tackling these research priorities is essential for developing a more comprehensive understanding of the atmospheric variability of INP and their impacts in different cloud regimes and climate "

Figure 1. Reference to different methods of calculating INP concentration depending on convective state. This is not discussed in the manuscript, and therefore is quite confusing. Please include an explanation of this in the text. Also consider removing this figure as it does not add much to the manuscript and is only briefly referred to.

We provide Figure 1 as a summary of ice nuclei formation in the atmosphere. Therefore, we kept it and added appropriate explanation in the text: "Figure 1 presents a comprehensive schematic illustration of INP formation derived from mineral dust, MPOA and PBAP. It includes emission processes, nucleation mechanisms, and the aerosol indirect effect, indicating the role of aerosols in cloud interactions and their impact on climate. ……

For this purpose, INP concentrations are presented in this study in two ways as $[INP]_{ambient}$ and as $[INP]_T$: $[INP]_{ambient}$ is calculated at ambient model temperature relevant to non-deep convective mixed-phase clouds using the ambient (model) temperature at each simulated level. $[INP]_T$ is calculated at a fixed temperature relevant to vertically extended clouds as deep-convective systems (Figure 1). $[INP]_{ambient}$ and $[INP]_T$ metrics can vary significantly."

later in Section 2.4 we also state:

"As outlined in the introduction $[INP]_T$ and $[INP]_{ambient}$ are determined by the simulated particle concentration and for $[INP]_T$ a specific temperature (T), while for $[INP]_{ambient}$ the model's ambient temperature is used. $[INP]_T$ is the appropriate metric when comparing modelled INP concentrations to observations, as measurements are typically conducted by exposing particles to specific, controlled temperatures within the instruments. Thus, for the model evaluation, $[INP]_T$ concentrations are used that are calculated at the temperature at which the measurements were performed."

Figure 2. Viewing this figure feels like an eye test. Please increase the font size of all elements. Also, this would greatly benefit from distinct colormaps for the two rows. One for concentrations, the other for percentage contributions. The same comments apply to Figures S2 and S3 in the supporting information.

All figures are redrawn

---

## Referee Report (RR1)

The authors have sufficiently dealt with all of the comments that I raised, and I believe the manuscript is much improved as a result. I recommend the paper is published following a final question for the authors. Why was new data added to Figure 10? There is substantially more data, which appears to have come from the addition of the Bigg datasets. Why was this added and why was it not included in the previous version? There is no explanation in the response.

---

## Referee Report (RR2)

**Summary:**

This study by Chatziparaschos et al. adds dust, terrestrial bioaerosols, and marine organic ice nucleating particles (INPs) to a global chemical-transport model (TM4-ECPL) and examines the contribution of each INP type at different latitudes, altitudes, and temperatures over simulations from 2009-2016. Dust INPs are found to dominate at higher altitudes (<-20°C) across all latitudes and seasons. PBAP are the most important INP species at warm temperatures (>-16 °C), especially in terrestrial equatorial and mid-latitude regions, although the $INP_{PBAP}$ parameterization appears to be overactive (Figure S6), so the exact contribution requires further study. Marine organics dominate the Southern Ocean at low altitudes in all seasons but have minimal contributions in the Northern Hemisphere. The results presented are in broad agreement with previous observational and modeling studies and provides a valuable addition to the INP literature.

The updated figures and restructuring of the discussion greatly enhance the manuscript and increase clarity for the reader. The manuscript now also includes a more detailed and nuanced discussion of potential model biases, which is appreciated and helps put the current study in context and provides suggestions for future modeling and observational studies. I have only one broader comment and some suggested minor (mainly grammatical errors and typos) edits.

**Major Comments:**

I still don't fully understand why 600 hPa was chosen as an example pressure level for Fig. 5. Although it is true that most INP measurements are made at a different temperature inside the instrument than the ambient temperature ($[INP]_T$ vs $[INP]_{ambient}$), the aerosol being measured is still more representative of the boundary layer than the free troposphere, just at a different temperature than ambient ($[INP]_T$). I agree that -20 °C is a reasonable temperature to choose to be representative of MPC glaciation, but at high latitudes -20 °C is reached within or very near to the boundary layer and the MPCs are typically not located in the free troposphere. Perhaps clarify that the pressure level chosen is representative of MPCs at low latitudes only, and not broadly representative of MPCs across all latitudes (lines 496-504). Much of the rest of the paper, and all the measurements being compared to in other figures are in the boundary layer, so it seems an odd choice to focus on the spatial distribution at a relatively high altitude, unless there is another reason I have missed.

The discussion of Fig. 5 is thorough, and the addition of the circled areas is helpful to follow the analyses. However, the mentions of "continental outflow" and "downwind of source areas", etc would make more sense for a lower altitude than 600 hPa, either near the surface, or at least within the boundary layer. Extensive long range transport and mixing is expected for aerosols above the boundary layer, as well as a longer time since emission. There is no problem with discussing the results at this pressure/altitude, but trying to connect the results at 600 hPa to surface emissions perhaps needs to be more nuanced. This is particularly true since the high and low latitudes are discussed together, but the fixed altitude used does not account for the changes in boundary layer height (temperature) or vertical mixing that occur between the equatorial regions and poles.

**Minor Comments:**

1. Line 35-37: Suggest re-arranging the sentence as follows so it is easier to read: "MPOA-derived INP ($INP_{MPOA}$) prevails in the SH **at low altitudes**, particularly at subpolar and polar latitudes for temperatures…"
2. Line 40: Add "the" between "enhance" and "model's"
3. Line 48: Suggest replacing "homogeneity of mixing ice" with "homogeneity **in** mixing **of** ice"
4. Line 64: "forms" should be "formed"
5. Line 66: Add "saturation" before "water vapor pressure"
6. Line 69: Remove "saturation" before "water vapor pressure" or replace with "**ambient** water vapor pressure are…". This and minor comment #5 appear to be a mix-up in line numbers for a comment in the original review.
7. Line 73: "while it can also occur in cold clouds" seems out of place in a paragraph focused on MPCs.
8. Line 75: The sentence is a bit confusing, suggest replacing "air – can activate temperatures" with "air can activate at temperatures".
9. Lines 94-97: Although it is true that INP prediction in models often uses empirical parameterizations, CNT (classical nucleation theory) is an alternative and can be used at temperatures above -35 °C. See Kanji et al. (2017) and references therein for a broad overview. A discussion of this topic is not required in this paper, and I suggest just removing the first half of the sentence and simply saying "INP prediction **in models** typically relies…"
10. Line 145: The acronym SIP typically refers to secondary ice **production**, not secondary ice particles as written in this line.
11. Line 149: What is meant by the "warmer parts of mid latitude atmosphere"?
12. Line 157: Replace "PBAP is critical contributor to INP." with "PBAP are a critical contributor to INP concentrations".
13. Line 182: Replace "this region" with "these regions"
14. Line 389: Suggest changing the title of Sec. 3.1.1 to "Global **aerosol** simulations" to separate them from the INP simulations presented later
15. Line 390: Add "multi-" before "annual" for clarity
16. Line 397: Suggest replacing "high latitudes" with "**mid-**high latitudes" since MPOA seems to peak ~50°.
17. Lines 410-412: This sentence ("This comparison suggested…") appears to have some typos and is hard to follow.
18. Line 423: Was "organic carbon derived from MPOA" meant to be "**MPOA** derived from **organic carbon**"?
19. Lines 431 and 441: Was "Austral Ocean" meant to be "**Southern** Ocean"?
20. Line 490: Suggest replacing "and are affecting the importance of various INP types" with "and affect the inferred importance of various INP types in the model".
21. Line 500: No parentheses needed around the sentence "(This metric is explained…)"
22. Line 511: Consider adding "outside of strong dust source regions" after "40° latitude", since it is clear that dust is a strong source in many terrestrial regions in the mid-latitudes.
23. Line 525: "isis" is a typo
24. Line 528: "shown in Figure 1) (Fig. 6a-d)" should be "shown in Figure 1) (Fig. 6a-**c**)"
25. Line 572: "mineral dust (Fig. 7d)" should be "mineral dust (Fig. 7**a**)"
26. Line 572: "seasons (Fig. S3)" should be "seasons (Fig. S**4**)"

27. Line 591-592: Consider removing the sentence "The impact of…in the NH." since it repeats the previous sentence.
28. Line 595: After "$INP_{MPOA}$", consider referencing Fig. S4c, S5c, so the reader knows what figure you are discussing.
29. Line 598: After "INP increases", consider referencing Fig. S4b, so the reader knows what figure you are discussing.
30. Line 657: I think the reference to "in summer (Fig. S2)" was meant to be "in summer (Fig. S4)"
31. Lines 689-691: Suggest clarifying the sentence as follows: "Since **only** a 20% overestimation of PBAP concentrations by the model can be deduced when compared to observations, an overestimation of the INP scheme seems to be the most plausible reason for the **large** $INP_{PBAP}$ overestimation."
32. Line 692: Does Fig. S6 compare the simulated INP types ($INP_D$, etc) against the total INP concentrations from observations? It does not appear that the observations have been separated by INP type. If not, suggest adding "total INP" between "INP types from" and "observations".
33. Lines 706-712: Same as #32, does Fig. S7 compare the simulated INP types against the total INP concentrations from observations? If so, the good agreement between simulated $INP_{PBAP}$ and total INP observations probably still indicates an overactive $INP_{PBAP}$ parameterization, rather than good agreement with observations (which likely contain multiple INP types). The rest of the paragraph is great, and the points about local variability and the importance of bioaerosols at warm temperatures are important to make.
34. Section 3.3: The paragraph beginning with "Our results suggest that…" seems like a more logical place to begin this section, since it is more general and not specific to $INP_{MPOA}$. I would move that up to be the first paragraph of Sec. 3.3. I suggest removing the current first sentence (lines 714-716, "Earth System Models (ESM) encounter…" and moving the rest of the first paragraph (lines 716-720) "McCluskey et al. (2018) have reported…model's underestimation of MPOA." down to line 740 and combining it with the paragraph starting "Furthermore, INP dust parameterizations can introduce…other INP types on cloud properties".
35. Line 728: "and used for global estimates" is unnecessary, since the rest of the sentence discusses their global applicability
36. Line 751: Was "**ice** nucleating ability" meant, rather than "dust nucleating ability"?
37. Lines 766-773: The paragraph "In summary, the comparison…account for $INP_{PBAP}$ in climate modeling." would fit better at the end of Sec. 3.2.3, or just remove since this information is covered in several places already.
38. Line 778: "singular description approach" should be "singular **hypothesis** approach"
39. Line 805: Suggest replacing "aerosol types as well as how" with "aerosol types **and** how"
40. Line 809: Suggest adding "on" between "cloud regimes and" and "climate"

**Main Text Figure Notes:**
1. Figure 3
   a. Caption: there are two mentions of the units (ng-C m$^{-3}$) and only one is necessary.
   b. Fig. 3 also seems blurry, consider exporting in a different image format or increasing the resolution.
2. Fig 6 caption

a. Square brackets are not needed around "[INP] number concentration"
   b. The reference to panels (a-c) is confusing in "The colours show (a-c) the INP number concentration." since the colors are the same in all panels (a-d) and not just (a-c).
3. Fig. 6 and 7 captions: Consider also specifying INP concentrations are only plotted if they exceed 0.01 m$^{-3}$, as in Fig. S4 and S5 captions.

**SI Notes:**
1. Figure S1
   a. Check the legend on this figure, for example, the "ISAC_CNR_2012_Antarctica" and "Yin_China_2012" data have the same symbol, and the Antarctic data doesn't appear to be plotted.
   b. Are all the points marked as "Bigg_1969-1989" from Bigg's papers, or are some from Welti et al. (2020) previously unpublished data (eg Tan1502, SHIPPO, etc)? I thought Bigg's measurements largely ended ~140°E south of Australia. Also a note that Bigg's Southern Ocean measurements are much higher (2-3 orders of magnitude) than all modern measurements (see e.g. McCluskey et al., 2018; Moore et al., 2024; Tatzelt et al., 2022) and may not provide the best comparison dataset for simulations in 2009-2016.
   c. The reference to "Figure 4" was probably meant to be for Figure 10, and perhaps also Fig. S6 and S7.
2. Figure S6: The x-axis is plotted as Kelvin, but the label reads °C and the discussion of Fig. S6 in the main text refers to °C also. Please update to have consistent units.
3. Table S1: The "Bigg_1969-1989" data that was added to Fig. S1 does not appear to also have been added to Table S1.

**References:**
Kanji, Z. A., Ladino, L. A., Wex, H., Boose, Y., Burkert-Kohn, M., Cziczo, D. J., & Krämer, M. (2017). Overview of Ice Nucleating Particles. *Meteorological Monographs*, *58*, 1.1-1.33. https://doi.org/10.1175/AMSMONOGRAPHS-D-16-0006.1
McCluskey, C. S., Hill, T. C. J., Humphries, R. S., Rauker, A. M., Moreau, S., Strutton, P. G., et al. (2018). Observations of Ice Nucleating Particles Over Southern Ocean Waters. *Geophysical Research Letters*, *45*(21), 11,989-11,997. https://doi.org/10.1029/2018GL079981
Moore, K. A., Hill, T. C. J., McCluskey, C. S., Twohy, C. H., Rainwater, B., Toohey, D. W., et al. (2024). Characterizing Ice Nucleating Particles Over the Southern Ocean Using Simultaneous Aircraft and Ship Observations. *Journal of Geophysical Research: Atmospheres*, *129*(2), e2023JD039543. https://doi.org/10.1029/2023JD039543
Tatzelt, C., Henning, S., Welti, A., Baccarini, A., Hartmann, M., Gysel-Beer, M., et al. (2022). Circum-Antarctic abundance and properties of CCN and INPs. *Atmospheric Chemistry and Physics*, *22*(14), 9721–9745. https://doi.org/10.5194/acp-22-9721-2022
Welti, A., Bigg, E. K., DeMott, P. J., Gong, X., Hartmann, M., Harvey, M., et al. (2020). Ship-based measurements of ice nuclei concentrations over the Arctic, Atlantic, Pacific and Southern oceans. *Atmospheric Chemistry and Physics*, *20*(23), 15191–15206. https://doi.org/10.5194/acp-20-15191-2020

---

## Referee Report (RR3)

**Comments:**

The authors have overall adequately addressed the Reviewer comments. There are, however, a few additional comments brought up by this last revision.

1.  In the response to the Reviewer #1 comment about the addition of the Bigg (1973, 1990) and Yin et al. (2012) datasets to Fig. 10, the authors indicated "*Following the other reviewer comment, we have added datasets even if they did not fall inside the simulated period.*" This is not what was suggested. The suggestion (from the original review) was to add different, publicly available INP observations to the comparison. And specifically, data that was within the simulated period, so a climatological comparison would not be necessary. Here is the original suggestion:
    "Oceanic INP measurements have been compiled in Welti et al. (2020), and many of them are from within the 2009-2016 period simulated here. Adding some of these to the evaluation would improve confidence in the simulated MPOA INP concentrations, in particular."
    These new datasets (ie TAN1502, ACAPEX, SHIPPO, NETCARE, CAPRICORN, etc from Welti et al. (2020), Table A1) were not added, but instead, a climatological comparison was made to the older Bigg (1973, 1990) and Yin et al. (2012) datasets.

    As noted by the authors in their response, the Fig. 10 caption very clearly indicates that some of the data used for comparison is based on a climatological average, instead of temporally matched measurements. However, the symbols in Fig. 10 are very small and the diamonds (climatological comparison) and circles (temporally matched comparison) are not easily distinguishable. Please choose more different symbol shapes and/or increase the symbol size to make them easier to distinguish.

2.  Thank you for re-drawing Figure S1 so the legend is easier to follow, and for clarifying that all the data are from Bigg (1973, 1990). Indeed, the large spatial coverage of Bigg's measurements is very valuable for these types of model comparisons. However, given that it is well known that they are about 2-3 orders of magnitude higher than all measurements made during or much closer to the period simulated here (ie McCluskey et al. (2018), Tatzelt et al. (2022), Moore et al. (2024)), this potential bias in the observations should be mentioned somewhere in the discussion. This is particularly true because the main metrics being used to assess the simulation accuracy are $Pt_1$ and $Pt_{1.5}$ (see comment #3), but if the observations are 2-3 orders of magnitude higher than they should be during this simulated period, these cannot be accurately calculated. A reasonable place to add this disclaimer would be in the paragraph discussing the $INP_{MPOA}$ predictions, Lines 681-687. This paragraph mentions uncertainties "*resulting from both the MPOA simulations and the $INP_{MPOA}$ parameterization*" and a brief mention of the uncertainties in the observations chosen for this comparison would fit in well.

3.  The discussion of Fig. 10 revolves around the $Pt_1$ and $Pt_{1.5}$ metrics defined by the authors in Sec. 2.4, plus correlation coefficients. These are both useful, however, neither captures any bias seen between the observations and simulations. Simulated overestimation/underestimation in INP concentration is discussed qualitatively in the text

(ie Lines 688-694) but if Modified Normalized Mean Bias, or a similar metric, could be listed in Fig. 10 in addition to the R value, $Pt_1$, and $Pt_{1.5}$, it would greatly enhance this assessment. For example, the $INP_{PBAP}$ simulation (Fig. 10c) has a fairly high correlation (R=0.79) and "predictability" ($Pt_1$=61%) but is clearly biased high at all relevant temperatures, and this isn't captured by the statistical measures used here.

4. The discussion of Fig. S7 (Lines 709-715) is this updated manuscript is limited and does not support the text. Given that the $INP_{PBAP}$ simulation is clearly overestimating the total INP concentration by itself (Fig. 10c, Fig S6), the updated sentence "*However, as seen in Figure S7, $INP_{PBAP}$ compares well with total INP observations in the NH and the extratropical SH (Figure S7 top and mid-rows), indicating an overactive $INP_{PBAP}$ parameterization. Therefore, in no case our results should diminish the role of INP derived from PBAP...*" does not support the point that PBAP are important. The current simulation results indicate that the $INP_{PBAP}$ concentrations predicted by the model are too high, but does not provide any information about how much too high- just that they're too high, even if PBAP was the only INP type present in the atmosphere (unlikely). So PBAP may not be important as INPs- you can't tell without an improved parameterization scheme. The new paragraph following this one (Lines 716-723) repeats the main conclusion of this paragraph- that $INP_{PBAP}$ are important in these simulations, which is accurate- so Lines 709-715 could be removed without losing any information. However, there is no other mention of Fig. S7, so the results from that figure should either be added to the rest of the discussion in Sec. 3.2.3 or the figure removed.

   I am not suggesting this as necessary for this manuscript, but a future article could do a sensitivity study where the $INP_{PBAP}$ parameterization is adjusted (for example, reduced by 10x, 50x, etc.), and the results examined again to see if more likely $INP_{PBAP}$ concentrations still show the findings of Fig. 5-7. This would be further enhanced, as I suggested previously, if the simulations of specific INP types ($INP_{PBAP}$, $INP_{MPOA}$, $INP_{Dust}$) were compared to observations where the dominant INP type was known and matched the simulations. For example- simulated $INP_{Dust}$ would be compared to observations where dust was known to be the dominant INP type present, instead of what is in Fig. 10, which compares each simulated INP type to the total INP observed.

**Minor Comments:**
1. Figure S2 caption has a typo: "MOA" is written, instead of "MPOA"
2. Figure S6 caption has a typo: The INP types are written as "$INP_{\_Dust}$", "$INP_{\_MPOA}$", etc, but should just be "$INP_{Dust}$", "$INP_{MPOA}$" to be consistent with the rest of the text.

**References:**
Bigg, E. K. (1973). Ice Nucleus Concentrations in Remote Areas. *Journal of the Atmospheric Sciences*, *30*(6), 1153–1157. https://doi.org/10.1175/1520-0469(1973)030<1153:INCIRA>2.0.CO;2

Bigg, E. K. (1990). Long-term trends in ice nucleus concentrations. *Atmospheric Research*, *25*(5), 409–415. https://doi.org/10.1016/0169-8095(90)90025-8

McCluskey, C. S., Hill, T. C. J., Humphries, R. S., Rauker, A. M., Moreau, S., Strutton, P. G., et al. (2018). Observations of Ice Nucleating Particles Over Southern Ocean Waters. *Geophysical Research Letters*, *45*(21), 11,989-11,997. https://doi.org/10.1029/2018GL079981

Moore, K. A., Hill, T. C. J., McCluskey, C. S., Twohy, C. H., Rainwater, B., Toohey, D. W., et al. (2024). Characterizing Ice Nucleating Particles Over the Southern Ocean Using Simultaneous Aircraft and Ship Observations. *Journal of Geophysical Research: Atmospheres*, *129*(2), e2023JD039543. https://doi.org/10.1029/2023JD039543

Tatzelt, C., Henning, S., Welti, A., Baccarini, A., Hartmann, M., Gysel-Beer, M., et al. (2022). Circum-Antarctic abundance and properties of CCN and INPs. *Atmospheric Chemistry and Physics*, *22*(14), 9721–9745. https://doi.org/10.5194/acp-22-9721-2022

Welti, A., Bigg, E. K., DeMott, P. J., Gong, X., Hartmann, M., Harvey, M., et al. (2020). Ship-based measurements of ice nuclei concentrations over the Arctic, Atlantic, Pacific and Southern oceans. *Atmospheric Chemistry and Physics*, *20*(23), 15191–15206. https://doi.org/10.5194/acp-20-15191-2020

Yin, J., Wang, D., & Zhai, G. (2012). An evaluation of ice nuclei characteristics from the long-term measurement data over North China. *Asia-Pacific Journal of Atmospheric Sciences*, *48*(2), 197–204. https://doi.org/10.1007/s13143-012-0020-8

---

## Referee Report (RR4)

**Comments:**

Thank you for addressing all the comments thoughtfully. I suggest publication following a few technical corrections. In Section 3.2.3, the discussion of Fig. 10 has some statistics listed in the text that do not match those written in Fig. 10. These include:

- mnMB for Fig. 10b is listed as 111% in the text, while in Fig. 10b it is listed as 198.01%
- mnMB for Fig. 10f is listed as 214% in the text, while in Fig. 10f it is listed as 103.31%

Please double check these calculations and update the text or Fig. 10 accordingly.

One final comment is that all the mnMB values given are positive, whereas usually negative biases indicate an underestimation, and positive biases indicate an overestimation. For example, both the dust and MPOA biases in Fig. 10a and 10b, respectively, are >100%, although the model simulations overall underestimate the total INP concentrations in both cases. This makes it harder to compare the model performance for the different scenarios shown in Fig. 10 and Fig. S7.

---

## Author Response (AR2)

We thank both the reviewers for the careful reading of our manuscript and their constructive comments that helped further improving the presentation of the manuscript.

Here-below our point-by-point replies to the reviewers' comments.

Reply to reviewer #1

*Comment: Why was new data added to Figure 10? There is substantially more data, which appears to have come from the addition of the Bigg datasets. Why was this added and why was it not included in the previous version? There is no explanation in the response.*

Response: Following the other reviewer comment, we have added datasets even if they did not fall inside the simulated period. These data (Bigg 1973 and 1990, Yin et al., 2012) have been compared with the climatological monthly mean simulations as explicitly stated in the caption of Figure 10.

Reply to reviewer #2

*Reviewer's statement: The updated figures and restructuring of the discussion greatly enhance the manuscript and increase clarity for the reader. The manuscript now also includes a more detailed and nuanced discussion of potential model biases, which is appreciated and helps put the current study in context and provides suggestions for future modeling and observational studies. I have only one broader comment and some suggested minor (mainly grammatical errors and typos) edits.*

Response: We thank the reviewer for the very positive comment and the careful reading of the manuscript.

The broader comment of the reviewer is addressed as follows:

*Comment: I still don't fully understand why 600 hPa was chosen as an example pressure level for Fig. 5. Although it is true that most INP measurements are made at a different temperature inside the instrument than the ambient temperature ([INP]T vs [INP]ambient), the aerosol being measured is still more representative of the boundary layer than the free troposphere, just at a different temperature than ambient ([INP]T). I agree that -20 °C is a reasonable temperature to choose to be representative of MPC glaciation, but at high latitudes -20 °C is reached within or very near to the boundary layer and the MPCs are typically not located in the free troposphere. Perhaps clarify that the pressure level chosen is representative of MPCs at low latitudes only, and not broadly representative of MPCs across all latitudes (lines 496-504). Much of the rest of the paper, and all the measurements being compared to in other figures are in the boundary layer, so it seems an odd choice to focus on the spatial distribution at a relatively high altitude, unless there is another reason I have missed.*

Response: We have modified part of this discussion that now reads: "The chosen pressure level is representative of the low free troposphere and average temperatures broadly consistent with those of the INP measurements. These conditions of temperature and pressure are representative of low latitude MPC's glaciation, where most of the INP is simulated to occur (Fig. 5)."

*Comment: The discussion of Fig. 5 is thorough, and the addition of the circled areas is helpful to follow the analyses. However, the mentions of "continental outflow" and "downwind of source areas", etc would make more sense for a lower altitude than 600 hPa, either near the surface, or at least within the boundary layer. Extensive long-range transport and mixing is expected for aerosols above the boundary layer, as well as a longer time since emission. There is no problem with discussing the results at this pressure/altitude, but trying to connect the results at 600 hPa to surface emissions perhaps needs to be more nuanced. This is particularly true since the high and low latitudes are discussed*

*together, but the fixed altitude used does not account for the changes in boundary layer height (temperature) or vertical mixing that occur between the equatorial regions and poles.*

Response: Indeed, advection and vertical mixing are important for heights above the boundary layer and these are taken into account in the model. We mentioned already this when discussing Fig. 5a (INP from dust): 'dust sources and long-range atmospheric transport pattern'
For clarity, we have rephrased the sentence mentioning 'continental outflow' as follows:
" $INP_D$[600hPa, -20$^o$C] shows also significant levels over the North Pacific, where dust is carried by continental outflows within the boundary layer or/and the free troposphere."

We have also added the following sentence that connects the discussion of Figure 5 (fixed pressure and temperature) with Figure 6 (averaged zonal mean profiles of INP number concentrations calculated at modelled temperature).
"Note that the spatial distribution of INP at a fixed pressure level and temperature shown in Fig. 5 and discussed above needs to be complemented by the vertical distribution of INP, which represents the combined effect of sources, long-range transport, vertical mixing and boundary layer height changes, that differ between low and high latitudes. In this context, figure 6 depicts…"

All minor suggested corrections have been considered in the revised version of the manuscript as can be seen in the track changes version. Figure captions have been rephrased for clarity. Fig 3 has not been redrawn as the original file is of excellent quality compared to the pdf version. The suggested corrections to the supplementary figures and the Table were implemented.

Regarding the comment on supplementary figure S1:
*Check the legend on this figure, for example, the "ISAC_CNR_2012_Antarctica" and "Yin_China_2012" data have the same symbol, and the Antarctic data doesn't appear to be plotted.*
*Are all the points marked as "Bigg_1969-1989" from Bigg's papers, or are some from Welti et al. (2020) previously unpublished data (eg Tan1502, SHIPPO, etc)? I thought Bigg's measurements largely ended ~140°E south of Australia. Also a note that Bigg's Southern Ocean measurements are much higher (2-3 orders of magnitude) than all modern measurements (see e.g. McCluskey et al., 2018; Moore et al., 2024; Tatzelt et al., 2022) and may not provide the best comparison dataset for simulations in 2009-2016. The reference to "Figure 4" was probably meant to be for Figure 10, and perhaps also Fig. S6 and S7.*

Response: The figure has been redrawn to avoid data with the same symbol. The caption has been also corrected. Note that the data points labelled "Bigg_1969-1989" are from Bigg's studies, accessible via the BACCHUS database at https://www.bacchus-env.eu/. While these measurements by Bigg are significantly higher—by two to three orders of magnitude— than contemporary data from sources such as McCluskey et al. (2018), Moore et al. (2024), and Tatzelt et al. (2022), they are nevertheless valuable for their extensive coverage of the Southern Hemisphere, particularly over the Southern Ocean. As can be seen in Figure 4 of the Bigg 1973 paper https://doi.org/10.1175/1520-0469(1973)030<1153:INCIRA>2.0.CO;2 there are data covering the area south of Australia from 60$^o$E to 140$^o$W. To our knowledge there are no modern measurements covering this large part of the hemisphere.

---

## Author Response (AR3)

Dear Editor,

We thank the reviewer for the time he/she has spent on our manuscript.

In reply to the reviewer's we have made the following changes, which helped improving the presentation of our results.

1. *about the extra more recent INP measurements,* we have now added the data from Welti et al 2020 including ACAPEX, NETCARE, CAPRICORN. Figure 10 and supplementary figures S7 and Figure S1 have been redrawn including these data. Table S1 has been also updated.

2. *Uncertainties in observations*: The differences of magnitude between the earlier and the most recent observations is now mentioned in the manuscript at the end of the first paragraph of section 3.2.3 where Figure 10 is discussed in lines 671-675 of the revised manuscript:

Note also that the climatological data by Bigg (1973, 1990), although very useful due to their geographic coverage, they are by about 2-3 orders of magnitude higher than recent observations made closer to the period simulated in our study (i.e. McCluskey et al. (2018), Tatzelt et al. (2022), Moore et al. (2024)). This difference has to be kept in mind when comparing model results with past observations.

3. *Statistics in Figure 10:* Figure 10 has been redrawn to include both the additional observations and the normalized mean bias. Similarly Figure S7 has been redrawn and the same statistics are provided. The statistics provided in section 3.2.3 have been updated accordingly and modified normalized mean bias (mnMB) values are now discussed.

4. *Duplications in section 2.3.2, last two paragraphs of that section in the earlier manuscript:* Lines 709-715 of the earlier manuscript, all this paragraph has been changed as follows and merged with the last paragraph of the section:

The discussion on the overprediction of INP by the PBAP parameterization has been moved earlier together with the reference to figure S7, which supports such conclusion. (lines 708-710 of the revised manuscript).

The next two sentences have been removed to avoid duplications and in the last paragraph a comment on the need for improved representation of INP of biological origin has been added.

Lines 732-734 of the revised version: Without improved representations of the sources and ice-nucleating activities of biological INP, models will struggle to simulate total INP concentrations at warmer temperatures and the resulting MPC.

The small typos in the supplement have been corrected.

All the above changes have improved the presentation of our results but did not alter the conclusions of the work.

Finally, we took note of the *recommendation of the reviewer* for a sensitivity study in a future article, which we will try to implement indeed in a future study.

Furthermore, Figure S3 haa been redrawn for clarity.

We hope that with these changes our manuscript is now suitable for publication in ACP.

Maria Kanakidou on the behalf of the co-authors

---

## Author Response (AR4)

Dear editor,

We sincerely thank the reviewer for the careful reading of the manuscript

We have addressed the comments as follows:

**Reviewer Comments:**

Thank you for addressing all the comments thoughtfully. I suggest publication following a few technical corrections. In Section 3.2.3, the discussion of Fig. 10 has some statistics listed in the text that do not match those written in Fig. 10. These include: • mnMB for Fig. 10b is listed as 111% in the text, while in Fig. 10b it is listed as 198.01% • mnMB for Fig. 10f is listed as 214% in the text, while in Fig. 10f it is listed as 103.31% Please double check these calculations and update the text or Fig. 10 accordingly. One final comment is that all the mnMB values given are positive, whereas usually negative biases indicate an underestimation, and positive biases indicate an overestimation. For example, both the dust and MPOA biases in Fig. 10a and 10b, respectively, are >100%, although the model simulations overall underestimate the total INP concentrations in both cases. This makes it harder to compare the model performance for the different scenarios shown in Fig. 10 and Fig. S7.

**Response to Reviewer Comment:**

We thank the reviewer for the thorough review and for pointing out these inconsistencies in the reported mnMB values in Section 3.2.3 and Figure 10. Indeed, the observed and predicted arrays were inadvertently inverted during the computation of the Mean Normalized Bias. This is now corrected and the mnMB values have been updated both in the main text and in Figure 10. We have also updated supplementary figure S7. The updated values show positive mnMB indicating model overestimation and negative mnMB indicating underestimation, in line with standard practice.

Please find below the corrected Figures 10 and S7 and the revised text in Section 3.2.3.

We sincerely appreciate the attention of the reviewer to this detail, which helped improve the accuracy of our manuscript.

Changes in the manuscript:

Line 660: Considering dust minerals as the sole INP types leads to underestimation against observations (Fig. 10a; Pt1 about 25%, R=0.84 and mnMB **about -128%)**.

Line 669: ..for dust alone to about 65% for MPOA alone with almost the same mnMB (about -75%) (Pt1, Fig. 10b).

Line 677: ... decreases compared to INPMPOA (Pt1 ~65%), while it increases the mnMB (about 75%).

Line 692:  ...correlation coefficient of 0.85, about 59% (Pt1) predictability and mnMB of about **110%** (Fig. 10f)

[Figure]

*Figure 10: Comparison of INP concentrations calculated at the temperature of the measurements against observations accounting for simulated mineral dust (a), MPOA (b), PBAP (c), mineral dust and MPOA (d), PBAP and mineral dust (e), and mineral dust, MPOA and PBAP (f). The dark grey dashed lines represent   one order of magnitude difference between modelled and observed concentrations, and the light-grey dashed lines depict 1.5 orders of magnitude. The simulated values correspond to monthly mean concentrations, and the error bars correspond to the error of the observed monthly mean INP values. The color bars show the corresponding instrument temperature of the measurement in Celsius (a-f). Pt1 and Pt1.5 are the percentages of data points reproduced by the model within an order of magnitude and 1.5 orders of magnitude, respectively. R is the correlation coefficient, which is calculated with the logarithm of the values. Diamonds correspond to measurements (Bigg, 1973, 1990; Yin et al., 2012) that are compared with the climatological monthly mean simulations. Circles indicate comparisons between temporally and spatially co-located observations and model results. The location of the observations is shown in Fig. S1 and Table S1.*

[Figure]

*Figure S7: Comparison of INP concentrations calculated at the temperature of the measurements against total INP observations accounting for mineral dust (yellow), MPOA (blue), PBAP (green) separated in high, middle and low latitudes.*